# Highly oxygenated organic molecules (HOM) formation in the isoprene oxidation by NO₃ radical

Defeng Zhao[1, 2, 3, 4], Iida Pullinen[2, a], Hendrik Fuchs[2], Stephanie Schrade[2], Rongrong Wu[2], Ismail-Hakki Acir[2, b], Ralf Tillmann[2], Franz Rohrer[2], Jürgen Wildt[2], Yindong Guo[1], Astrid Kiendler-Scharr[2], Andreas Wahner[2], Sungah Kang[2], Luc Vereecken[2], Thomas F. Mentel[2]

[1]Department of Atmospheric and Oceanic Sciences & Institute of Atmospheric Sciences, Fudan University, Shanghai, 200438, China;
[2]Institute of Energy and Climate Research, IEK-8: Troposphere, Forschungszentrum Jülich, 52425, Jülich, Germany
[3]Big Data Institute for Carbon Emission and Environmental Pollution, Fudan University, Shanghai, 200438, China
[4]Institute of Eco-Chongming (IEC), 20 Cuiniao Rd., Chenjia Zhen, Chongming, Shanghai 202162, China
[a]Now at: Department of Applied Physics, University of Eastern Finland, Kuopio, 7021, Finland.
[b]Now at: Institute of Nutrition and Food Sciences, University of Bonn, Bonn, 53115, Germany;

*Correspondence to*: Thomas F. Mentel (t.mentel@fz-juelich.de), Defeng Zhao (dfzhao@fudan.edu.cn)

## Abstract

Highly oxygenated organic molecules (HOM) are found to play an important role in the formation and growth of secondary organic aerosol (SOA). SOA is an important type of aerosol with significant impact on air quality and climate. Compared with the oxidation of volatile organic compounds by ozone ($O_3$) and hydroxyl radical (OH), HOM formation in the oxidation by nitrate radical ($NO_3$), an important oxidant at night-time and dawn, has received less attention. In this study, HOM formation in the reaction of isoprene with $NO_3$ was investigated in the SAPHIR chamber (Simulation of Atmospheric PHotochemistry In a large Reaction chamber). A large number of HOM including monomers ($C_5$), dimers ($C_{10}$), and trimers ($C_{15}$), both closed-shell compounds and open-shell peroxy radicals ($RO_2$), were identified and were classified into various series according to their formula. Their formation pathways were proposed based on the peroxy radicals observed and known mechanisms in the literature, which were further constrained by the time profiles of HOM after sequential isoprene addition to differentiate first- and second-generation products. HOM monomers containing one to three N atoms (1-3N monomers) were formed, starting with $NO_3$ addition to carbon double bond, forming peroxy radicals, followed by autoxidation. 1N monomers were formed by both the direct reaction of $NO_3$ with isoprene and of $NO_3$ with first-generation products. 2N-monomers (e.g. $C_5H_8N_2O_{n\ (n=7-13)}$, $C_5H_{10}N_2O_{n\ (n=8-14)}$) were likely the termination products of $C_5H_9N_2O_n\bullet$, which was formed by the addition of $NO_3$ to C5-hydroxynitrate ($C_5H_9NO_4$), a first-generation product containing one carbon double bond. 2N-monomers, which were second-generation products, dominated in monomers and accounted for ~34% of all HOM, indicating the important role of second-generation oxidation in HOM formation in the isoprene+$NO_3$ reaction under our experimental conditions. H-shift of alkoxy radicals to form peroxy radicals and subsequent autoxidation ("alkoxy-peroxy" pathway) was found to be an important pathway of HOM formation. HOM dimers were mostly formed by the accretion reaction of various HOM monomer $RO_2$ and via the termination reactions of dimer $RO_2$ formed by further reaction of closed-shell dimers with $NO_3$ and possibly by the reaction of C5-$RO_2$ with isoprene. HOM trimers were likely formed by the accretion reaction of dimer $RO_2$ with monomer $RO_2$. The concentrations of different HOM showed distinct time profiles during the reaction, which was linked to their formation pathway. HOM concentrations either showed a typical time profile of first-generation products, or of second-generation products, or a combination of both, indicating multiple formation pathways and/or multiple isomers. Total HOM molar yield was estimated to be $1.2\%^{+1.3\%}_{-0.7\%}$, which corresponded to a SOA yield of ~3.6% assuming the molecular weight of $C_5H_9NO_6$

as the lower limit. This yield suggests that HOM may contribute a significant fraction to SOA yield in the reaction

of isoprene with $NO_3$.

## 1    Introduction

Highly oxygenated organic molecules (HOM) are an important class of compounds formed in the oxidation of volatile of organic compounds (VOC) including biogenic VOC (BVOC) and anthropogenic VOC (Crounse et al., 2013; Ehn et al., 2014; Jokinen et al., 2014; Rissanen et al., 2014; Jokinen et al., 2015; Krechmer et al., 2015; Mentel et al., 2015; Rissanen et al., 2015; Kenseth et al., 2018; Molteni et al., 2018; Garmash et al., 2019; McFiggans et al., 2019; Molteni et al., 2019; Quelever et al., 2019). A number of recent studies have demonstrated that HOM play a pivotal role in both nucleation and also particle growth of pre-existing particles, thus contributing to secondary organic aerosol (SOA) (Ehn et al., 2014; Kirkby et al., 2016; Tröstl et al., 2016). Particularly, in the early stage of aerosol growth, HOM may contribute a significant fraction of SOA mass (Tröstl et al., 2016).

HOM are formed by the autoxidation of peroxy radicals ($RO_2$), which means they undergo intramolecular H-shift forming alky radicals, followed by $O_2$ addition leading to formation of new $RO_2$ as shown below (Vereecken et al., 2007; Crounse et al., 2013; Ehn et al., 2017; Bianchi et al., 2019; Møller et al., 2019; Nozière and Vereecken, 2019; Vereecken and Nozière, 2020).

Besides autoxidation, the $RO_2$ can also react with $HO_2$, $RO_2$ and $NO_3$, either forming a series of termination products (R1-3), including organic hydroxyperoxide, alcohol, and carbonyl, or forming alkoxy radicals (RO, R4-5) via the following reactions.

$$RO_2 + R'O_2 \rightarrow ROH + R=O \tag{R1}$$

$$RO_2 + R'O_2 \rightarrow R=O + R'OH \tag{R2}$$

$$RO_2 + HO_2 \rightarrow ROOH \tag{R3}$$

$$RO_2 + R'O_2 \rightarrow RO + R'O + O_2 \tag{R4}$$

$$RO_2 + NO_3 \rightarrow RO + NO_2 + O_2 \tag{R5}$$

$$RO_2 + R'O_2 \rightarrow ROOR' + O_2 \tag{R6}$$

The termination products are detected in the mass spectra at masses M+1, M-15, M-17 respectively with M being the molecular mass of the parent $RO_2$ (Ehn et al., 2014; Mentel et al., 2015). In case that $RO_2$ is an acyl peroxy radical, percarboxylic acids and carboxylic acids are formed instead of hydroperoxides and alcohols in R3 and R1, respectively (Atkinson et al., 2006; Mentel et al., 2015). $RO_2$ can also form HOM dimers by the accretion reaction of two $RO_2$ (R6) (Berndt et al., 2018a; Berndt et al., 2018b; Valiev et al., 2019). Additionally, HOM can be formed via H-shift in RO followed by $O_2$ addition (referred to as "alkoxy-proxy" pathway) (Finlayson-Pitts and Pitts, 2000; Vereecken and Peeters, 2010; Vereecken and Francisco, 2012; Mentel et al., 2015). These pathways are summarized in a recent comprehensive review (Bianchi et al., 2019), which also further clarifies HOM definition.

Currently, most laboratory studies of HOM formation focus on the VOC oxidation by OH and $O_3$ (Crounse et al., 2013; Ehn et al., 2014; Jokinen et al., 2014; Rissanen et al., 2014; Jokinen et al., 2015; Krechmer et al., 2015; Mentel et al., 2015; Rissanen et al., 2015; Kirkby et al., 2016; Tröstl et al., 2016; Kenseth et al., 2018; Molteni et al., 2018; Garmash et al., 2019; McFiggans et al., 2019; Molteni et al., 2019; Quelever et al., 2019; Wang et al., 2020; Yan et al., 2020). HOM formation in the oxidation of VOC with $NO_3$ has received much less attention. $NO_3$ is another important oxidant of VOC mainly operating during nighttime. Particularly, $NO_3$ has high reactivity with unsaturated BVOC such as monoterpene and isoprene. It is often the dominant oxidant of these compounds at night, especially in regions where biogenic and anthropogenic emissions mix (Geyer et al., 2001; Brown et al., 2009; Brown et al., 2011). The reaction products contribute to SOA formation (Xu et al., 2015; Lee et al., 2016). Also, the organic nitrates produced in these reactions play an important role in nitrogen chemistry by altering $NO_x$ concentration, which further influences photochemical recycling and ozone formation in the next day. Among these reaction products, HOM can also be formed (Xu et al., 2015; Lee et al., 2016; Yan et al., 2016). Despite the potential importance, studies of HOM formation in the oxidation of BVOC by $NO_3$ are still limited compared with the HOM formation via oxidation by $O_3$ and OH. Although a number of laboratory studies have investigated the reaction of $NO_3$ with BVOC (Ng et al., 2008; Fry et al., 2009; Rollins et al., 2009; Fry et al., 2011; Kwan et al., 2012; Fry et al., 2014; Boyd et al., 2015; Schwantes et al., 2015; Nah et al., 2016; Boyd et al., 2017; Claflin and Ziemann, 2018; Faxon et al., 2018; Draper et al., 2019; Takeuchi and Ng, 2019; Novelli et al., 2021; Vereecken et al., 2021), these studies mostly focus on either SOA yield and composition, or on the gas-phase chemistry mechanism mainly for "traditional" oxidation products that stem from few oxidation steps.

Importantly, HOM formation in the reaction of $NO_3$ with isoprene, the most abundant BVOC accounting for more than half of the global BVOC emissions, has not been explicitly addressed yet, to the best of our knowledge. Although isoprene from plants are mainly emitted under light conditions, i.e., in the daytime, isoprene can remain high after sunset in significant concentrations (Starn et al., 1998; Stroud et al., 2002; Brown et al., 2009) because of the reduced consumption by OH and is found to decay rapidly. A substantial fraction of isoprene can then be oxidized by $NO_3$ (Brown et al., 2009). Regarding the budget of $NO_3$, the reaction of isoprene with $NO_3$ can contribute to a significant or even dominant fraction of $NO_3$ loss at night in regions where VOC is dominated by isoprene such as Northeast US (Brown et al., 2009). Under some circumstances, the reaction of isoprene with $NO_3$ can contribute to a significant fraction during the afternoon and afterwards (Ayres et al., 2015; Hamilton et al., 2021). The reaction of isoprene with $NO_3$ is the subject of a number of studies (Ng et al., 2008; Perring et al., 2009; Rollins et al., 2009; Kwan et al., 2012; Schwantes et al., 2015; Vereecken et al., 2021). These studies focus on the oxidation mechanism and "traditional" oxidation products, as well as SOA yields. The initial step is the $NO_3$ addition to one of the C=C double bounds, preferentially to the carbon C1 (Schwantes et al., 2015), followed by $O_2$ addition forming a nitrooxyalkyl peroxy radical ($RO_2$). This $RO_2$ can undergo the reactions described above, forming a series of products such as C5-nitrooxyhydroperoxide, C5-nitrooxycarbonyl, and C5-hydroxynitrate (Ng et al., 2008; Kwan et al., 2012), as well as methyl vinyl ketone (MVK), potentially methacrolein (MACR), formaldehyde, OH radical, and $NO_2$ as minor products (Schwantes et al., 2015). A high

nitrate yield (57-95%) was found (Perring et al., 2009; Rollins et al., 2009; Kwan et al., 2012; Schwantes et al.,
2015). Products in the particle phase such as $C_{10}$ dimers were also detected (Ng et al., 2008; Kwan et al., 2012;
Schwantes et al., 2015). The SOA yield varies from 2% to 23.8% depending on the organic aerosol concentration
(Ng et al., 2008; Rollins et al., 2009). These studies have provided valuable insights in oxidation mechanism,
particle yield and composition. However, because HOM formation was not the focus of these studies, only a
limited number of products, mainly moderately oxygenated ones (oxygen number ≤2 in addition to $NO_3$
functional groups), were detected in the gas phase. The detailed mechanism of HOM formation and their yields
in the reaction of BVOC+$NO_3$ are still unclear.

In this study, we investigated the HOM formation in the oxidation of isoprene by $NO_3$. We report the

identification of HOM, including HOM monomers, dimers, and trimers. According to the reaction products and
literature, we discuss the formation mechanism of these HOM. The formation mechanism of various HOM is
further constrained with time series of HOM upon repeated isoprene additions. We also provide an estimate of
HOM yield in the isoprene+$NO_3$ reaction and assess their roles in SOA formation.
**2    Experimental**
**2.1    Chamber setup and experiments**

Experiments investigating the reaction of isoprene with $NO_3$ were conducted in the SAPHIR chamber

(Simulation of Atmospheric PHotochemistry In a large Reaction chamber) at Forschungszentrum Jülich,
Germany. The details of the chamber have been described before (Rohrer et al., 2005; Zhao et al., 2015a; Zhao
et al., 2015b; Zhao et al., 2018). Briefly, SAPHIR is a Teflon chamber with a volume of 270 $m^3$. It can utilize
natural sunlight for illumination and is equipped with a louvre system to switch between light and dark
conditions. In this study, the experiments were conducted in the dark with the louvres closed.

Temperature and relative humidity were continuously measured. Gas and particle phase species were

characterized using a comprehensive set of instruments with the details described before (Zhao et al., 2015b).
VOC were characterized using a Proton Transfer Reaction Time-of-Flight Mass Spectrometer (PTR-ToF-MS,
Ionicon Analytik, Austria). $NO_x$ and $O_3$ concentrations were measured using a chemiluminescence $NO_x$ analyzer
(ECO PHYSICS TR480) and an UV photometer $O_3$ analyzer (ANSYCO, model O341M), respectively. OH,
$HO_2$ and $RO_2$ concentrations were measured using a laser induced fluorescence system (LIF) (Fuchs et al., 2012).
$NO_3$ and $N_2O_5$ were detected by a custom-built instrument based on cavity ring-down spectroscopy. The design
of the instrument is similar to that described by Wagner et al. (2011). $NO_3$ was directly detected in one cavity
by its absorption at 662 nm and the sum of $NO_3$ and $N_2O_5$ in a second, heated cavity, which had a heated inlet
to thermally decompose $N_2O_5$ to $NO_3$. The sampling flow rate was 3 to 4 liters per minute. The detection by
cavity ring-down spectroscopy was achieved by a diode laser that was periodically switched on and off with a
repetition rate of 200 Hz. Ring-down events were observed by a digital oscilloscope PC card during the time
when the laser was switched off and were averaged over 1s. The zero-decay time that is needed to calculate the
concentration of $NO_3$ was measured every 20 s by chemically removing $NO_3$ in the reaction with excess nitric
oxide (NO) in the inlet system. The accuracy of measurements was limited by the uncertainty in the correction
for inlet losses of $NO_3$ and $N_2O_5$. In the case of $N_2O_5$ a transmission of $(85\pm10)$ % was achieved and in the case
of $NO_3$ of $(50\pm30)$ %.
Before an experiment, the chamber was flushed with high purity synthetic air (purity>99.9999% $O_2$ and $N_2$).
Experiments were conducted under dry condition (RH<2 %) and temperature was at $302\pm3$ K. $NO_2$ and $O_3$ were
added to the chamber first to form $N_2O_5$ and $NO_3$, reaching concentrations of ~60 ppb for $NO_2$ and ~100 ppb for $O_3$.
After around half an hour, isoprene was sequentially added into the chamber for three times at intervals of ~1 h.
Around 40 min after the third isoprene injection, $NO_2$ was added to compensate the loss of $NO_3$ and $N_2O_5$. Afterwards,
three isoprene additions were repeated in the same way as before. $O_3$ was added before the fifth and the sixth isoprene
addition to compensate for its loss by reaction. The schematic for the experimental procedure is shown in Fig. S1.
Experiments were designed such that the chemical system was dominated by the reaction of isoprene with $NO_3$ and
the reaction of isoprene with $O_3$ did not play a major role (<3% of the isoprene consumption). Figure S2 shows the
relative contributions of the reaction of $O_3$ and $NO_3$ with isoprene to the total chemical loss of isoprene using the
$NO_3$ and $O_3$ concentrations measured. The reaction with $NO_3$ accounted for >95% of the isoprene consumption for
the whole experiments. The contribution of the reaction of isoprene with trace amount of OH, mainly produced in
the reaction of isoprene+$O_3$ via Criegee intermediates (Nguyen et al., 2016), is negligible as the OH yield is less than
one (Malkin et al., 2010) and thus its contribution is less than that of isoprene+$O_3$. This is consistent with the
contribution determined using measured OH concentration, despite some uncertainty in measured OH concentration
due to the interference from $NO_3$. In these experiments, $RO_2$ fate is estimated to be dominated by its reaction with
$NO_3$ according to the measured $NO_3$, $RO_2$, and $HO_2$ concentration and their rate constants for the reactions with $RO_2$
(MCM v3.2(Jenkin et al., 1997; Jenkin et al., 2003; Saunders et al., 2003; Jenkin et al., 2015), via website:
http://mcm.leeds.ac.uk/MCM) despite uncertainties of the measured $RO_2$ and $HO_2$ concentration due to interference
from $NO_3$. As a large portion of $RO_2$ is not measured by LIF (Vereecken et al., 2021) and thus $RO_2$ is underestimated,
we expected the reaction of $RO_2$+$RO_2$ to be also important. Overall, we estimate that he $RO_2$ fate is dominated the
reaction $RO_2$+$NO_3$ with significant contribution of $RO_2$+$RO_2$.
**2.2    Characterization of HOM**
In this study we refer to similar definition for HOM by Bianchi et al. (2019), i.e., HOM typically contain six or
more oxygen atoms formed via autoxidation and related chemistry of peroxy radicals. HOM were detected using a
Chemical Ionization time-of-flight Mass Spectrometer (Aerodyne Research Inc., USA) with nitrate as the reagent ion
(CIMS) (Eisele and Tanner, 1993; Jokinen et al., 2012). $^{15}$N nitric acid was used to produce $^{15}NO_3^-$ in order to
distinguish the $NO_3$ group in target molecules formed in the reaction from the reagent ion. The details of the
instrument are described in our previous publications (Ehn et al., 2014; Mentel et al., 2015; Pullinen et al., 2020).
The CIMS has a mass resolution of ~4000 (m/dm). Examples of peak fitting are shown in Fig. S3. HOM
concentrations were estimated using the calibration coefficient of $H_2SO_4$ as described by Pullinen et al. (2020)
because the charge efficiency of HOM and $H_2SO_4$ can be assumed to be equal and close to the collision limit (Ehn et
al., 2014; Pullinen et al., 2020). The details of the calibration with $H_2SO_4$ are provided in the supplement S1. Since
HOM contain more than six oxygen atoms and their clusters with nitrate ions are quite stable (Ehn et al., 2014), the
charge efficiency of HOM is thus assumed to be equal to that of $H_2SO_4$, which is close to the collision limit (Viggiano
et al., 1997). If HOM do not charge with nitrate ions at their collision limit or the clusters formed break during the
short residence time in the charger, its concentration would be underestimated as pointed by Ehn et al. (2014). Thus,
our assumption provides a lower limit of the HOM concentration. The HOM yield was derived using the
concentration of the HOM produced, divided by the concentration of isoprene that was consumed by $NO_3$. The
uncertainty of HOM yield was estimated to -55%/+103%. The loss of HOM to the chamber was corrected using a
wall loss rate of $6\times10^{-4}$ $s^{-1}$ as quantified previously (Zhao et al., 2018). HOM concentrations were also corrected for
dilution due to the replenishment flow needed to maintain a constant overpressure of the chamber (loss rate $\sim1\times10^{-6}$
$s^{-1}$) (Zhao et al., 2015b). The influence of wall loss correction and dilution correction on HOM yield was ~12% and
<1%, respectively. Although the wall loss rate of vapors in this study might not be exactly the same as in our previous
photo-oxidation experiments (Zhao et al., 2018), HOM yield is not sensitive to the vapor wall loss rate. An increase
of wall loss rate by 100% or a decrease by 50% only changes the HOM yield by 11% and -6%, respectively.
**3    Results and discussion**
**3.1    Overview of HOM**
The mass spectra of HOM in the gas phase formed in the oxidation of isoprene by $NO_3$ are shown in
Fig. 1. A large number of HOM were detected. Almost all peaks are assigned HOM containing nitrogen atoms
with possibly few exceptions such as $C_5H_{10}O_8$ and $C_5H_8O_{11}$ with very minor peaks (<~1% of the maximum
peak). The reaction products can be roughly divided into three classes: monomers (C5, ~200-400 Th), dimers
(C10, ~400-600 Th), and trimers (C15, ~>600 Th), according to their mass to charge ratio (m/z). The detailed
peak assignment of monomers, dimers, and trimers is discussed in the following sections.
**3.2    HOM monomers and their formation**
**3.2.1    Overview of HOM monomers**
HOM monomers showed a roughly repeating pattern in the mass spectrum at every 16 Th
(corresponding to the mass of oxygen) (Fig. 1a). Here a number of series of HOM monomers with continuously
increasing oxygenation were found, such as $C_5H_9NO_n$, $C_5H_7NO_n$, $C_5H_8N_2O_n$, $C_5H_{10}N_2O_n$ (Table 1, Table S1-2
and Fig. 2). These monomers included both stable closed-shell molecules and open-shell radicals, such as
$C_5H_8NO_n\bullet$ and $C_5H_9N_2O_n\bullet$. The open-shell molecules were likely $RO_2$ radicals because of their much longer life
time and hence higher concentrations compared with alkoxy radicals (RO) and alkyl radicals (R). Since the
observed stable products were mostly termination products of $RO_2$ reactions, we describe the stable products in
a $RO_2$-oriented approach. It is worth noting that some of the termination products may contain multiple isomers
formed from different pathways.

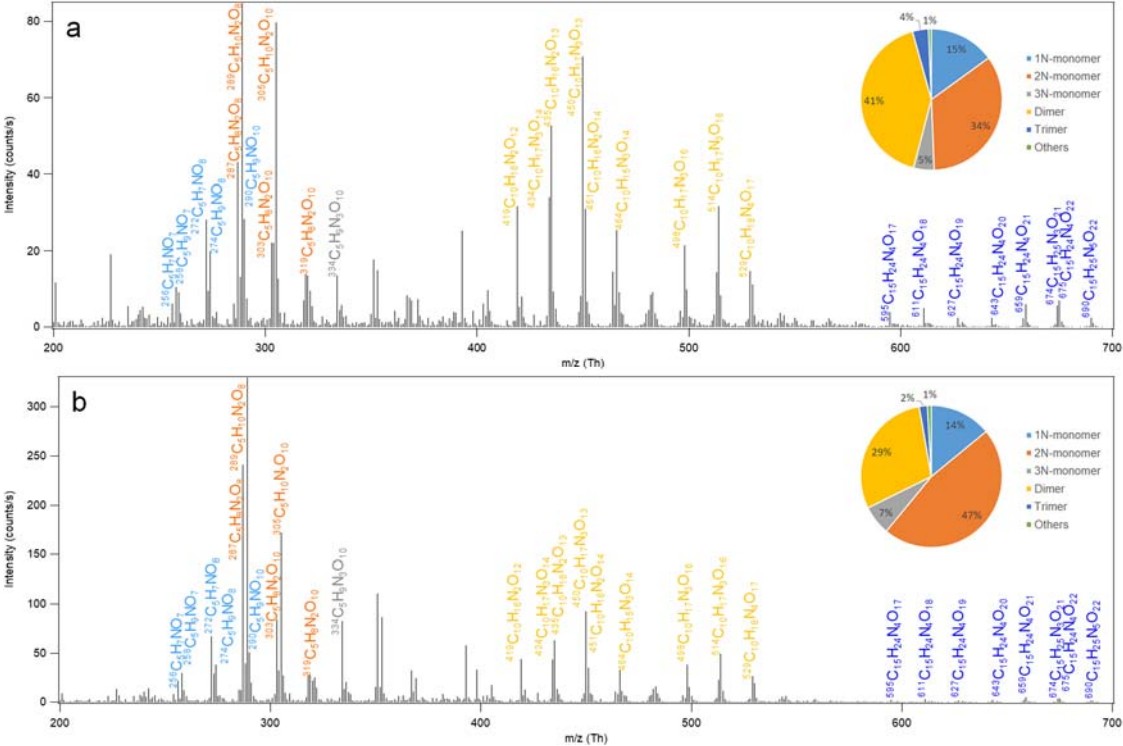

Figure 1. Mass spectrum of the HOM formed in the oxidation of isoprene by $NO_3$. HOM are detected as clusters with the reagent ion $^{15}NO_3^-$, which is not shown in the molecular formula in the figure for simplicity. Panel a and b show the average spectrum during the first isoprene addition period (P1) and for the whole period of six isoprene additions (P1-6), respectively. The insets show the contributions of different classes of HOM. 1-3N-monomer refers to the monomers containing 1-3 nitrogen atoms in the molecular formula.

HOM monomers were classified into 1N-, 2N-, and 3N-monomers according to the number of nitrogen atoms that they contain. HOM without nitrogen atoms were barely observed except for very minor peaks (<~1% of the maximum peak) possibly assigned to $C_5H_{10}O_8$ and $C_5H_8O_{11}$. The contribution of 2N-monomers such as $C_5H_{10}N_2O_n$ and $C_5H_8N_2O_n$ was higher than that of the 1N-HOM monomers, and that of 3N-monomers was the least (Fig. 1, inset). The most abundant monomers were $C_5H_{10}N_2O_8$, $C_5H_{10}N_2O_9$, and $C_5H_8N_2O_8$. The termination products of $C_5H_9NO_8$, $C_5H_9NO_9$, and $C_5H_7NO_8$ also showed relatively high abundance. These limited number of compounds dominated the HOM monomers. Since 2N-monomers were second-generation products as discussed below, the higher abundance 2N- monomers indicate that the second-generation HOM play an important role in the reaction of $NO_3$ with isoprene in the reaction conditions of our study, as also seen by Wu et al. (2020) . This is more evident for the mass spectrum averaged over six isoprene addition periods (Fig. 1b), where the abundance of $C_5H_{10}N_2O_n$ and $C_5H_8N_2O_n$ were more dominant. This observation is in contrast with the finding for the reaction of $O_3$ with BVOC which contains only one double bond such as α-pinene (Ehn et al., 2014), where HOM are mainly first-generation products formed via autoxidation. The higher abundance of HOM 2N-monomers than 1N-monomers is likely because HOM production rate via the autoxidation of 1N-monomer $RO_2$ following the reaction of isoprene with $NO_3$ may be slower than that of the reaction of 1N-monomers (including both HOM and non-HOM monomers) with $NO_3$. We would like to note that some less oxygenated 1N-monomers such as $C_5H_9NO_{4/5}$ and $C_5H_7NO_4$ may have high abundance but are not detected by $NO_3^-$-CIMS and are not HOM and thus not included in HOM 1N-monomers.

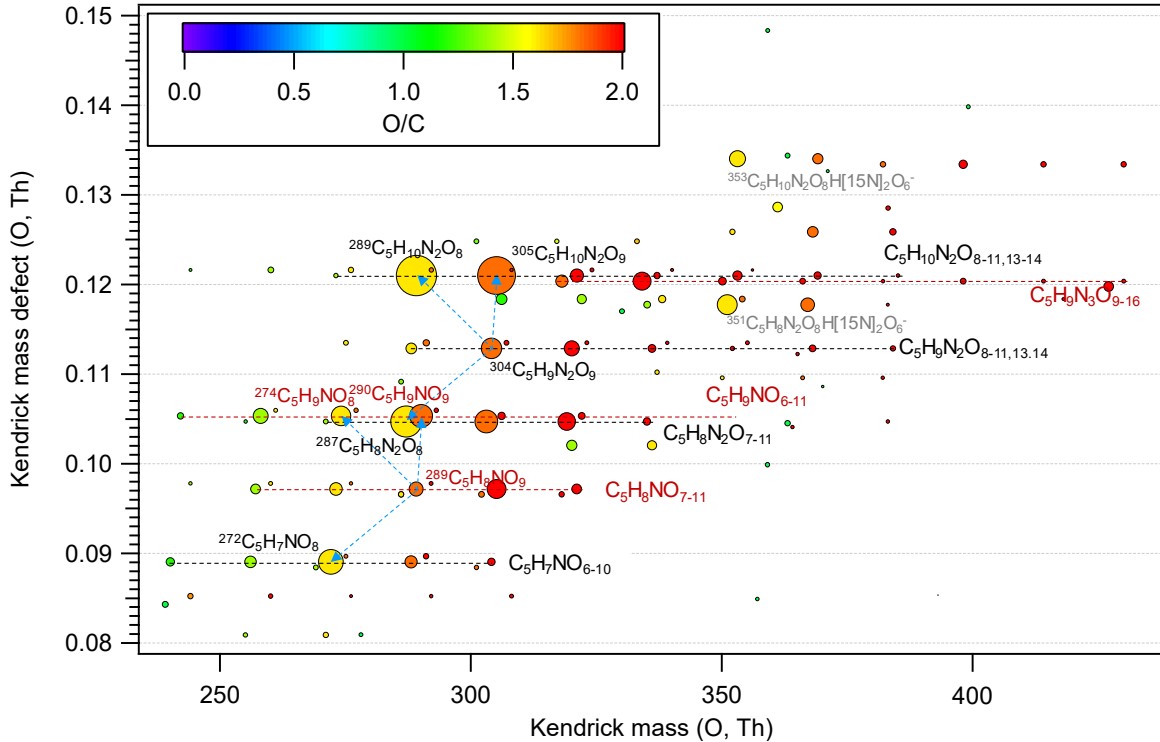


Figure 2. Kendrick mass defect plot for O of HOM monomers. The m/z in the molecular formula include the
reagent ion $^{15}NO_3^-$, which is not shown for simplicity. The size (area) of circles is set to be proportional to the average
peak intensity of each molecular formula during the first isoprene addition period (P1). The species at m/z 351 and 353
(labelled in grey) are the adducts of $C_5H_8N_2O_8$ and $C_5H_{10}N_2O_8$ with $H[^{15}N]_2O_6^-$, respectively. The blue dashed lines with
arrows indicate the termination product hydroperoxide (M+H), alcohol (M-O+H), and ketone (M-O-H) with M the
molecular formula of a HOM $RO_2$.
**3.2.2    1N-monomers**
In our experiments we observed a $C_5H_8NO_n\bullet$ (n=7-12) series (series M1), as well as its corresponding
termination products $C_5H_7NO_{n-1}$, $C_5H_9NO_{n-1}$, and $C_5H_9NO_n$ via the reactions with $RO_2$ and $HO_2$, which contain
carbonyl, hydroxyl, and hydroperoxy group, respectively. Overall, the peak intensities of $C_5H_9NO_n$ and
$C_5H_7NO_n$ series first increased and then decreased as oxygen number increased (Fig. 2), with the peak intensity
of $C_5H_9NO_8$ and $C_5H_7NO_8$ being the highest within their respective series when averaged over the whole
experiment period.
Table 1. HOM monomers formed in the oxidation of isoprene by $NO_3$.

| Series Number | Product | Type[a] | Pathway of $RO_2$ |
| --- | --- | --- | --- |
| M1a/b | $C_5H_8NO_{n\ (n=7-11)}$ | $RO_2$ | Isoprene+$NO_3$ |
| | $C_5H_9NO_{n\ (n=6-11)}$ | ROOH/ROH | Isoprene+$NO_3$+$NO_3$ |
| | $C_5H_7NO_{n\ (n=6-10)}$ | R=O | |
| M2a/b | $C_5H_9N_2O_{n\ (n=8-11,13,14)}$[b] | $RO_2$ | |
| | $C_5H_{10}N_2O_{n\ (n=8-11,13,14)}$[b] | ROOH/ROH | Isoprene +$NO_3$+$NO_3$ |
| | $C_5H_8N_2O_{n\ (n=7-11)}$ | R=O | |

| | | | |
|---|---|---|---|
| | $C_5H_9N_3O_n$ (n=9-16)[b] | $RO_2NO_2$ | |
| M3 | $C_5H_7N_2O_n$ (n=9) | $RO_2$ | Isoprene $+NO_3+NO_3$ |
| | $C_5H_8N_2O_n$ (n=8, 9) | ROOH/ROH | |
| | $C_5H_6N_2O_n$ (n=8) | R=O | |
| M4 | $C_5H_{10}NO_n$ (n=8-9) | $RO_2$ | Isoprene $+NO_3+OH$ |
| | $C_5H_{11}NO_n$ (n=7-9) | ROOH/ROH | |
| | $C_5H_9NO_n$ (n=7-8) | R=O | |

[a]: $RO_2$ denotes peroxy radical and ROOH, ROH, R=O, and $RO_2NO_2$ denote the termination products
containing hydroperoxy, hydroxyl, carbonyl group, and peroxynitrate, respectively.
[b]: Peak assignment of compounds with n=13,14 may be subject to uncertainties.

Scheme 1(a):
$C_5H_8$ → (NO₃, O₂) → $C_5H_8NO_5\bullet$ → (H-shift, O₂) → $C_5H_8NO_7\bullet$ → (Cyclization, O₂) → $C_5H_8NO_9\bullet$ → (H-shift) → $C_5H_8NO_9\bullet$ → (H-shift) → $C_5H_8NO_9\bullet$ → (Decompose) → $C_5H_8NO_9\bullet$ → (H-shift) → $C_5H_8NO_9\bullet$ → (H-shift, O₂) → $C_5H_8NO_{11}\bullet$

(a)

Scheme 1(b):
$C_5H_8NO_9\bullet$ → (RO₂/NO₃) → $C_5H_8NO_8\bullet$ → (H-shift) → $C_5H_8NO_8\bullet$ → (H-shift) → $C_5H_8NO_8\bullet$ → (Decompose) → $C_5H_8NO_8\bullet$ → (H-shift) → $C_5H_8NO_8\bullet$ → (H-shift, O₂) → $C_5H_8NO_{10}\bullet$

(b)
Scheme 1. The example pathways to form HOM $RO_2$ $C_5H_8NO_n\bullet$ (n=7, 9, 11) series (a) and $C_5H_8NO_n\bullet$
(n=8, 10) series (b) in the reaction of isoprene with $NO_3$. The detected products are in bold.
$C_5H_8NO_n\bullet$ with odd number oxygen atoms (n=7, 9, 11, series M1a) were possibly formed by the attack
of $NO_3$ to one double bond (preferentially to C1 according to previous studies (Skov et al., 1992; Berndt and
Böge, 1997; Schwantes et al., 2015) and followed by autoxidation (Scheme 1a). We would like to note that
$NO_3^-$-CIMS only observed HOM with oxygen numbers $\geq$ 6 in this study due to its selectivity of detection.
$C_5H_8NO_n\bullet$ with even number oxygen atoms (n=8, 10, series M1b in Table 1) were possibly formed after H-shift
of an alkoxy radical formed in reaction R4 or R5 and subsequent $O_2$ addition ("alkoxy-peroxy" channel)
(Scheme 1b), where the alkoxy radicals can be formed both from the $RO_2+NO_3$ and $RO_2+RO_2$ reactions. The
hydroxy$RO_2$ formed can undergo further autoxidation adding two oxygen atoms after each H-shift. We would
like to note that the scheme and other schemes in this study only show example isomers and pathways to form these

molecules. It is likely that many of the reactions occurring are not the dominant channels as otherwise there would be much higher HOM yield as discussed below.

Some HOM monomers may contain multiple isomers and be formed via different pathways. For example, $C_5H_9NO_n$ can contain alcohols derived from $RO_2$ $C_5H_8NO_{n+1}\bullet$, hydroperoxides derived from $RO_2$ $C_5H_8NO_n\bullet$ or the ketones from $RO_2$ $C_5H_{10}NO_{n+1}\bullet$. Some $RO_2$ $C_5H_8NO_n\bullet$ may be formed via the reaction of first-generation products with $NO_3$ in addition to direct reaction of isoprene with $NO_3$. For example, $C_5H_8NO_7\bullet$ can be formed by the reaction of $NO_3$ with $C_5H_8O_2$, which is a first-generation product observed previously in the reaction of isoprene with $NO_3$ or OH (Scheme S1b) (Kwan et al., 2012). Moreover, $RO_2$ $C_5H_8NO_n\bullet$ can be formed from C5-carbonylnitrate, a first-generation product, with OH (Scheme S1a). Trace amount of OH can be produced in the reaction of isoprene with $NO_3$ (Kwan et al., 2012; Wennberg et al., 2018). OH can also be formed via Criegee intermediates formed in the isoprene+$O_3$ reaction (Nguyen et al., 2016), but this OH source was likely minor because the contribution of the isoprene+$O_3$ reaction to total isoprene loss was negligible (<5%, Fig. S2). In addition, $C_5H_8NO_8\bullet$ may also be formed by the reaction of $NO_3$ with $C_5H_8O_3$, which is a first-generation product observed in the reaction of isoprene with OH (Kwan et al., 2012). The $C_5H_8NO_n\bullet$ formed via direct reaction of isoprene with $NO_3$ is a first-generation $RO_2$ while that formed via other indirect pathways is a second-generation $RO_2$. The time profile of the isomers from these two pathways, however, are expected to be different as will be discussed below.

Time series of HOM can shed light on their formation mechanisms. It is expected that first-generation products increase fast with isoprene addition and reach a maximum earlier in the presence of wall loss of organic vapour, while second-generation products reach a maximum in the later stage or increase continuously if the production rate is higher than the loss rate. As a reference to analyze the time profiles of HOM, the times profile of isoprene, $NO_3$, and $N_2O_5$ are also shown (Fig. S4). After isoprene was added in each period, $NO_3$ and $N_2O_5$ dropped dramatically and then gradually increased. We found that termination products within the same M1 series showed different time profiles. For example, in $C_5H_9NO_n$ series, $C_5H_9NO_8$ clearly increased instantaneously with isoprene addition, and decreased fast afterwards (Fig. 3a), indicating that it was a first-generation product, which was expected according to the mechanism Scheme 1. $C_5H_9NO_6$ and $C_5H_9NO_{10}$ had a general increasing trend with time. While $C_5H_9NO_6$ increased continuously with time, $C_5H_9NO_{10}$ reached maximum intensity in the late phase of each isoprene addition period and then decreased naturally or after isoprene addition. The faster loss of $C_5H_9NO_{10}$ than $C_5H_9NO_6$ may result from the faster wall loss due to its lower volatility. $C_5H_9NO_7$ and $C_5H_9NO_9$ showed a mixing time profile with features of the former two kinds of time profiles, increasing almost instantaneously with isoprene additions, especially in the first two periods, while increasing continuously or decreasing first with isoprene additions and then increasing later in each period. This kind of time series indicates that there were significant contributions from both first- and second-generation products.

The second-generation products may be different isomers formed in pathways other than shown in Scheme 1. Second-generation $C_5H_9NO_6$ can be formed via $C_5H_8NO_7\bullet$, which can also be formed by the reaction of $NO_3$ and $O_2$ with $C_5H_8O_2$ as mentioned above (Scheme S2b), or by the reaction of OH with $C_5H_7NO_4$ (Scheme

S2a). The time profiles of $C_5H_8NO_7\bullet$ did show more contribution of second-generation processes because it
continuously increased with time in general. If the pathways via the reaction of $NO_3$ and $O_2$ with $C_5H_8O_2$ and
the reaction of OH with $C_5H_7NO_4$ contribute most to $C_5H_9NO_6$, $C_5H_9NO_6$ would show mostly a time profile of
second-generation products. Similarly, second-generation $C_5H_9NO_7$ can be formed via $C_5H_8NO_7\bullet$ or $C_5H_8NO_8\bullet$.
The time series of $C_5H_8NO_8\bullet$ did show the contribution of both the first- and second-generation processes, which
generally increased with time while also responding to isoprene addition (Fig. S5). Similar to $C_5H_9NO_6$, the
second-generation pathway for $C_5H_9NO_7$, $C_5H_9NO_9$, and $C_5H_9NO_{10}$ are shown in Scheme S1, S3, S4. For the
$RO_2$ in $C_5H_8NO_n\bullet$ series other than $C_5H_8NO_{7/8}\bullet$, the peak of $C_5H_8NO_n\bullet$ overlaps with $C_5H_{10}N_2O_n$ in the mass
spectra, which is a much larger peak, and thus cannot be differentiated from $C_5H_{10}N_2O_n$. Therefore, it is not
possible to obtain reliable separate time profiles in order to differentiate their major sources. It is worth noting
that nitrate CIMS may not be able to detect all isomers of $C_5H_9NO_6$ due to the sensitivity limitation. Therefore,
we cannot exclude the possibility that the absence of some first-generation isomers of $C_5H_9NO_6$ was due to the
low sensitivity of these isomers.

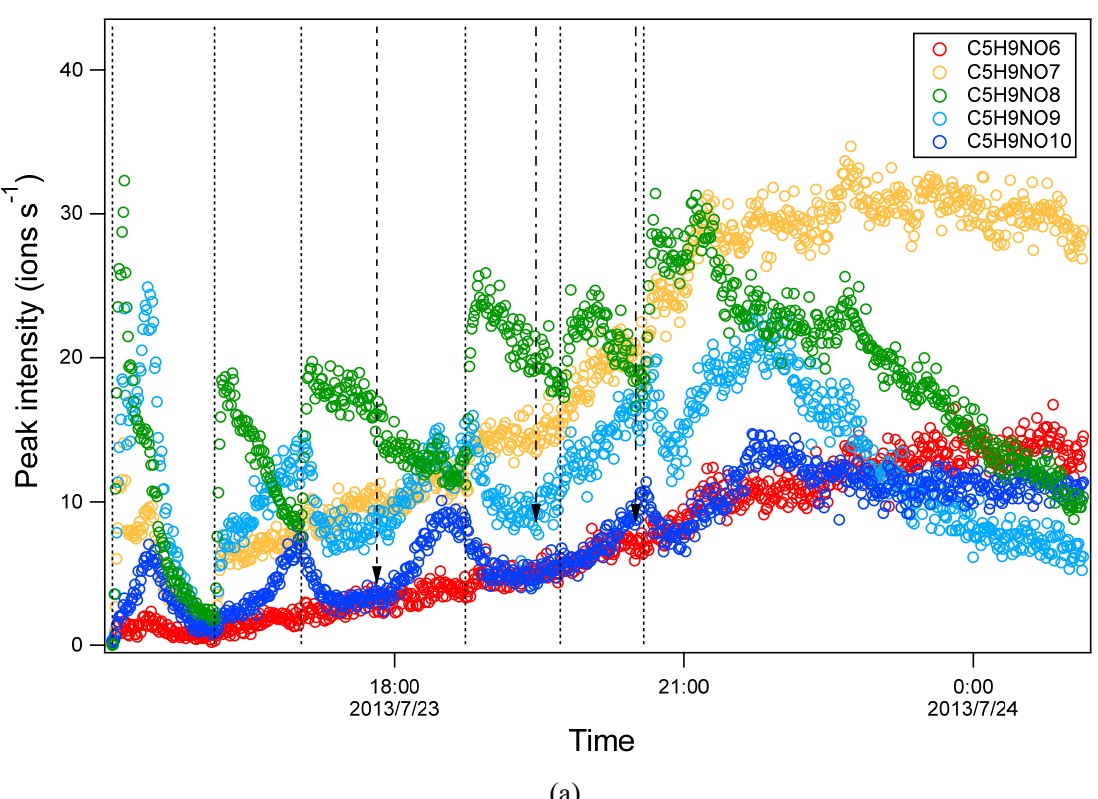


(a)

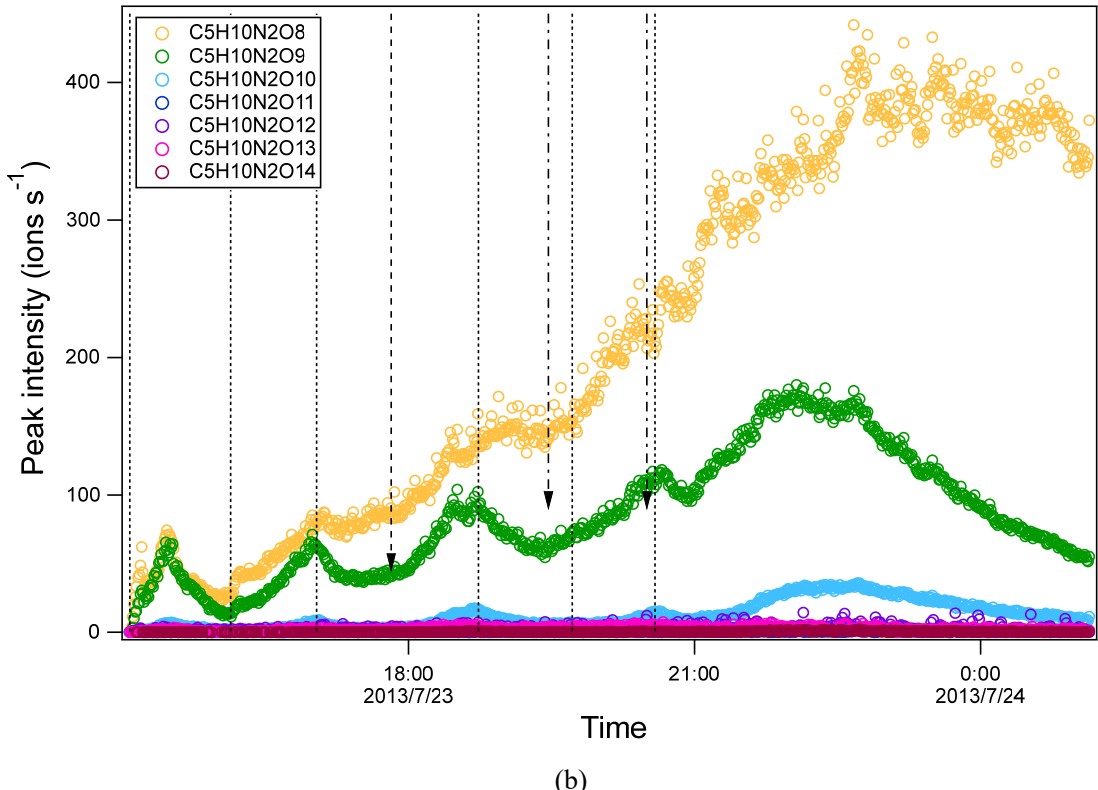

(b)

Figure 3. Time series of peak intensity of several HOM monomers of $C_5H_9NO_n$ series (a) and of $C_5H_{10}N_2O_n$ series (b). They are likely the termination products of $RO_2$ $C_5H_8NO_n\bullet$ and $C_5H_9N_2O_n\bullet$, respectively. The dashed lines indicate the time of isoprene additions. The long-dashed arrow indicates the time of $NO_2$ addition. The dash-dotted arrows indicate the time of $O_3$ additions.

Among the termination products of the 1N-monomer $RO_2$, carbonyl and hydroxyl/hydroperoxide species had comparable abundance in general (Table S1), suggesting that disproportionation reactions between $RO_2$ and $RO_2$ forming hydroxy and carbonyl species (R1-2) was likely an important $RO_2$ termination pathway. However, dependence of the exact ratio of carbonyl species to hydroxyl/hydroperoxide species on the number of oxygen atoms did not show a clear trend (Table S1), suggesting that the reactions of HOM $RO_2$ depended on their specific structure. There was no clear difference in the abundance between the termination products from $C_5H_8NO_n\bullet$ with odd and even number of oxygen atom in general, although the most abundant termination product of $C_5H_8NO_n\bullet$ , i.e. $C_5H_7NO_8$, was likely formed from $C_5H_8NO_9\bullet$ in series M1a. This fact indicates that both the peroxy pathway and alkoxy-peroxy pathway were important for the HOM formation in the isoprene+$NO_3$ reaction under our conditions, in agreement with the significant formation of alkoxy radicals from the reaction of $RO_2$ with $NO_3$ and $RO_2$.

In addition to the termination products of $RO_2$ M1, minor peaks of the $RO_2$ series $C_5H_{10}NO_n\bullet$ (n=8-9) (M4, Table 1) and their corresponding termination products including hydroperoxide, alcohol and carbonyl species were detected (Table S3). $C_5H_{10}NO_n$ were likely formed by sequential addition of $NO_3$ and OH to two double bonds of isoprene (Scheme S5). OH can react fast with isoprene or with the first-generation products of the reaction of isoprene

with NO$_3$, thus forming C$_5$H$_{10}$NO$_n\bullet$. In addition, a few very minor but noticeable peaks of C$_5$H$_9$O$_n\bullet$ and their corresponding termination products C$_5$H$_{10}$O$_n$ and C$_5$H$_8$O$_n$ were also observed. These HOM may be formed by the reactions of isoprene with trace amount of OH and with O$_3$, although their contributions to reacted isoprene were negligible. These HOM were also observed in the reaction of isoprene with O$_3$ with and without OH scavengers (Jokinen et al., 2015).

Among 1N-monomer HOM, C$_5$H$_9$NO$_7$ has been observed in the particle phase using ESI-TOFMS by Ng et al. (2008) while others have not been observed in previous laboratory studies of the reaction of isoprene with NO$_3$, to our knowledge. A number of C$_5$ organic nitrates have been observed in field studies. For example, C$_5$H$_{7-11}$NO$_{6-8}$ and C$_5$H$_{7-11}$NO$_{4-9}$ have been observed in the gas phase (Massoli et al., 2018) and the particle phase (Lee et al., 2016; Chen et al., 2020), respectively in a rural area of southeast US, where isoprene is abundant. Xu et al. (2021) observed a number of C$_5$ 1N-HOM such as C$_5$H$_{7,9,11}$NO$_{6,7}$ in polluted megacities of Nanjing and Shanghai of east China during summer. While many of these HOM have daytime sources and are attributed to photo-oxidation in the presence of NO$_x$., nighttime oxidation with NO$_3$ also contribute to their formation (Lee et al., 2016; Chen et al., 2020; Xu et al., 2021). C$_5$H$_{7-11}$NO$_{4-9}$ were also observed in chamber experiments of the reaction of isoprene with OH in the presence of NO$_x$ (Lee et al., 2016). C$_5$H$_x$NO$_{4-9}$ and C$_5$H$_x$NO$_{4-10}$ have been also observed in the gas phase and particle phase, respectively, in a monoterpene-dominating rural area in southwest Germany (Huang et al., 2019).

### 3.2.3    2N-mononmers

The 2N-monomer RO$_2$ series C$_5$H$_9$N$_2$O$_n\bullet$(n=8-14), were observed, as well as its likely termination products, C$_5$H$_8$N$_2$O$_n$ and C$_5$H$_{10}$N$_2$O$_n$, which contain a carbonyl and hydroxyl or hydroperoxide functional group, respectively. The RO$_2$ series C$_5$H$_9$N$_2$O$_n\bullet$ with odd number of oxygen atoms (n=9, 11) (M2a in Table 1) were likely formed from the first-generation product C$_5$H$_9$NO$_4$ (C5-hydroxynitrate) by adding NO$_3$ to the remaining double bond, forming C$_5$H$_9$N$_2$O$_9\bullet$, followed by autoxidation (Scheme 2a). This RO$_2$ series can also be formed by the addition of NO$_3$ to the double bond of first-generation products (e.g. C$_5$H$_9$NO$_5$, C5-nitrooxyhydroperoxide) and a subsequent alkoxy-peroxy step (Scheme 2b). C$_5$H$_9$N$_2$O$_n\bullet$ with even number of oxygen atoms (n=8, 10, 12) (M2b in Table 1), can be formed by the addition of NO$_3$ to the double bond of C$_5$H$_9$NO$_5$ followed by autoxidation (Scheme. 3a), or of C$_5$H$_9$NO$_4$ followed by an alkoxy-peroxy step (Scheme. 3b). The formation pathways of C$_5$H$_9$N$_2$O$_{13/14}\bullet$ and C$_5$H$_9$N$_2$O$_8\bullet$ cannot be well explained, as they contain too many or too few oxygen atoms to be formed via the pathways in Scheme 2 or 3. In Scheme 2 and 3, we show the reactions starting from 1-NO$_3$-isoprene-4-OO as an example. In the supplement, we have also shown the pathways starting from 1-NO$_3$-isoprene-2-OO peroxy radicals, which is indicated in a recent study by Vereecken et al. (2021) to be the dominant RO$_2$ in the reaction of isoprene with NO$_3$.

Formation through either Scheme 2 or 3 means that C$_5$H$_8$N$_2$O$_n$ and C$_5$H$_{10}$N$_2$O$_n$ were second-generation products. The time series of C$_5$H$_{10}$N$_2$O$_n$ species clearly indicates that they were indeed second-generation products. C$_5$H$_{10}$N$_2$O$_n$ species generally did not increase immediately with isoprene addition (Fig. 3b), but increased gradually with time and reached its maximum in the later stage of each period before decreasing with

time (in the period 1 and 6), or decreasing after the next isoprene addition (periods 2-5). This time profile can
be explained by the time series of the precursor of $C_5H_{10}N_2O_n$, $C_5H_9N_2O_n\bullet$ (RO$_2$) (Fig. S6). The changing rate
(production rate minus destruction rate) of $C_5H_{10}N_2O_n$ concentration was dictated by the concentration of
$C_5H_9N_2O_n\bullet$ and the wall loss rate. During periods 2 to 5, $C_5H_9N_2O_n\bullet$ gradually increased but decreased sharply
after the isoprene additions, resulted from chemical reactions of $C_5H_9N_2O_n\bullet$ and additionally from wall loss.
When the rate of change of the $C_5H_{10}N_2O_n$ concentration was positive, the concentration of $C_5H_{10}N_2O_n$ increased
with time. After isoprene additions, the rate of change of the $C_5H_{10}N_2O_n$ concentration decreased dramatically
to even negative, leading to decreasing concentrations. Similar to $C_5H_{10}N_2O_n$, the $C_5H_8N_2O_n$ series did not
respond immediately to isoprene additions (Fig. S7), which is expected for second-generation products
according to the mechanism discussed above (Scheme 2-3). Particularly, the continuing increase of $C_5H_8N_2O_n$
even after isoprene was completely depleted (at ~21:40, Fig. S7) clearly indicates that these compounds were
second-generation products, although in the end they decreased due to wall loss.

(a)

(b)
Scheme 2. The example pathways to form $C_5H_9N_2O_n$ (n=9, 11) HOM RO$_2$ series by RO$_2$ channel (a)
and alkoxy-peroxy channel. The detected products are in bold.

(a)

$C_5H_8NO_5\bullet$     $\xrightarrow{RO_2}$     $C_5H_9NO_4$     $\xrightarrow[O_2]{NO_3}$     $\mathbf{C_5H_9N_2O_9\bullet}$     $\xrightarrow{RO_2/NO_3}$     $C_5H_9N_2O_8\bullet$     $\xrightarrow[O_2]{H\text{-}shift}$     $\mathbf{C_5H_9N_2O_{10}\bullet}$

(b)

Scheme 3. The example pathways to form $C_5H_9N_2O_n$ (n=10, 12) HOM $RO_2$ series by $RO_2$ channel (a) and alkoxy-peroxy channel (b). The detected products are in bold.

According to the finding of Ng et al. (2008), C5-hydroxynitrate decays much faster than C5-nitrooxyhydroperoxides. Additionally, C5-hydroxynitrate concentration is expected to be higher than that of nitrooxyhydroperoxides because $RO_2+RO_2$ forming alcohol is likely more important than $RO_2+HO_2$ forming hydroperoxide in this study. Therefore, it is likely that $C_5H_9N_2O_n\bullet$ M2a series was mainly formed from $C_5H_9NO_4$ instead of $C_5H_9NO_5$, while $C_5H_9N_2O_n\bullet$ M2b were formed from $C_5H_9NO_4$ followed by an alkoxy-peroxy step. That is, Scheme 2a and 3b appear more likely.

Similar to $C_5H_8NO_n\bullet$, the intensity of carbonyl species from $C_5H_9N_2O_n\bullet$ was also comparable with that of hydroxyl/hydroperoxide species, suggesting that $RO_2+RO_2$ reaction forming ketone and alcohol was likely an important pathway of HOM formation in the isoprene+$NO_3$ reaction. In general, the intensity of the termination products from $C_5H_9N_2O_n\bullet$ with both even and odd oxygen numbers were comparable. This again suggests that both peroxy and alkoxy-peroxy pathways were important for HOM formation in the isoprene+$NO_3$ reaction. The intensity of $C_5H_8N_2O_n$ first increased and then decreased with oxygen number while $C_5H_{10}N_2O_n$ decreased with oxygen number, with $C_5H_{10}N_2O_8$ and $C_5H_8N_2O_8$ being the most abundant within their respective series.

Some 2N-monomers have been detected in previous studies of the reaction of isoprene with $NO_3$. $C_5H_{10}N_2O_8$ has been detected in the particle phase by Ng et al. (2008) and $C_5H_8N_2O_7$ was detected in the gas phase by Kwan et al. (2012). $C_5H_9N_2O_9\bullet$ has been proposed to be formed via the pathway as in Scheme 2a (Ng et al., 2008), and it was directly detected in our study. $C_5H_8N_2O_7$ species has been proposed to be a dinitrooxy epoxide formed by the oxidation of nitrooxyhydroperoxide (Kwan et al., 2012), instead of being a dinitrooxy ketone proposed in our study, a termination product of $C_5H_9N_2O_8\bullet$. Admittedly, $C_5H_8N_2O_7$ may contain both isomers. In addition, Ng et al. (2008) detected $C_5H_8N_2O_6$ in the gas phase, which was not detected in this study likely due to the selectivity of $NO_3^-$-CIMS. 2N-monomers have also been observed in previous field studies. For example, Massoli et al. (2018) observed $C_5H_{10}N_2O_{8-10}$ in rural Alabama US during the SOAS campaign. Xu et al. (2021) observed $C_5H_{8,10}N_2O_8$ and $C_5H_{10}N_2O_8$ in polluted megacities of Nanjing and Shanghai during summer.

One could suppose that $C_5H_7N_2O_n\bullet$ should also be formed since C5-nitrooxycarbonyl ($C_5H_7NO_4$) also contains one double bond that can be attacked by $NO_3$ in a second oxidation step. However, concentrations of $C_5H_7N_2O_n$ were too low to assign molecular formulas with confidence except for $C_5H_7N_2O_9\bullet$, clearly showing that $C_5H_7N_2O_n\bullet$ was not important. This fact is consistent with the finding of Ng et al. (2008) that C5-

nitrooxycarbonyls react slowly with $NO_3$. Additionally, the peroxy radical formed in the reaction of C5-
nitrooxycarbonyls with $NO_3$ likely leads to more fragmentation in H-shift as found in the OH oxidation of
methacrolein (Crounse et al., 2012), which may also contribute to the low abundance of $C_5H_7N_2O_n$. The presence of
HOM containing two N atoms is in line with the finding by Faxon et al. (2018) who detected products containing
two N atoms in the reaction of $NO_3$ with limonene, which also contain two carbon double bonds. It is anticipated
that for VOC with more than one double bond, $NO_3$ can add to all the double bonds as for isoprene and limonene.

### 3.2.4 3N-monomers

HOM containing three nitrogen atoms, $C_5H_9N_3O_n$ (n=9-16), were observed. These compounds were
possibly peroxynitrates formed by the reaction of $RO_2$ ($C_5H_9N_2O_n\bullet$) with $NO_2$. The time series of $C_5H_9N_3O_n$
was examined to check whether they match such a mechanism. If $C_5H_9N_3O_n$ were formed by the reaction of
$C_5H_9N_2O_{n-2}\bullet$ with $NO_2$, the concentration would be a function of the concentrations of $C_5H_9N_2O_{n-2}\bullet$ and $NO_2$ as
follows:
$$\frac{d[C_5H_9N_3O_n]}{dt} = k[C_5H_9N_2O_{n-2}\bullet][NO_2] - k_{wall}[C_5H_9N_3O_n]$$

where $[C_5H_9N_3O_n]$, $[C_5H_9N_2O_{n-2}\bullet]$, and $[NO_2]$ are the concentration of these species, k is the rate
constant and $k_{wall}$ is the wall loss rate. Because the products of $C_5H_9N_2O_{n-2}\bullet$ and $NO_2$ were at their maximum at
the end of each period and decreased rapidly after isoprene addition (Fig. S8), the concentration should have its
maximum increasing rate at the end of each isoprene addition period. However, we found that only $C_5H_9N_3O_{12,}$
$_{15, 16}$ showed such a time profile (Fig. S9), while $C_5H_9N_3O_{9, 10, 11, 13, 14}$ generally increased with time, different
from what one would expect based on the proposed pathway. Therefore, it is likely that $C_5H_9N_3O_{12, 15, 16}$ were
mainly formed via the reaction of $C_5H_9N_2O_n\bullet$ with $NO_2$, whereas $C_5H_9N_3O_{9,10,11,13,14}$ were not. Moreover,
$C_5H_9N_3O_9$ cannot be explained by the reaction $C_5H_9N_2O_n\bullet$ (n≥9) with $NO_2$ or $NO_3$, because these reactions
would add at least one more oxygen atom. One possible pathway to form $C_5H_9N_3O_9$ was the direct addition of
$N_2O_5$ to the carbon double bond of C5-hydroxynitrate, forming a nitronitrate. Such a mechanism has been
proposed previously in the heterogeneous reaction of $N_2O_5$ with 1-palmitoyl-2-oleoyl-sn-glycero-3-
phosphocholine (POPC) because $-NO_2$ and $-NO_3$ groups were detected (Lai and Finlayson-Pitts, 1991). This
pathway generally matched the time series of $C_5H_9N_3O_{9,10,11,13,14}$ typical of second-generation products since
C5-hydroxynitrate was a first-generation product. It is possible that the main pathway of $C_5H_9N_3O_{9,10,11,13,14}$ was
the reaction of $C_5H_9NO_{4,5,6}$ with $N_2O_5$, although the reaction of $N_2O_5$ with C=C double bonds in common alkenes
and unsaturated alcohols are believed to be not important (Japar and Niki, 1975; Pfrang et al., 2006).
3N-monomer, $C_5H_9N_3O_{10}$, has been observed in the particles formed in the isoprene+$NO_3$ reaction by
Ng et al. (2008). Here a complete series of $C_5H_9N_3O_n$ were observed. $C_5H_9N_3O_{10}$ was previously proposed to
be formed by another pathway, i.e. the reaction of $RO_2$ ($C_5H_9N_2O_9\bullet$) and $NO_3$ (Ng et al., 2008). We further
examined the possibility of such a pathway in our study. Similar to $NO_2$, if $C_5H_9N_3O_n$ were formed by the
reaction of $C_5H_9N_2O_{n-2}\bullet$ with $NO_3$, the concentration would have its maximum increasing rate at the end of each
isoprene addition period. Among $C_5H_9N_2O_n\bullet$, the precursors of $C_5H_9N_3O_n$, $C_5H_9N_2O_{9, 10, 13, 14}\bullet$ showed a

maximum increasing rate and a subsequent decrease after isoprene addition. The difference in oxygen number between $C_5H_9N_3O_{12, 15, 16}$, the termination products, and $C_5H_9N_2O_{9, 10, 13, 14}\bullet$, the corresponding $RO_2$ with the consistent time profile is mostly two. Since the reaction of $C_5H_9N_2O_n$ with $NO_2$ and $NO_3$ result an increased oxygen number by two and by one, respectively, we infer that it is more likely that $C_5H_9N_3O_{12, 15, 16}$ were formed by the reaction of $C_5H_9N_2O_{10, 13, 14}\bullet$ with $NO_2$ rather than $NO_3$, and thus they were likely peroxynitrates rather than nitrates formed by the reaction of $RO_2$ with $NO_3$. Since alkyl peroxynitrates decompose rapidly (Finlayson-Pitts and Pitts, 2000; Ziemann and Atkinson, 2012), it is possible that these compounds contained peroxyacylnitrates.

Little attention has been paid to the $RO_2+NO_2$ pathway in nighttime chemistry of isoprene in the literature (Wennberg et al., 2018), which is likely due to the instability of the products. According to this pathway, $C_5H_8N_2O_n$, which was proposed to be a ketone formed via $C_5H_9N_2O_9\bullet$ in the M2 series (Table 1) as discussed above, can also comprise peroxynitrate formed by the reaction of $C_5H_8NO_n\bullet$ (M1a $RO_2$) with $NO_2$. 3N dimer such as $C_5H_9N_3O_{10}$ or have been observed in a recent field study in polluted cities in east China (Xu et al., 2021).

### 3.3 HOM dimers and their formation

Table 2. HOM dimers and trimers formed in the oxidation of isoprene by $NO_3$.

| Series Number | Formula | Type | Pathway of $RO_2$ |
|---|---|---|---|
| Dimer 1 | $C_{10}H_{16}N_2O_{n\,(n=10-17)}$ | ROOR[a] | M1[b] +M1 |
| Dimer 2 | $C_{10}H_{17}N_3O_{n\,(n=11-19)}$ | ROOR | M1+M2/M3+M4 |
| Dimer 3 | $C_{10}H_{18}N_4O_{n\,(n=15-18)}$ | ROOR | M2+M2 |
| Dimer 4 | $C_{10}H_{18}N_2O_{n\,(n=10-16)}$ | ROOR | M1+M4 |
| Dimer 5 | $C_{10}H_{15}N_3O_{n\,(n=13-17)}$ | ROOR | M1+M3 |
| Dimer 6 | $C_{10}H_{19}N_3O_{n\,(n=14-15)}$ | ROOR | M2+M4 |
| Dimer 7 | $C_{10}H_{14}N_2O_{n\,(n=10-16)}$ | ROOR | Unknown |
| Dimer 8 | $C_{10}H_{15}NO_{n\,(n=9-12)}$ | ROOR | $C_{10}H_{16}NO_n$ |
| Dimer 9 | $C_{10}H_{17}NO_{n\,(n=9-15)}$ | ROOR | $C_{10}H_{16}NO_n$ |
| Dimer R1 | $C_{10}H_{16}N_3O_{n\,(n=12-15)}$ | $RO_2$ | Dimer 1+$NO_3$ |
| Dimer R2 | $C_{10}H_{17}N_2O_{n\,(n=11-12)}$ | $RO_2$ | Dimer 1+OH |
| Dimer R3 | $C_{10}H_{17}N_4O_{n\,(n=16-18)}$ | $RO_2$ | Dimer 2+$NO_3$ |
| Dimer R4 | $C_{10}H_{16}NO_{n\,(n=10-14)}$ | $RO_2$ | M1+$C_5H_8$ |
| Trimer 1 | $C_{15}H_{24}N_4O_{n\,(n=17-22)}$ | ROOR | Dimer R1+M1 |
| Trimer 2 | $C_{15}H_{25}N_5O_{n\,(n=20-22)}$ | ROOR | Dimer R3+M1; Dimer R1+M2 |
| Trimer 3 | $C_{15}H_{25}N_3O_{n\,(n=13-20)}$ | ROOR | Dimer R2+M1; Dimer R4+M2 |
| Trimer 4 | $C_{15}H_{26}N_4O_{n\,(n=17-21)}$ | ROOR | Dimer R2+M2 |

[a]: ROOR denotes for organic peroxide.

[b]: The numbering is referred to Table 1.

A number of HOM dimer series were observed, including $C_{10}H_{16}N_2O_n$ (n=10-17), $C_{10}H_{17}N_3O_n$ (n=11-19), and $C_{10}H_{18}N_4O_n$ (n=15-18), $C_{10}H_{18}N_2O_n$ (n=10-16), $C_{10}H_{15}N_3O_n$ (n=13-17), and $C_{10}H_{19}N_3O_n$ (n=14-15) series (Table 2,

Table S3). $C_{10}H_{16}N_2O_n$ series (dimer 1, Table 2) was likely formed by the accretion reaction of two monomer $RO_2$ of M1a/b (Reaction R7).

$$C_5H_8NO_{n1}\bullet + C_5H_8NO_{n2}\bullet \rightarrow C_{10}H_{16}N_2O_{n1+n2-2}+O_2 \qquad\qquad R7$$

Similarly, $C_{10}H_{18}N_4O_n$ series (dimer 2, Table 2) were likely formed by the accretion reaction of two monomer $RO_2$ of M2 (Reaction R8). As n1 and n2 are $\geq$ 9, the number of oxygen in $C_{10}H_{18}N_4O_n$ is expected to be $\geq$16. This is consistent with our observation that only $C_{10}H_{18}N_4O_n$ with n$\geq$16 had significant concentrations.

$$C_5H_9N_2O_{n1}\bullet + C_5H_9N_2O_{n2}\bullet \rightarrow C_{10}H_{18}N_4O_{n1+n2-2}+O_2 \qquad\qquad R8$$

$C_{10}H_{17}N_3O_n$ series (dimer 3, Table 2) were likely formed by the cross accretion reaction of one M1 $RO_2$ and one M2 $RO_2$ (reaction R9). Since n1 is $\geq$5 and n2 is $\geq$9, the number of oxygen atoms in $C_{10}H_{17}N_3O_n$ is expected to be $\geq$12, which is also roughly consistent with our observation that only $C_{10}H_{17}N_3O_n$ with n$\geq$11 were detected.

$$C_5H_8NO_{n1}\bullet + C_5H_9N_2O_{n2}\bullet \rightarrow C_{10}H_{17}N_3O_{n1+n2-2}+O_2 \qquad\qquad R9$$

Similarly, $C_{10}H_{18}N_2O_{n\ (n=10-16)}$ and $C_{10}H_{15}N_3O_{n\ (n=13-17)}$ series (dimer 4, dimer 5, Table 2) were likely formed from the accretion reaction between one M1 $RO_2$ and one M4 $RO_2$, and between one M1 $RO_2$ and one M3 $RO_2$ ($C_5H_7N_2O_9\bullet$). Other dimer series than dimer 1-5 were also present. However, they had quite low intensity (Fig. 4), which was consistent with the low abundance of their parent monomer $RO_2$. They can be formed from various accretion reactions of monomer $RO_2$. For example, $C_{10}H_{19}N_3O_n$ can be formed by the accretion reaction of $C_5H_9N_2O_n\bullet$ and $C_5H_{10}NO_n\bullet$ (Table 2).

Similar to monomers, a few species dominated in HOM dimers spectrum. The dominant dimer series were $C_{10}H_{17}N_3O_x$ and $C_{10}H_{16}N_2O_x$ series, with $C_{10}H_{17}N_3O_{12-14}$ and $C_{10}H_{16}N_2O_{12-14}$ showing highest intensity among each series (Fig. 4). In addition, the O/C ratio or oxidation state of HOM dimers were generally lower than that of monomers (Fig. 2, Fig. 4), which resulted from the loss of two oxygen atoms in the accretion reaction of two monomer $RO_2$.

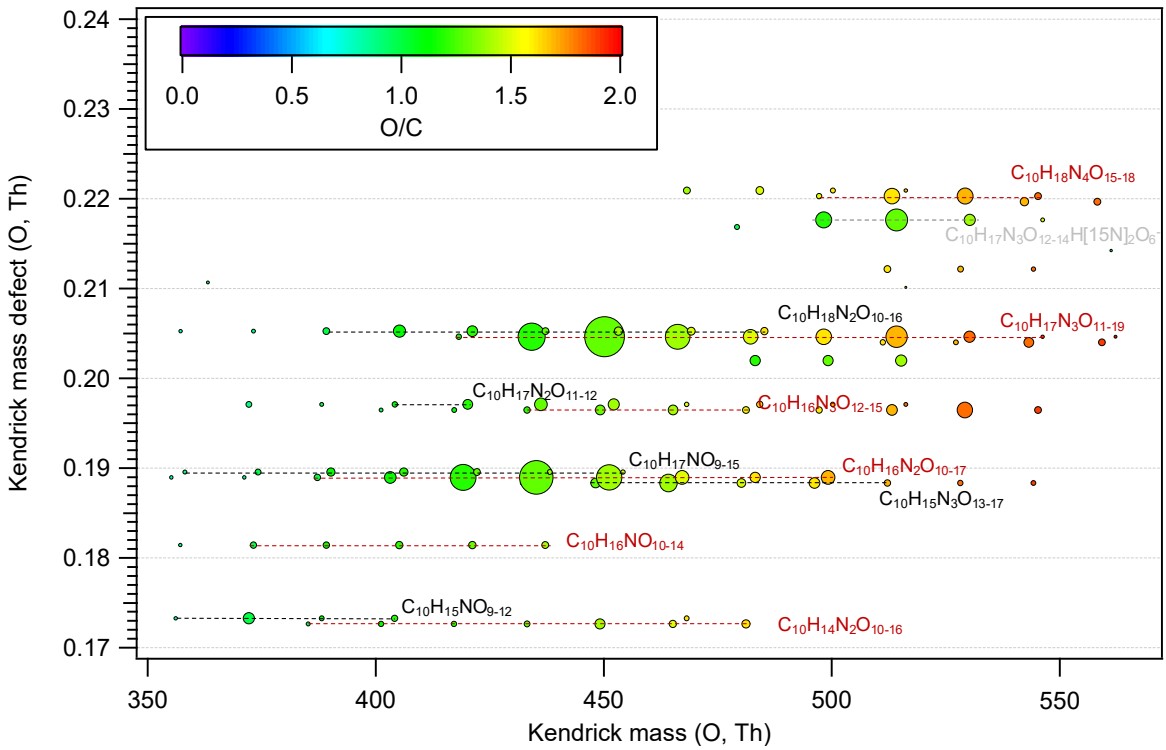

Figure 4. Kendrick mass defect plot for O of HOM dimers formed in the isoprene+NO$_3$ reaction. The size (area) of circles is set to be proportional to the average peak intensity of each molecular formula during the first isoprene addition period (P1). The molecular formula include the reagent ion $^{15}$NO$_3^-$, which is not shown for simplicity. The species labelled in grey (C$_{10}$H$_{17}$N$_3$O$_{12-14}$ H[15N]$_2$O$_6^-$) are the adducts of C$_{10}$H$_{17}$N$_3$O$_{12-14}$ with H[15N]$_2$O$_6^-$.

According to the mechanism above (R7-9), we attempt to explain the relative intensities of the dimers using the signal intensities of monomer RO$_2$. Assuming that the rate constant for each of HOM-RO$_2$+ HOM-RO$_2$ reaction forming dimers is the same considering that all HOM-RO$_2$ are highly oxygenated with a number of functional groups, it is expected that the dimer formed by the recombination between the most abundant RO$_2$ has the highest intensity. The most abundant monomer RO$_2$ were C$_5$H$_9$N$_2$O$_9\bullet$ and C$_5$H$_9$N$_2$O$_{10}\bullet$ and thus the most abundant dimers are expected to be C$_{10}$H$_{16}$N$_4$O$_{16}$, C$_{10}$H$_{16}$N$_4$O$_{17}$, and C$_{10}$H$_{16}$N$_4$O$_{18}$. This expected result is in contrast with our observation showing that the most abundant dimers were C$_{10}$H$_{17}$N$_3$O$_{12-14}$ and C$_{10}$H$_{16}$N$_2$O$_{12-14}$ (Fig. 4). The discrepancy is possibly attributed to the presence of less oxygenated RO$_2$ (with O$\leq$5) that have a low detection sensitivity in the NO$_3$-CIMS (Riva et al., 2019) due to their lower oxygenation compared with other HOM RO$_2$ shown above. These RO$_2$ may react with C$_5$H$_9$N$_2$O$_9\bullet$ and C$_5$H$_9$N$_2$O$_{10}\bullet$. For example, C$_5$H$_8$NO$_5\bullet$ (RO$_2$) is proposed to be an important first-generation RO$_2$ in the oxidation of isoprene by NO$_3$ (Ng et al., 2008; Rollins et al., 2009; Kwan et al., 2012; Schwantes et al., 2015). Although C$_5$H$_8$NO$_5\bullet$ showed very low signal in our mass spectra, it was likely to have high abundance since it was the first RO$_2$ formed in the reaction of isoprene with NO$_3$. Indeed, we found that the termination products of C$_5$H$_8$NO$_5\bullet$ such as C$_5$H$_9$NO$_5$, C$_5$H$_7$NO$_4$, and C$_5$H$_9$NO$_4$ had high abundance in another study, indicating the high abundance of C$_5$H$_8$NO$_5\bullet$. The accretion reaction of C$_5$H$_8$NO$_5\bullet$ with C$_5$H$_9$N$_2$O$_{9-10}\bullet$ and C$_5$H$_8$NO$_{9-10}\bullet$ can explain the high abundance of C$_{10}$H$_{17}$N$_3$O$_{12-14}$ and C$_{10}$H$_{16}$N$_2$O$_{12-14}$ among all dimers.

Provided that C$_5$H$_8$NO$_5\bullet$ is abundant, we still cannot explain the relative intensity of C$_{10}$H$_{17}$N$_3$O$_{12}$, C$_{10}$H$_{17}$N$_3$O$_{13}$, and C$_{10}$H$_{17}$N$_3$O$_{14}$ that were all formed by the accretion reaction with C$_5$H$_8$NO$_5\bullet$. C$_{10}$H$_{17}$N$_3$O$_{12}$ should have the highest intensity among C$_{10}$H$_{17}$N$_3$O$_{12-14}$ as its precursor RO$_2$, C$_5$H$_9$N$_2$O$_9\bullet$, is the most abundant. This suggests that accretion reactions other than those of C$_5$H$_8$NO$_5\bullet$ with C$_5$H$_9$N$_2$O$_{9-10}\bullet$ also contributed to C$_{10}$H$_{17}$N$_3$O$_{12-14}$. Admittedly, the assumption of different RO$_2$ having similar rate constants in accretion reactions may not be valid. For example, self-reaction of tertiary RO$_2$ is slower than secondary and primary RO$_2$ (Jenkin et al., 1998; Finlayson-Pitts and Pitts, 2000). Different rate constants may also lead to the observation that the most abundant dimers could not be explained the most abundant RO$_2$.

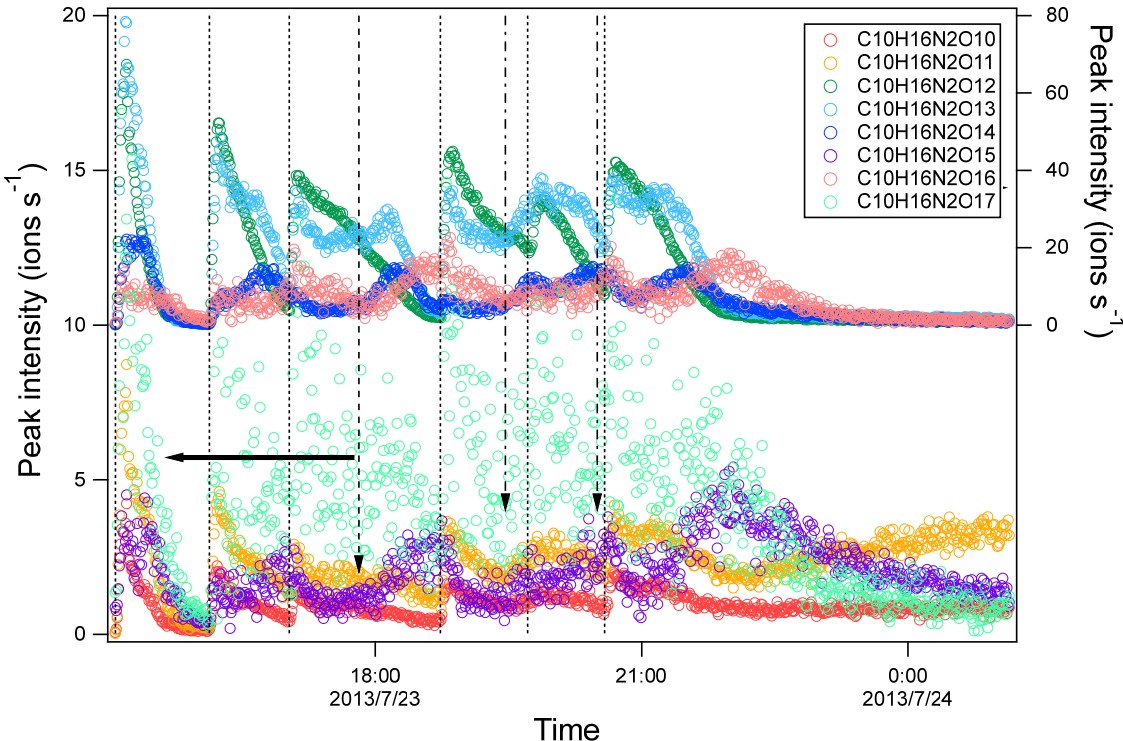

544 Figure 5. Time series of peak intensity of several HOM dimers of $C_{10}H_{16}N_2O_n$ series. The dashed lines

545 indicate the time of isoprene additions. The long-dashed arrow indicates the time of $NO_2$ addition. The dash-dotted

546 arrows indicate the time of $O_3$ additions. The horizontal arrows indicate y-axis scales for different markers.

547  The time profiles of $C_{10}H_{16}N_2O_n$ indicate contributions of both the first- and second-generation products.

548 The dominance of the first- or second-generation products depended on the specific compounds. Most $C_{10}H_{16}N_2O_n$

549 compounds increased instantaneously after isoprene additions, indicating significant contributions of first-generation

550 products. Since the formation of $C_{10}H_{16}N_2O_n$ likely involved $C_5H_8NO_5\bullet$ as discussed above, the instantaneous

551 increase may result from the increase of $C_5H_8NO_5\bullet$ as well as other first-generation $RO_2$. After the initial increase,

552 $C_{10}H_{16}N_2O_{10-12}$ then decayed with time (Fig. 5) while $C_{10}H_{16}N_2O_{13-15}$ increased again in the later phase of a period

553 and when $NO_2$ and $O_3$ were added. The second increase indicated that $C_{10}H_{16}N_2O_{13-15}$ may contain more than one

554 isomer, which had different production pathways. As discussed above, $C_5H_8NO_n\bullet$ can be either a first-generation

555 $RO_2$ formed directly via the reaction of isoprene with $NO_3$ and autoxidation, or a second-generation $RO_2$, e.g. formed

556 via the reaction of with $C_5H_8O_2$ with $NO_3$. Therefore the second increase of $C_{10}H_{16}N_2O_{13-15}$ may result from the

557 reaction of two first-generation $RO_2$ and of two second-generation $RO_2$ or between one first-generation and one

558 second-generation $RO_2$. The increase of $C_{10}H_{16}N_2O_{14-15}$ after isoprene addition was not large, indicating the

559 larger contributions from second-generation products compared with other $C_{10}H_{16}N_2O_n$. Overall, as the number

560 of oxygen increased, the contribution of second-generation products to $C_{10}H_{16}N_2O_n$ increased.

561  In contrast to $C_{10}H_{16}N_2O_n$ series, $C_{10}H_{18}N_4O_n$ increased gradually after each isoprene addition and then

562 decreased afterward (Fig. 6), either naturally or after isoprene additions, which is typical for second-generation

563 products. Since $C_{10}H_{18}N_4O_n$ was likely formed by the accretion reaction of $C_5H_9N_2O_n\bullet$ ($RO_2$), the time profile

564 of $C_{10}H_{18}N_4O_n$ was as expected since $C_5H_9N_2O_n\bullet$ was formed via the reaction of $NO_3$ with first-generation

products $C_5H_9NO_n$. The $C_{10}H_{18}N_4O_n$ concentration depended on the product of the concentrations of two
$C_5H_9N_2O_n\bullet$. Taking $C_{10}H_{18}N_4O_{16}$ as an example, its concentration can be expressed as follows:
$$\frac{d[C_{10}H_{18}N_4O_{16}]}{dt} = k[C_5H_9N_2O_9][C_5H_9N_2O_9] - k_{wl}[C_{10}H_{18}N_4O_{16}]$$

When the concentration of $C_5H_9N_2O_9\bullet$ increased, the changing rate of $C_{10}H_{18}N_4O_{16}$ was positive and increased
and thus the concentration of $C_{10}H_{18}N_4O_{16}$ increased. When the concentration $C_5H_9N_2O_9\bullet$ decreased sharply
after isoprene additions, the changing rate of $C_{10}H_{18}N_4O_{16}$ decreased and even became negative values, and thus
the concentration of $C_{10}H_{18}N_4O_{16}$ decreased after isoprene addition.
Similar to the $C_{10}H_{16}N_2O_n$ series, while $C_{10}H_{17}N_3O_n$ first increased instantaneously with isoprene
addition, it increased again during the later stage of each period (Fig. S10), showing a mixed behavior of the
first-generation products and second-generation products. The time series of $C_{10}H_{17}N_3O_n$ was as expected in
general because $C_{10}H_{17}N_3O_n$ was likely formed via the accretion reaction of $C_5H_8NO_n\bullet$ (M1 $RO_2$) and
$C_5H_9N_2O_n\bullet$ (M2 $RO_2$), which were first- or second-generation, and second-generation $RO_2$, respectively,

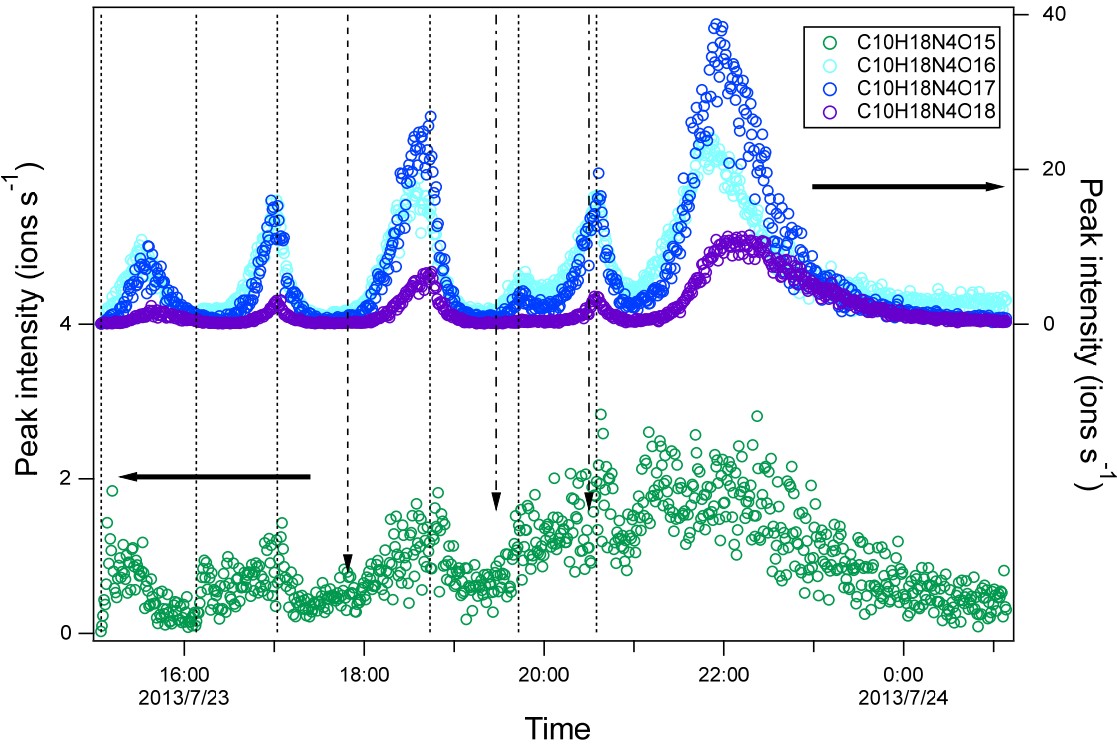


Figure 6. Time series of peak intensity of several HOM dimers of $C_{10}H_{18}N_4O_n$ series. The dashed lines indicate the
time of isoprene additions. The long-dashed arrow indicates the time of $NO_2$ addition. The dash-dotted arrows
indicate the time of $O_3$ additions. The horizontal arrows indicate y-axis scales for different markers.
Some dimers that cannot be explained by accretion reactions such as $C_{10}H_{16}N_3O_{n\ (n=12-15)}\bullet$, $C_{10}H_{17}N_2O_{n\ (n=11-12)}\bullet$,
$C_{10}H_{16}NO_{n\ (n=10-14)}\bullet$, $C_{10}H_{15}NO_{n\ (n=9-12)}$, $C_{10}H_{17}NO_{n\ (n=9-15)}$ were also observed. These dimers had low abundance.
We note that due to their low signals in the mass spectra, their assignment and thus range of n may be subject to
uncertainties. Since $C_{10}H_{16}NO_{n\ (n=10-16)}\bullet$, $C_{10}H_{16}N_3O_{n\ (n=12-15)}\bullet$, and $C_{10}H_{17}N_2O_n\bullet$ contain unpaired electrons, they
cannot be formed via the direct accretion reaction of two $RO_2$. Instead, $C_{10}H_{16}N_3O_{n\ (n=12-15)}\bullet$ (dimer R1) and
$C_{10}H_{17}N_2O_n\bullet$ (dimer R2) were likely $RO_2$ formed by the reaction of HOM dimers containing a double bond (dimer
1) with $NO_3$ and with OH, respectively, followed by the reaction with $O_2$.

$C_{10}H_{16}N_2O_n + NO_3 + O_2 \rightarrow C_{10}H_{16}N_3O_n\bullet$                         R10

$C_{10}H_{16}N_2O_n + OH + O_2 \rightarrow C_{10}H_{17}N_2O_n\bullet$                          R11

The corresponding termination products of $C_{10}H_{16}N_3O_n\bullet$ $RO_2$ series such as $C_{10}H_{15}N_3O_n$ (ketone), $C_{10}H_{17}N_3O_n$
(hydroperoxide/alcohol) were also observed, although these compounds can also be formed via reactions between
two $RO_2$ radicals (R9 and R11). Among the termination products, $C_{10}H_{15}N_3O_n$ had low intensity. Reaction R13 and
the termination reaction of $C_{10}H_{17}N_2O_n\bullet$ with $HO_2$ provided an additional pathway to $C_{10}H_{17}N_3O_n$ besides the R9
pathway discussed above. Similarly, other dimers may also be formed by the termination reactions of dimer $RO_2$
with $RO_2$ or $HO_2$. E.g., $C_{10}H_{18}N_4O_n$ can be formed via termination reaction of $C_{10}H_{17}N_4O_n\bullet$ with another $RO_2$ wherein
$C_{10}H_{17}N_4O_n\bullet$ can be formed as follows:

$C_{10}H_{17}N_3O_n + NO_3 + O_2 \rightarrow C_{10}H_{17}N_4O_n\bullet$                        R12

$C_{10}H_{16}NO_{n\ (n=10-14)}\bullet$ could be explained by the reaction of monomer $RO_2$ with isoprene.

$C_5H_8NO_n\bullet + C_5H_8 + O_2 \rightarrow C_{10}H_{16}NO_n\bullet$                        R13

Only $C_{10}H_{16}NO_n\bullet$ with $n \geq 10$ were detected, while according to the mechanism of self-reaction between $C_5H_8NO_n\bullet$,
the n range of $C_{10}H_{16}NO_n\bullet$ is expected to be 7-14. The absence of $C_{10}H_{16}NO_{n(n<10)}\bullet$ is likely attributed to their low
abundance, which might result from low precursor concentrations, low reaction rates with isoprene, and/or faster
reactive losses with other radicals. Such a reaction of $RO_2$ with isoprene has been proposed by Ng et al. (2008) and
Kwan et al. (2012). The corresponding termination products of $C_{10}H_{16}NO_n\bullet$ are $C_{10}H_{15}NO_n$ (ketone) and $C_{10}H_{17}NO_n$
species (hydroperoxide/alcohol). $C_{10}H_{17}NO_n$ species showed a time profile of typical first-generation products (Fig.
S11), i.e. increasing immediately with isoprene addition and then decaying with time. This behaviour further supports
the possibility of reaction R13. Yet, the reaction rate of alkene with $RO_2$ is likely low due to the high activation
energy (Stark, 1997, 2000). It is worth noting that to our knowledge no experimental kinetic data on the addition of
$RO_2$ to alkenes in the gas phase in atmospheric relevant conditions are available, though fast, low-barrier ring closure
reactions in unsaturated $RO_2$ radicals have been reported (Vereecken and Peeters, 2004, 2012; Kaminski et al., 2017;
Richters et al., 2017; Chen et al., 2021). We would like to note that there is unlikely interference to $C_{10}$-HOM from
monoterpenes, which has been reported previously (Bernhammer et al., 2018), as the concentration of monoterpenes
in the chamber during this study was below the limit of detection, which was ~50 ppt ($3\sigma$).

Some of the dimers discussed above have been observed in previous laboratory studies. Ng et al. (2008)

found $C_{10}H_{16}N_2O_8$ and $C_{10}H_{16}N_2O_9$ in the gas phase and $C_{10}H_{17}N_3O_{12}$, $C_{10}H_{17}N_3O_{13}$, $C_{10}H_{18}N_4O_{16}$, and $C_{10}H_{17}N_5O_{18}$
in the particle phase. $C_{10}H_{16}N_2O_8$ and $C_{10}H_{16}N_2O_9$ were also observed in our study, but their intensity in the MS was
too low to assign molecular formulas with high confidence. The low intensity may be due to the low sensitivity of
$C_{10}H_{16}N_2O_{8,\ 9}$ in $NO_3^-$-CIMS. According to modelling results of the products formed in cyclohexene ozonolysis by
Hyttinen et al. (2015), at least two hydrogen bond donor functional groups are needed for a compound to be detected
in a nitrate CIMS. As $C_{10}H_{16}N_2O_8$ and $C_{10}H_{16}N_2O_9$ have no and only one H-bond donor function groups, respectively,
they are expected to have low sensitivity in $NO_3^-$-CIMS. Moreover, the low intensity can be partly attributed to the
much lower isoprene concentrations used in this study compared to previous studies, leading to the low concentration
of $C_{10}H_{16}N_2O_8$ and $C_{10}H_{16}N_2O_9$ (Ng et al., 2008). $C_{10}H_{17}N_3O_{12}$, $C_{10}H_{17}N_3O_{13}$, $C_{10}H_{18}N_4O_{16}$, and $C_{10}H_{17}N_5O_{18}$ were
all observed in the gas phase in this study, wherein the concentration of $C_{10}H_{17}N_5O_{18}$ was very low. The formation
pathways of $C_{10}H_{17}N_3O_{12}$, $C_{10}H_{17}N_3O_{13}$, and $C_{10}H_{18}N_4O_{16}$ (R8) were generally similar to those proposed by Ng et al.
(2008) except that the products from H-shift of $RO_2$ were involved in the formation of $C_{10}H_{17}N_3O_{13}$. Among the two
pathways of $C_{10}H_{18}N_4O_{16}$ formation (R8 and via R12), our results indicate that R8 was the main pathway, based on
the low concentrations of $C_{10}H_{17}N_4O_{16/17}\bullet$ and other termination product of them, $C_{10}H_{16}N_4O_{15/16}$. That the time
profile of $C_{10}H_{18}N_4O_{16}$ was consistent with what is expected from R8 as discussed above offers additional evidence
to that conclusion.
Few field studies have reported HOM dimers formed via the reaction $NO_3$ with isoprene. This might be
because $NO_3$+isoprene-HOM dimers can have the identical molecular formula to the HOM monomers from
monoterpene oxidation. Possible contribution of dimer formation in the isoprene oxidation to C6-10 HOM in the
particle phase observed at a rural site Yorkville, US is reported by Chen et al. (2020), although these HOM are
attributed to be more likely from monoterpene oxidation.
**3.4    HOM trimers and their formation**
A series of HOM trimers were observed, such as $C_{15}H_{24}N_4O_{n\ (n=17-22)}$, $C_{15}H_{25}N_5O_{n\ (n=20-22)}$, $C_{15}H_{25}N_3O_n$
$_{(n=13-20)}$, $C_{15}H_{26}N_4O_{n\ (n=17-21)}$, and $C_{15}H_{24}N_2O_{n\ (n=12-16)}$. Among the trimers, $C_{15}H_{24}N_4O_n$was the most abundant series
(Fig. S12). The $C_{15}H_{24}N_4O_n$ series can be explained by the accretion reaction of one monomer HOM $RO_2$ and
one dimer HOM $RO_2$.

$C_{10}H_{16}N_3O_{n1}\bullet + C_5H_8NO_{n2}\bullet \rightarrow C_{15}H_{24}N_4O_{n1+n2-2}+O_2$                  R14

The formation pathways of dimer $RO_2$ $C_{10}H_{16}N_3O_n$ (n=12-15) and $C_{10}H_{17}N_2O_n$ are shown above (reaction R10 and
R11).
The other trimers were likely formed via similar pathways (Table 2 and Supplement S2). Since $NO_3^-$-CIMS
cannot provide the structural information of these HOM trimers, we cannot elucidate the major pathways. However,
in all these pathways, dimer-$RO_2$ is necessary to form a trimer, and most of the dimer-$RO_2$ formation pathways
require at least one double bond in the dimer molecule except for the reaction of $RO_2$ with isoprene. Since one
double bond has already reacted in the monomer-$RO_2$ formation, we anticipate that in the reaction with $NO_3$ it is
more favourable for precursors (VOC) containing more than one double bonds to form trimer molecules than
precursors containing only one double bond, as it is easier to generate new $RO_2$ radicals from these dimers by
attack on the remaining double bond(s).
The time profile of $C_{15}H_{24}N_4O_n$ showed the mixed behavior of first- and second-generation products (Fig.
S13), consistent with the mechanism discussed above since $C_5H_8NO_n\bullet$ and $C_{10}H_{16}N_3O_n\bullet$ were of first- or second-
generation and second-generation, respectively. The contributions of the second-generation products became
larger as the number of oxygen atoms increased. In contrast, $C_{15}H_{25}N_3O_n$ showed instantaneous increase with
isoprene addition (Fig. S14), which was typical for time profiles of first-generation products. Both proposed
formation pathways of $C_{15}H_{25}N_3O_n$ (RS6 and RS7) contained a second-generation $RO_2$, which was not in line with
the time profile observed. The observation cannot be well explained, unless we assume molecular adducts of a dimer
with one monomer. It is also possible that some $C_{10}H_{17}N_2O_n$• were formed very fast or that there were other
formation pathways of $C_{15}H_{25}N_3O_n$ not accounted for here.

We are not aware of field studies reporting $NO_3$+isoprene-HOM trimers, which is likely due to the same

reason for dimers discussed above. It is challenging to distinguish HOM trimers formed in the reaction $NO_3$ with
isoprene from the dimers formed by cross reaction of the $RO_2$ from monoterpene oxidation (C10-$RO_2$) with that from
isoprene oxidation (C5-$RO_2$) as their molecular formula can be identical.

**3.5    Contributions of monomers, dimer, and trimers to HOM**


The concentration (represented by peak intensity) of monomers was higher than that of dimers, but overall

their concentrations remained of the same order of magnitude (Fig 1a, inset). The concentration of trimers was much
lower than that of monomers and dimers. The relative contributions of monomers, dimers, and trimers evolved in
time due to the changing concentration of each HOM species. Comparing the contributions of various classes of
HOM in period 1 with those in periods 1-6 reveals that the relative contribution of monomers increased with time,
especially that of 2N-monomers, while the contribution of dimers decreased. This trend is attributed to the larger wall
loss of dimers compared to monomers because of their lower volatility and also to the continuous formation of
second-generation monomers, mostly 2N-momomers. Overall, the relative contribution of total HOM monomers
decreased immediately after isoprene addition while the contribution of HOM dimers increased rapidly (Fig. S15),
which was attributed to the faster increase of dimers intensity due to their rapid formation. Afterwards, the
contribution of monomers to total HOM gradually increased and that of dimers decreased, which was partly due to
the faster wall loss rate of dimers and to the continuous formation of second-generation monomers.

**3.6    Yield of HOM**


The HOM yield in the oxidation of isoprene by $NO_3$ was estimated using the sensitivity of $H_2SO_4$. It was

derived for the first isoprene addition period to minimize the contribution of multi-generation products and to better
compare with the data in literature, thus denoted as primary HOM yield (Pullinen et al., 2020) and was estimated to
be $1.2\%^{+1.3\%}_{-0.7\%}$. The uncertainty was estimated as shown in the Supplement S1. Despite the uncertainty, the primary
HOM yield here was much higher than the HOM yield from the ozonolysis and photooxidation of isoprene (Jokinen
et al., 2015). The difference may be attributed to the more efficient oxygenation in the addition of $NO_3$ to carbon
double bonds. Compared with the reaction with $O_3$ or OH, the initial peroxy radicals contains 5 oxygen atoms when
isoprene reacts with $NO_3$, while the initial peroxy radicals contains only 3 oxygen atoms when reacting with OH, and
the ozonide contains 3 oxygen atoms in the case of $O_3$.

**4    Conclusion and implications**


HOM formation in the reaction of isoprene with $NO_3$ was investigated in the SAPHIR chamber. A number

of HOM monomers, dimers, and trimers containing one to five nitrogen atoms were detected, and their time-
dependent concentration profiles were tracked throughout the experiment. Some formation mechanisms for various
HOM were proposed according to the molecular formula identified, and the available literature. HOM showed a
variety of time profiles with multiple isoprene additions during the reaction. First-generation HOM increased
instantaneously after isoprene addition and then decreased while second-generation HOM increased gradually and
then decreased with time, reaching a maximum concentration at the later stage of each period. The time profiles
provide additional constraints on their formation mechanism beside the molecular formula, suggesting whether they
were first-generation products or second-generation products or a combination of both. 1N-monomers (mostly $C_5$)
were likely formed by $NO_3$ addition to a double bond of isoprene, forming monomer $RO_2$, followed by autoxidation
and termination via the reaction with $HO_2$, $RO_2$, and $NO_3$. Time series suggest that some 1N-monomer could also be
formed by the reaction of first-generation products with $NO_3$, and thus be of second-generation. 2N-monomers were
likely formed via the reaction of first-generation products such as C5-hydroxynitrate with $NO_3$ and thus second-
generation products. 3N-monomers likely comprised peroxy/peroxyacyl nitrates formed by the reaction of 2N-
monomer $RO_2$ with $NO_2$, and possibly nitronitrates formed via the direct addition of $N_2O_5$ to the first-generation
products. HOM dimers were mostly formed by the accretion reactions between various HOM monomer $RO_2$, either
first-generation or second-generation or with the contributions of both, and thus showed time profiles typical of either
first-generation products, or second-generation products, or a combination of both. Additionally, some dimers peroxy
radicals (dimer $RO_2$) were formed by the reaction of $NO_3$ with dimers containing a C=C double bond. HOM trimers
were proposed to be formed by accretion reactions between the monomer $RO_2$ and dimer $RO_2$.

Overall, both HOM monomers and dimers contribute significantly to total HOM while trimers only

contributed a minor fraction. Within both the monomer and dimer compounds, a limited set of compounds dominated
the abundance, such as $C_5H_8N_2O_n$, $C_5H_{10}N_2O_n$, $C_{10}H_{17}N_3O_n$, and $C_{10}H_{16}N_2O_n$ series. 2N-monomers, which were
second-generation products, dominated in monomers and accounted for ~34% of all HOM, indicating the important
role of second-generation oxidation in HOM formation in the isoprene+$NO_3$ reaction. Both $RO_2$ autoxidation and
"alkoxy-peroxy" pathways were found to be important for 1N- and 2N-HOM formation. In total, the yield of HOM
monomers, dimers, and trimers accounted for $1.3\%^{+1.3\%}_{-0.7\%}$ of the isoprene reacted, which was much higher than the HOM
yield in the oxidation of isoprene by OH and $O_3$ reported in the literature (Jokinen et al., 2015). This means that the
reaction of isoprene with $NO_3$ is a competitive pathway of HOM formation from isoprene.

The HOM in the reaction of isoprene with $NO_3$ may account for a significant fraction of SOA. If all the

HOM condense on particles, using the molecular weight of the HOM with the least molecular weight observed in
this study ($C_5H_9NO_6$), the HOM yield corresponds to a SOA yield of 3.6%. Although SOA concentrations were not
measured in this study, Ng et al. (2008) reported a SOA yield of the isoprene+$NO_3$ reaction of 4.3%-23.8%. Rollins
et al. (2009) reported a SOA yield of 2% at low organic aerosol loading (~0.52 $\mu g\ m^{-3}$) and 14% if the further
oxidation of the first-generation products are considered in the isoprene+$NO_3$ reaction. Comparing the potential
SOA yield produced by HOM with SOA yields in the literature suggests that HOM may play an important role in the
SOA formation in the isoprene+$NO_3$ reaction.

The $RO_2$ lifetime is approximately 20-50 s in our experiments, which is generally comparable or shorter than

the lifetime of $RO_2$ in the ambient atmosphere at night, varying from several 10 s to several 100 s (Fry et al., 2018),
depending on the $NO_3$, $HO_2$, and $RO_2$ concentrations. Assuming a $HO_2$, $RO_2$, and $NO_3$ concentration of 5 ppt, 5 ppt
(Tan et al., 2019), and 300 ppt (Brown and Stutz, 2012) respectively, the $RO_2$ lifetime in our study is comparable to
the nighttime $RO_2$ lifetime (50 s) found in urban locations and areas influenced by urban plume. In areas with longer
$RO_2$ lifetime such as remote areas, the autoxidation is expected to be more important relative to bimolecular reactions.
This may enhance HOM yield and thus enhance SOA yield. However, on the other hand, at lower $RO_2$ concentration
and thus longer $RO_2$ lifetime, reduced rates of $RO_2+RO_2$ reactions producing low-volatility dimers can reduce the
SOA yield via reducing dimer yield (McFiggans et al., 2019; Pullinen et al., 2020). The $RO_2$ fate in our experiments
is dominated the reaction $RO_2+NO_3$ with significant contribution of $RO_2+RO_2$, which can also represent the $RO_2$ fate
in the urban areas and areas influenced by urban plume. Our experiment condition cannot represent the chemistry in
$HO_2$-dominated regions such as clean forest environment (Schwantes et al., 2015).
We observed the second-generation products formed by the reaction of first-generation products. The
lifetime of first-generation nitrates in the ambient atmosphere, according their rate constants with OH and $NO_3$
(Wennberg et al., 2018), are ~5 h and ~1.3-4 h, respectively, with respect to the reaction with OH and $NO_3$ assuming
a typical OH concentration of $2\times10^6$ molecules $cm^{-3}$ (Lu et al., 2014; Tan et al., 2019) and $NO_3$ concentration of 100-
300 ppt in urban areas (Brown and Stutz, 2012). Therefore, they have the chance to react further with OH and $NO_3$
at dawn. In our experiments, the lifetimes of these first-generation nitrates with respect to OH and $NO_3$ are
comparable to the aforementioned lifetime due to comparable OH and $NO_3$ concentrations with these ambient
conditions. Therefore, our findings on the second-generation products are relevant to the ambient urban atmosphere
and areas influenced by urban plumes. Some of these products such as $C_5H_{8,10}N_2O_8$ and multi-generation
nitrooxyorganosulfates have been observed in recent field studies in polluted megacities in east China (Hamilton et
al., 2021; Xu et al., 2021).
**Data availability**
All the data in the figures of this study are available upon request to the corresponding author (t.mentel@fz-juelich.
de or dfzhao@fudan.edu.cn).
**Competing interests**
The authors declare that they have no conflict of interest.
**Author contribution**
TFM, HF, SS, DZ, IP, AW, and AKS designed the experiments. Instrument deployment and operation were carried
out by IP, HF, SS, IA, RT, FR, DZ, and RW. Data analysis was done by DZ, HF, SS, RW, IA, RT, FR, YG, SK. DZ,
TFM, RW, JW, SK, and LV interpreted the compiled data set. DZ and TFM wrote the paper. All co-authors discussed
the results and commented on the paper.
**Acknowledgements**
We thank the SAPHIR team for supporting our measurements and providing helpful data. D. Zhao and Y. Guo would
like to thank the support of National Natural Science Foundation of China (41875145). We would like to thank three
anonymous reviewers and Kristian Møller for their helpful comments.

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
