# Peer review of "Highly oxygenated organic molecules (HOM) formation in the 1 isoprene oxidation by NO3 radical 2"

_Atmospheric Chemistry and Physics, 2020_

## Referee Comment (RC1) · Anonymous Referee #1 · 18 Dec 2020

Isoprene is one of most critical biogenic VOCs precursor world widely towards forming secondary organic aerosols (SOA). This work investigated detailly the HOM formation from NO3 oxidation of isoprene. Molecules of isoprene-HOM monomer, dimer, and trimer containing 1-5 nitrogen atoms were detected, and their detailed formation pathways were discussed. These HOMs can contribute to SOA significantly globally. I, therefore, recommend this manuscript can be published in ACP after some minor revision.

Specific comments

1. What's the definition of HOM in this work? Does it follow the definition in Bianchi

et al., Chemical Reviews 2019, e.g. contains at least 6 oxygens formed from RO2 auto-oxidation.

2. Did the authors find some molecules that can be identified from NO3 oxidation but not contain any N atom?

3. Line 83-84: There was some discussion on NO3 oxidation of monoterpene to form HOM, e.g. Yan et al., 2016; 2020.

4. Line 149-150: I may suggest adding more statements on how to rule out the reaction with O3 and OH.

5. Line 158-150: may need to add the reference Jokinen et al., ACP, 2012.

6. The first panel of Table 1: why molecules with 1 N atom (one nitrate group) can be formed from isoprene+NO3+NO3.

7. The 2nd panel of Table 2: what is PN?

8. Figure 3: how C5H10N2O7 formed? Besides the two nitrate groups, only one oxygen.

9. Scheme 2: Panel 1: -ONO2 is missed from the 2nd molecule, RO radical (C5H9N2O9.) should not be detected. Panel 2: the 3rd reaction stop should not be H-shift.

10. Scheme 3: Panel 2: the structure of the final molecule maybe not correct.

11. How molecules with 7 H atoms formed? E.g. C5H7N2O9.

12. Is there any observational evidence on the formation of NO3-isoprene-HOM dimer and trimer in the real atmosphere?
* * *

---

## Referee Comment (RC2) · Anonymous Referee #2 · 21 Dec 2020

General:

The authors investigate organic nitrates formed from the oxidation of isoprene with NO3 radicals, illustrate the formation mechanisms of these organic nitrates (including HOM monomer, dimer, and trimers), their yield, and their contribution to SOA yield. The study is well designed and the data are well presented. If the authors can address my points and questions below, I would recommend the publication of the manuscript in Atmospheric Chemistry and Physics.

**Specific:**

1. Line 46 - It's a bit ambiguous for "both nucleation and growth of SOA". HOM are important in nucleation of gaseous vapours, and they contribute to the growth of aerosol particles. Maybe a bit better to say e.g. HOM play a pivotal role in the atmospheric nucleation and also particle growth on pre-existing particles (secondary organic aerosol, SOA).

2. Line 92 - How about the chemical lifetime of the reaction of isoprene with NO3?

3. Line 94 – How significant is the reaction of isoprene with NO3 contributing to NO3 loss at night? Is it dominating in isoprene-dominated region? How about in monoterpene-dominated region? And during the day, how does it compare to the isoprene oxidation with OH?

4. Line 143 – Please add the RH to describe how dry the condition was, e.g. RH<2% or lower. Also add the temperature inside the SAPHIR chamber somewhere in the Experimental part, e.g. line 125.

5. Line 158 - With a mass resolution of 4000, I am a bit curious is it difficult to distinguish different compounds at the same m/z (especially with one dominating compound), such as CHON1 from CHON3,5 compounds, CHON2 from CHON4 compounds, 1N-radicals from 2N-compounds, etc? It would be nicer to show a few masses of peak-fitting results of some organic nitrates in the supplement.

6. Line 163 – Is the wall loss rate the same at different temperature in the SAPHIR chamber? Did you have the same chamber temperature as in Zhao et al., 2018?

7. Line 195-197 – Why are the 2N-monomers dominating over 1N-monomers? Is it (partly) due to the reaction of NO3 radical with the remaining double bond of 1N-monomer (forming 2N-monomers) being more reactive compared to the reaction of NO3 radical with the first double bond (C1) of isoprene (forming 1N-monomers)?

8. Line 254-255 – If the fast loss of C5H9NO10 after isoprene addition is due to faster wall loss, why did the compound decay slower after  $\sim$ 21:40 PM (which I think is partly due to wall loss) compared to those after isoprene addition? Maybe some other reasons are more important for its fast loss. Maybe it is similar to C5H9N2On radicals, that both its reactions and additionally wall loss contribute.

9. Line 264-265 – If the peak intensity of C5H8NO7 radicals in Fig. S4 is plotted in log scale, maybe it's more obvious that it increased during isoprene addition? As far as in the current figure, I cannot see that clearly and can only see it continuously increasing over time. But it's more clear for C5H8NO8 radicals that is responding to isoprene addition.

10. Line 267-270 - For the RO2 in C5H8NOn radicals series, do you mean the radicals with n>=9 cannot be distinguished from 2N-monomers, but C5H8NO7 radicals and C5H8NO8 radicals (you shown in Fig. S4) can be distinguished because they don't have interference compounds nearby them?

11. Line 303 – Other than SOAS, C5 organic nitrates (C5H7-11NO4-9) were also observed in both gas phase and particle phase with FIGAERO-CIMS in a rural area in Germany (Huang et al., EST, 2019), although their measurement site was not an isoprene-dominating region.

12. Line 349-350 – "C5H9N2On radicals M2b were formed from C5H9NO4 followed by an alkoxy-peroxy step" is from Scheme 3b, not from Scheme 2.

13. Line 356 - It's not the case for C5H10N2On and C5H8N2On with n=7. With increasing oxygen number, they increased and then decreased.

14. Line 506 – For these 1N-dimers, have you checked the monoterpene concentration from PTR in the SAPHIR chamber? Is the monoterpene concentration in the chamber low enough not to produce any C10H16NOn molecules to interference/contaminate the results?

15. Line 523-524 – Why only C5H8NOn radicals with n>8 can react with isoprene? The C5H8NOn radicals detected (in Table 1) has an n range of 7-12, and the n range for C10H16NOn are expected to be between 9 and 14, based on R13. But in Table 2,

СЗ

the n range for C10H16NOn is between 10 and 16. Could you infer whether the smaller n (n<=8) for C5H8NOn radicals can work or not to form C10H16NOn from R13? If not, could you give a possible explanation why the smaller n cannot work? And also how was C10H16NO16 formed?

16. Line 568-571 – Can you also check the sesquiterpene concentration in the chamber to exclude the contamination of its products?

17. Line 603-604 – From the results, it's suggesting whether they were first-generation products or second-generation products, or a mix/combination of both, if I didn't misunderstand.

Technical:

1. Line 98 - "initials". Do you mean "initial"?

2. Line 116 – Change to "in the isoprene+NO3 system" throughout the manuscript.

3. Line 123 – The first letter "s" in the word "system" is in Italic.

4. Line 148 – Can you plot the Fig. S1 similar to Fig. S2 so that easier to compare with Fig. S2? Or just label those additions directly in Fig. S2? Because there is no x axis in the SAPHIR box in the figure, but it seems the x axis is time.

5. Line 186 - Please specify in the Figure 1 caption that the m/z has included the reagent ion (15NO3-).

6. Line 204 - (a) the labelled species in the figure 2 caption at m/z 351 and 353 should be C5H8N2O8 and C5H10N2O8 with H[15N]2O6-. Also in the figure itself, they are labeled as C5H8N2O14H[15N]2O6- and C5H10N2O14H[15N]2O6-. Please double check them and correct. (b) I suppose the number on the upper left corner of each compound is the m/z including the reagent ion (15NO3-). Please also specify that in the Fig. 2 caption. (c) What's the blue circle in the figure at m/z around 390?

7. Line 211 –In Scheme 1 caption, it says (b) is for n = 8, 10, 12. But in the Scheme 1b

itself, it's showing compounds with n = 4, 6, 8. Complete the Scheme 1b to show the compounds with n = 10 and 12.

8. Line 235 – Remove the first "in".

9. Line 310 - n = 9, 11 are odd number of oxygen atoms, instead of even number.

10. Line 339 – In Scheme 2a, -ONO2 is missed from the 2nd molecule.

11. Line 345 – In Scheme 3b, -ONO2 is missed from the 2nd molecule.

12. Line 372 – Change "VOCs" to "VOC". You have been using "VOC" previously in the manuscript. So make it consistent.

13. Line 385 – Could you separate these 3N-monomers in Fig. S8 into two panels, or use log scale? It's not obvious to see their time profile, especially for compounds with oxygen number bigger than 12, which were overlapping on top of each other.

14. Line 390 - Change "Such an mechanism" to "Such a mechanism".

Line 438-439 – Could you visualize the mass defect plots in Fig. 4 in a better way? Compound series were so close to each other to see clearly which compound belongs to which line. And also it's better to label the number of oxygen atoms to guide audience since you are discussing a lot of oxygen numbers. Both for Fig. 4 and Fig. 2. For Fig. S11, compound series were not so many and therefore not so difficult to distinguish.

16. Line 443 – Since you have mentioned the compounds clustered with H[15N]2O6-were labelled in grey in Figure 2 caption. Do the same for Figure 4 caption for the compound in grey (C10H17N3O12-14H[15N]2O6-).

17. Line 551-552 - Most abundant trimers were C15H24N4On (n=17-23), but in Fig. S11, n=16-22. Double check. Also double check all the n ranges for each compound series in the figures, tables, and text throughout the manuscript and supplement.

18. Line 556 – The formation pathways of dimer RO2 C10H16N3On (n=14-20) and C10H17N2On are shown in R10 and R11.

---

## Referee Comment (RC3) · Anonymous Referee #3 · 21 Dec 2020

General Comments

This study identifies important HOMs (highly oxygenated organic molecules) from isoprene + NO3 reaction through chamber experiments. The identification of HOMs from NO3 oxidation have been less studied than those from OH or O3 oxidation, so this study fills an important gap in atmospheric chemistry. This study uniquely and in great detail connects many measured compounds to possible mechanistic formation pathways. I suggest this paper be published with some minor revisions as specified below. These minor revisions include some improvements to the mechanistic understanding and providing more information on how to interpret these laboratory results within the

context of how SOA forms from isoprene + NO3 in the ambient atmosphere.

Specific Comments:

Page 5, 149. From the measurements of RO2, HO2, and NO3, can you approximate the fate of the RO2 radical in your experiment? Were conditions such that the RO2 predominantly reacted with another RO2, NO3, or HO2? Do you have an estimate of the lifetime of the RO2 radical in your experiments and how this compares to the RO2 lifetime in the ambient atmosphere. RO2 radical lifetime is often longer in the atmosphere compared to experiments. Would this possibly enhance the SOA yield for ambient conditions for HOMs?

Page 5, line 160. Please provide more detail here on using the H2SO4 sensitivity for the HOMs. Are there certain HOMs this assumption would apply more too? For example, does this assumption apply more to HOMs that are more oxygenated or have a higher C*? Please specify the overall uncertainty in HOMs in the main text (It looks like you calculate this in the supplement). Is there need to add uncertainty here for using the H2SO4 sensitivity directly for the HOM sensitivity?

Figure1. Please add the names for the top m/z on panel b like done for panel a. It looks like many of the top m/z's are the same, but maybe some are unique. It's hard to compare by eye because the m/z lines are so small. Coloring the m/z label by their type listed in the pie chart would also be useful for the reader.

Page 11 line 292: Because you can measure OH and NO3, can you approximate how much isoprene and the first-generation NO3 nitrates react with OH versus NO3 in your experiments? This may lend insight into the products you are detecting. For example, the C5H8O2 compounds mentioned above seems more likely to form from OH oxidation than the H-shift in scheme S1a and S1b (Kwan 2012 Fig 5)? The reaction rate constant for the first-generation nitrates reaction with NO3 is low compared to OH rate constant (Wennberg 2018). From this information, can you connect how your laboratory results should be interpreted to the ambient atmosphere? For example, how long lived are

[Figure]

NO3 derived first-generation nitrates in the ambient atmosphere are they likely to react again with NO3 or with OH at dawn?

Adding pictures of the molecules to schemes S1-S4 would be very beneficial for the reader.

Page 13 line 350. Can you explain how this statement connects with these schemes more. I do not follow as both scheme 2 and scheme 3 have an example of a nitroxyhydroperoxide and a hydroxy nitrate? Also the likelihood of each pathway being relevant in your experiments seems more related to the RO2 fate (i.e., reaction with another RO2 or HO2) than with the loss rate of nitrooxy hydroperoxides and hydroxy nitrates in Ng et al., 2008. Can you include this into your explanation too?

Page 13 line 370: Is it also possible that instead of C5 nitrooxy carbonyls reacting more slowly with NO3 than C5 hydroxy nitrates that instead less HOMs are formed from C5 nitrooxy carbonyls because of the carbonyl group leading to more fragmentation (e.g., in MACR OH oxidation H-shifts lead to losing CO - Crounse 2012)? Have you considered this?

Page 16 line 447: The rate constants for RO2 + RO2 reaction are heavily structure dependent, so this assumption does not really hold in atmospheric chemistry. This should be considered here. For example, in schemes 2 and 3, the dominant RO2 isomers of C5H9N2O9 and C5H9N2O10 will not be the one pictured. The one pictured will most likely lead to HOMs. The dominant one will be the peroxy radical in the tertiary position, which will likely lead to fragmentation and not HOMs. This tertiary peroxy radical will react with other RO2 much more slowly than secondary or primary peroxy radicals (Jenkin 1998, https://doi.org/10.1023/A:1005940332441), so you would not necessarily expect very much ROOR from these RO2 radicals even though they are dominantly detected. Have you considered this?

Page 21 line 588: How was this HOM yield calculated? Is it from the first injection of isoprene or over the entire experiment?

Conclusions: As related to the questions above, please include in more detail how to interpret these laboratory results within the context of how SOA forms from isoprene + NO3 in the ambient atmosphere. How do your laboratory conditions compare to the ambient atmosphere (e.g., RO2 fate (reaction with NO3, RO2, HO2, isomerize), RO2 lifetime, fate of the first-generation organic nitrates reaction with NO3 at night or OH at sunrise)?

Technical comments:

Scheme 2a: missing NO3 group on second molecule. In Scheme 2b, is the 3rd label really a H-shift? It looks like this should be reaction with RO2/NO3?

Scheme 3b: missing NO3 group on second molecule. And the last molecule OOH should be OH?

Page 9 line 235 there are two "in"

Figure S3, isoprene is spelled incorrectly.

---

## Author Comment (AC2) · 26 Apr 2021

**Responses to Referee # 2**

We thank the reviewer for the careful review of our manuscript. The comments and suggestions are greatly appreciated. All the comments have been addressed. In the following, please find our responses to the comments one by one and the corresponding revisions made to the manuscript. The original comments are shown in italics. The revised parts of the manuscript are highlighted.

*Anonymous Referee #2*

*General:*

*The authors investigate organic nitrates formed from the oxidation of isoprene with NO3 radicals, illustrate the formation mechanisms of these organic nitrates (including HOM monomer, dimer, and trimers), their yield, and their contribution to SOA yield. The study is well designed and the data are well presented. If the authors can address my points and questions below, I would recommend the publication of the manuscript in Atmospheric Chemistry and Physics.*

*Specific:*

*1. Line 46 – It's a bit ambiguous for "both nucleation and growth of SOA". HOM are im-portant in nucleation of gaseous vapours, and they contribute to the growth of aerosol particles. Maybe a bit better to say e.g. HOM play a pivotal role in the atmospheric nucleation and also particle growth on pre-existing particles (secondary organic aerosol, SOA).*

**Response:**

Accepted.

In the revised manuscript, we have revised this sentence as follows.

"HOM play a pivotal role in the atmospheric nucleation and also particle growth of pre-existing particles thus contributing to secondary organic aerosol (SOA)."

*2. Line 92 – How about the chemical lifetime of the reaction of isoprene with NO3?*

**Response:**

The chemical lifetime of isoprene with respect to $NO_3$ is calculated to be ~1.6 h and ~600 s using a $NO_3$ concentration of 10 ppt and 100 ppt, respectively. We did not intend to compare the lifetime of isoprene with respect to OH and $NO_3$. In the revised manuscript, we have revised this sentence as follows.

"Although isoprene from plants are mainly emitted under light conditions, i.e., in the daytime, isoprene can remain high after sunset in significant concentrations (Starn et al., 1998; Stroud et al., 2002; Brown et al., 2009) because of the reduced consumption by OH and is found to decay rapidly."

*3. Line 94 – How significant is the reaction of isoprene with NO3 contributing to NO3 loss at night? Is it dominating in isoprene-dominated region? How about in monoterpene-dominated region? And during the day, how does it compare to the isoprene oxidation with OH?*

**Response:**

The contribution of the reaction of isoprene with $NO_3$ to $NO_3$ loss depends on VOC composition. According to a number of field studies, the reaction of isoprene can be the dominant $NO_3$ loss channel in isoprene-dominated region, e.g. in Northeast US (Brown et al., 2009). In the monoterpene-dominated regions such as boreal forests, the reaction of isoprene with $NO_3$ may be not the dominant loss of $NO_3$. During most of the day, the reaction of isoprene with $NO_3$ cannot compete with its reaction with OH due to the fast reaction of $NO_3$ with NO and fast photolysis of $NO_3$. In the late afternoon, the isoprene oxidation by $NO_3$ can be comparable to that by OH under certain conditions e.g. reduced solar radiation and lower NO concentrations, and thus reduced OH concentration and reduced $NO_3$ loss rate (Ayres et al., 2015; Hamilton et al., 2021).

In the revised manuscript, we have modified this sentence to further define the significance of the reaction of isoprene

with NO$_3$ as follows.

"Regarding the budget of NO$_3$, the reaction of isoprene with NO$_3$ can contribute to a significant or even dominant fraction of NO$_3$ loss at night in regions where VOC is dominated by isoprene such as Northeast US (Brown et al., 2009). Under some circumstances, the reaction of isoprene with NO$_3$ can contribute to a significant fraction during the afternoon and afterwards (Ayres et al., 2015; Hamilton et al., 2021)."

*4. Line 143 – Please add the RH to describe how dry the condition was, e.g. RH<2% or lower. Also add the temperature inside the SAPHIR chamber somewhere in the Experimental part, e.g. line 125.*

**Response:**

Accepted.

In the revised manuscript, we have added the description of temperature and RH.

"Experiments were conducted under dry condition (RH<2 %) and temperature was at 302±3 K."

*5. Line 158 – With a mass resolution of 4000, I am a bit curious is it difficult to distinguish different compounds at the same m/z (especially with one dominating compound), such as CHON1 from CHON3,5 compounds, CHON2 from CHON4 compounds, 1N-radicals from 2N-compounds, etc? It would be nicer to show a few masses of peak-fitting results of some organic nitrates in the supplement.*

Response:

Accepted. We agree that it is not always easy to distinguish different compounds at the same m/z as their peaks overlap. Our approach is to "toggle" the inclusion of one peak and check the changes in the residue of peak fitting. If the toggling of inclusion of the peak does not significantly change the residue, we tend to not include the peak. Additionally, we also consider the double bond equivalence in the formula. For example, to distinguish CHON$_2$ from CHON$_4$, as in C$_{10}$H$_{16}$N$_2$O$_{12}$, and C$_{10}$H$_{20}$N$_4$O$_{10}$ at the same unit m/z, we consider 1) it is less likely to form H$_2$0 in the reactions of isoprene with NO$_3$; 2) it is unlikely to form a compound with N4O10 considering the functionality (e.g. a NO$_3$- and a NO- group). When these peaks could not be distinguished, we did not include them in the peak list even if they may be actually present.

In the revised manuscript, we have shown some examples of peak-fitting results in the supplement.

[Figure]

Figure S3. Examples of peak fitting. Formula in grey indicate compounds that have no noticeable effect on fitting residues and thus not included in the peak list.

*6. Line 163 – Is the wall loss rate the same at different temperature in the SAPHIR chamber? Did you have the same chamber temperature as in Zhao et al., 2018?*

**Response:**

The wall loss rate might be influenced by temperature in our chamber. It affects gas-wall equilibrium by changing evaporation rates and affects the condensation by changing diffusion in the boundary layer of the chamber. Because the reaction of NO$_3$ with isoprene cannot be stopped instantaneously as easy as photo-oxidation reactions by switching off illumination, we cannot directly determine the vapor wall loss rate from the experiments themselves. The temperature in this study 302±3 K is not the same as but within the range of our previous photo-oxidation

experiments (298-314 K) (Zhao et al., 2018). Moreover, HOM yield is not sensitive to the vapor wall loss rate. An increase of wall loss rate by 100% or a decrease by 50% only changes the HOM yield by 11% and -6%, respectively. Therefore, we used the vapor loss rate determined in the photo-oxidation experiments. In the revised manuscript, we have discussed the influence of vapor wall loss rate as follows.

"Although the wall loss rate of vapors in this study might not be exactly the same as in our previous photo-oxidation experiments (Zhao et al., 2018), HOM yield is not sensitive to the vapor wall loss rate. An increase of vapor wall loss rate by 100% or a decrease by 50% only changes the HOM yield by 11% and -6%, respectively."

*7. Line 195-197 – Why are the 2N-monomers dominating over 1N-monomers? Is it (partly) due to the reaction of NO3 radical with the remaining double bond of 1Nmonomer (forming 2N-monomers) being more reactive compared to the reaction of NO3 radical with the first double bond (C1) of isoprene (forming 1N-monomers)?*

**Response:**

According to kinetic data of the reaction of $NO_3$ with isoprene and with the first-generation products, isoprene hydroxy nitrate, isoprene carbonyl nitrate, and isoprene peroxy nitrate, the reaction of $NO_3$ with the remaining double is slower than its reaction with isoprene (Wennberg et al., 2018). The higher abundance of HOM 2N-monomers than 1N-monomers is likely because HOM production rate via the autoxidation of 1N-$RO_2$ following the reaction of isoprene with $NO_3$ may be slower than that of the reaction of 1N-monomers (including both HOM and non-HOM monomers) with $NO_3$. We would like to note that some less oxygenated 1N-monomers such as $C_5H_9NO_{4/5}$ and $C_5H_7NO_4$ may have high abundance but are not detected by $NO_3^-$-CIMS and are not HOM and thus not included in HOM 1N-monomers.

In the revised manuscript, we have added the following discussion.

"The higher abundance of HOM 2N-monomers than 1N-monomers is likely because HOM production rate via the autoxidation of 1N-$RO_2$ following the reaction of isoprene with $NO_3$ may be slower than that of the reaction of 1N-monomers (including both HOM and non-HOM monomers) with $NO_3$. We would like to note that some less oxygenated 1N-monomers such as $C_5H_9NO_{4/5}$ and $C_5H_7NO_4$ may have high abundance but are not detected by $NO_3^-$-CIMS and are not HOM and thus not included in HOM 1N-monomers."

*8. Line 254-255 – If the fast loss of C5H9NO10 after isoprene addition is due to faster wall loss, why did the compound decay slower after 21:40 PM (which I think is partly due to wall loss) compared to those after isoprene addition? Maybe some other rea-sons are more important for its fast loss. Maybe it is similar to C5H9N2On radicals, that both its reactions and additionally wall loss contribute.*

**Response:**

We meant the faster decay $C_5H_9NO_{10}$ relative to $C_5H_9NO_6$. After 21:40 PM, $C_5H_9NO_{10}$ showed a decay while $C_5H_9NO_6$ did not.

In the revised manuscript, we have modified this sentence to avoid ambiguity.

"The faster loss of $C_5H_9NO_{10}$ than $C_5H_9NO_6$ may result from the faster wall loss due to its lower volatility."

*9. Line 264-265 – If the peak intensity of C5H8NO7 radicals in Fig. S4 is plotted in log scale, maybe it's more obvious that it increased during isoprene addition? As far as in the current figure, I cannot see that clearly and can only see it continuously increasing over time. But it's more clear for C5H8NO8 radicals that is responding to isoprene addition.*

**Response:**

The description of $C_5H_8NO_7\bullet$ and $C_5H_8NO_8\bullet$ was swapped by mistake and it is $C_5H_8NO_7\bullet$ that showed more mostly a time profile of second-generation products instead. In the revised manuscript, we have corrected this error and re-organized this paragraph as follows.

"The second-generation products may be different isomers formed in pathways other than Scheme 1. Second-generation $C_5H_9NO_6$ can be formed via $C_5H_8NO_7\bullet$, which can also be formed by the reaction of $NO_3$ and $O_2$ with

$C_5H_8O_2$ as mentioned above (Scheme S2b), or by the reaction of OH with $C_5H_7NO_4$ (Scheme S2a). The time profiles of $C_5H_8NO_7\bullet$ did show more contribution of second-generation processes because it continuously increased with time in general. If the pathways via the reaction of $NO_3$ and $O_2$ with $C_5H_8O_2$ and the reaction of OH with $C_5H_7NO_4$ contribute most to $C_5H_9NO_6$, $C_5H_9NO_6$ would show mostly a time profile of second-generation products. Similarly, second-generation $C_5H_9NO_7$ can be formed via $C_5H_8NO_7\bullet$ or $C_5H_8NO_8\bullet$. The time series of $C_5H_8NO_8\bullet$ showed the contribution of both the first- and second-generation processes, which generally increased with time while also responding to isoprene addition (Fig. S4). Similar to $C_5H_9NO_6$, the second-generation pathway for $C_5H_9NO_7$, $C_5H_9NO_9$, and $C_5H_9NO_{10}$ are shown in Scheme S1, S3, S4. For the $RO_2$ in $C_5H_8NO_n\bullet$ series other than $C_5H_8NO_{7/8}\bullet$, the peak of $C_5H_8NO_n\bullet$ overlaps with $C_5H_{10}N_2O_n$ in the mass spectra, which is a much larger peak, and thus cannot be easily differentiated from $C_5H_{10}N_2O_n$. Therefore, it is not possible to obtain reliable separate time profiles in order to differentiate their major sources. It is worth noting that nitrate CIMS may not be able to sensitively detect all isomers of $C_5H_9NO_6$ due to the sensitivity limitation. Therefore, we cannot exclude the possibility that the absence of some first-generation isomers of $C_5H_9NO_6$ was due to the low sensitivity of these isomers."

*10. Line 267-270 – For the RO2 in C5H8NOn radicals series, do you mean the radicals with n>=9 cannot be distinguished from 2N-monomers, but C5H8NO7 radicals and C5H8NO8 radicals (you shown in Fig. S4) can be distinguished because they don't have interference compounds nearby them?*

**Response:**

Yes. To make it more clear, in the revised manuscript, we have modified this sentence as follows.

"For the $RO_2$ in $C_5H_8NO_n\bullet$ series other than $C_5H_8NO_{7/8}\bullet$, the peak of $C_5H_8NO_n\bullet$ overlaps with $C_5H_{10}N_2O_n$ in the mass spectra, which is a much larger peak, and thus cannot be differentiated from $C_5H_{10}N_2O_n$."

*11. Line 303 – Other than SOAS, C5 organic nitrates (C5H7-11NO4-9) were also observed in both gas phase and particle phase with FIGAERO-CIMS in a rural area in Germany (Huang et al., EST, 2019), although their measurement site was not an isoprene-dominating region.*

**Response:**

We thank the reviewer's reminder. In the revised manuscript, we have added this citation as follows.

"$C_5H_xNO_{4-9}$ and $C_5H_xNO_{4-10}$ have been observed in the gas phase and particle, respectively, in a rural area in southwest Germany (Huang et al., 2019)."

*12. Line 349-350 – "C5H9N2On radicals M2b were formed from C5H9NO4 followed by an alkoxy-peroxy step" is from Scheme 3b, not from Scheme 2.*

**Response:**

Corrected.

*13. Line 356 – It's not the case for C5H10N2On and C5H8N2On with n=7. With increasing oxygen number, they increased and then decreased.*

**Response:**

We apologize that Fig. 3b was not updated. In the revised manuscript, we have updated the legend with $C_5H_{10}N_2O_7$ omitted as we are not confident with their assignment. For $C_5H_8N_2O_n$, we have modified this sentence as follows.

"The intensity of $C_5H_8N_2O_n$ first increased and then decreased with oxygen number while $C_5H_{10}N_2O_n$ decreased with oxygen number with the $C_5H_{10}N_2O_8$ and $C_5H_8N_2O_8$ being the most abundant within their respective series."

*14. Line 506 – For these 1N-dimers, have you checked the monoterpene concentration from PTR in the SAPHIR chamber? Is the monoterpene concentration in the chamber low enough not to produce any C10H16NOn molecules to interference/contaminate the results?*

**Response:**

The monoterpene concentration in the chamber during the study is below the limit of detection, which is ~50 ppt ($3\sigma$). Therefore, there is unlikely interference to $C_{10}H_{16}NO_n$ from monoterpenes. In the revised manuscript, we have added

the following note:

"We would like to note that there is unlikely interference to $C_{10}$-HOM from monoterpenes, which has been reported previously (Bernhammer et al., 2018), as the concentration of monoterpenes in the chamber during this study was below the limit of detection, which was ~50 ppt (3σ)."

*15. Line 523-524 – Why only C5H8NOn radicals with n>8 can react with isoprene? The C5H8NOn radicals detected (in Table 1) has an n range of 7-12, and the n range for C10H16NOn are expected to be between 9 and 14, based on R13. But in Table 2, the n range for C10H16NOn is between 10 and 16. Could you infer whether the smaller n (n<=8) for C5H8NOn radicals can work or not to form C10H16NOn from R13? If not, could you give a possible explanation why the smaller n cannot work? And also how was C10H16NO16 formed?*

**Response:**

We have further checked our data and the n range for $C_{10}H_{16}NO_n$ should be 10-14. In the revised manuscript, we have corrected this. We expected the n range of $C_{10}H_{16}NO_n\bullet$ to be 7-14 because besides the $C_5H_8NO_n\bullet$ detected by our $NO_3^-$-CIMS, there should be $C_5H_8NO_n(n=5,6)$ according to the reaction mechanism. Among these $C_{10}H_{16}NO_n\bullet$ compounds, $C_{10}H_{16}NO_{7-9}\bullet$ are expected to be detectable by $NO_3^-$-CIMS. The absence of these compounds in the mass spectra is likely attributed to their low concentration, which might result from low precursor concentration, low reaction rate with isoprene, and/or fast reaction with other radicals. In addition, we would like to note that the $C_{10}H_{16}NO_n\bullet$ series has low signal in the mass spectra and their assignment and thus range of n may be subject to uncertainties. In the revised, we have added this note and revised this part as follows.

"Only $C_{10}H_{16}NO_n\bullet$ with n≥10 were detected, while according to the mechanism of self-reaction between $C_5H_8NO_n\bullet$, the n range of $C_{10}H_{16}NO_n\bullet$ is expected to be 7-14. The absence of $C_{10}H_{16}NO_{n(n<10)}\bullet$ is likely attributed to their low abundance, which might result from low precursor concentrations, low reaction rates with isoprene, and/or faster reactive losses with other radicals."

"We note that due to their low signals in the mass spectra, their assignment and thus range of n may be subject to uncertainties."

*16. Line 568-571 – Can you also check the sesquiterpene concentration in the chamber to exclude the contamination of its products?*

**Response:**

The sesquiterpene concentration in the chamber during this study was below the limit of detection, which was ~50 ppt (3σ). Therefore, there is unlikely contamination from sesquiterpene.

*17. Line 603-604 – From the results, it's suggesting whether they were first-generation products or second-generation products, or a mix/combination of both, if I didn't misunderstand.*

**Response:**

Yes. In the revised manuscript, we have revised this sentence as follows.

"The time profiles provide additional constraints on their formation mechanism beside the molecular formula, suggesting whether they were first-generation products, second-generation products or a combination of both."

*Technical:*

*1. Line 98 – "initials". Do you mean "initial"?*

**Response:**

In the revised manuscript, we have changed it to "initial".

*2. Line 116 – Change to "in the isoprene+NO3 system" throughout the manuscript.*

**Response:**

In the revised manuscript, we have changed the phrase to "in the isoprene+$NO_3$ reaction" throughout the manuscript.

*3. Line 123 – The first letter "s" in the word "system" is in Italic.*

**Response:** Corrected.

*4. Line 148 – Can you plot the Fig. S1 similar to Fig. S2 so that easier to compare with Fig. S2? Or just label those additions directly in Fig. S2? Because there is no x axis in the SAPHIR box in the figure, but it seems the x axis is time.*

**Response:**

Accepted. In the revised manuscript, we have added further labels in Fig. S2 to show the times of additions of isoprene, $NO_2$, and $O_3$.

*5. Line 186 – Please specify in the Figure 1 caption that the m/z has included the reagent ion (15NO3-).*

**Response:** Accepted.

*6. Line 204 – (a) the labelled species in the figure 2 caption at m/z 351 and 353 should be C5H8N2O8 and C5H10N2O8 with H[15N]2O6-. Also in the figure itself, they are labeled as C5H8N2O14H[15N]2O6- and C5H10N2O14H[15N]2O6-. Please double check them and correct. (b) I suppose the number on the upper left corner of each compound is the m/z including the reagent ion (15NO3-). Please also specify that in the Fig. 2 caption. (c) What's the blue circle in the figure at m/z around 390?*

**Response:** We thank the reviewer for pointing out this typo. We have corrected the captions and labeling in the figure, and also added the note of the m/z.

The blue circle is $C_6F_{11}HO_3^-$, coming from chamber wall. In the revised manuscript, we chose to not plot this compound in the figure to avoid confusion.

*7. Line 211 – In Scheme 1 caption, it says (b) is for n = 8, 10, 12. But in the Scheme 1b itself, it's showing compounds with n = 4, 6, 8. Complete the Scheme 1b to show the compounds with n = 10 and 12.*

**Response:**

Accepted. In the revised manuscript, we have added the pathway to form $C_5H_9NO_{10}\bullet$. The peak of $C_5H_9NO_{12}\bullet$ overlaps with $C_5H_9NO_8\bullet H(^{15}NO_3)_2^-$. After further checking the spectra, to be cautious we decided to not assign it.

*8. Line 235 – Remove the first "in".*

**Response:** accepted.

*9. Line 310 – n = 9, 11 are odd number of oxygen atoms, instead of even number.*

**Response:**

We have corrected this error in the revised manuscript.

*10. Line 339 – In Scheme 2a, -ONO2 is missed from the 2nd molecule.*

**Response:**

We have corrected this mislabeling in the revised manuscript.

*11. Line 345 – In Scheme 3b, -ONO2 is missed from the 2nd molecule.*

**Response:**

We have corrected this mislabeling in the revised manuscript.

*12. Line 372 – Change "VOCs" to "VOC". You have been using "VOC" previously in the manuscript. So make it consistent.*

**Response:**

Accepted.

*13. Line 385 – Could you separate these 3N-monomers in Fig. S8 into two panels, or use log scale? It's not obvious to see their time profile, especially for compounds with oxygen number bigger than 12, which were overlapping on top of each other.*

**Response:**

Accepted. We use two panels to show them in the revised manuscript.

*14. Line 390 – Change "Such an mechanism" to "Such a mechanism".*

**Response:** Accepted.

*15. Line 438-439 – Could you visualize the mass defect plots in Fig. 4 in a better way? Compound series were so close to each other to see clearly which compound belongs to which line. And also it's better to label the number of oxygen atoms to guide audience since you are discussing a lot of oxygen numbers. Both for Fig. 4 and Fig. 2. For Fig. S11, compound series were not so many and therefore not so difficult to distinguish.*

**Response:** Accepted. In the revised manuscript, we have added the number of oxygen atoms. Also we have adjusted the scale of y axis, removed the grid line, and colored the HOM series lines in different colors to show the data points more clearly.

*16. Line 443 – Since you have mentioned the compounds clustered with H[15N]2O6- were labelled in grey in Figure 2 caption. Do the same for Figure 4 caption for the compound in grey (C10H17N3O12-14H[15N]2O6-).*

**Response:** Accepted.

*17. Line 551-552 – Most abundant trimers were C15H24N4On (n=17-23), but in Fig. S11, n=16-22. Double check. Also double check all the n ranges for each compound series in the figures, tables, and text throughout the manuscript and supplement.*

**Response:** We thank the reviewer for pointing this inconsistency. In the revised manuscript, we have corrected the numbers. Also we have checked throughout the manuscript for all series and fixed the inconsistency in the n ranges.

*18. Line 556 – The formation pathways of dimer RO2 C10H16N3On (n=14-20) and C10H17N2On are shown in R10 and R11.*

**Response:**

In the revised manuscript, we have corrected this error.

**References**

Ayres, B. R., Allen, H. M., Draper, D. C., Brown, S. S., Wild, R. J., Jimenez, J. L., Day, D. A., Campuzano-Jost, P., Hu, W., de Gouw, J., Koss, A., Cohen, R. C., Duffey, K. C., Romer, P., Baumann, K., Edgerton, E., Takahama, S., Thornton, J. A., Lee, B. H., Lopez-Hilfiker, F. D., Mohr, C., Wennberg, P. O., Nguyen, T. B., Teng, A., Goldstein, A. H., Olson, K., and Fry, J. L.: Organic nitrate aerosol formation via NO3 + biogenic volatile organic compounds in the southeastern United States, Atmos. Chem. Phys., 15, 13377-13392, 10.5194/acp-15-13377-2015, 2015.

Brown, S. S., deGouw, J. A., Warneke, C., Ryerson, T. B., Dube, W. P., Atlas, E., Weber, R. J., Peltier, R. E., Neuman, J. A., Roberts, J. M., Swanson, A., Flocke, F., McKeen, S. A., Brioude, J., Sommariva, R., Trainer, M., Fehsenfeld, F. C., and Ravishankara, A. R.: Nocturnal isoprene oxidation over the Northeast United States in summer and its impact on reactive nitrogen partitioning and secondary organic aerosol, Atmos. Chem. Phys., 9, 3027-3042, 10.5194/acp-9-3027-2009, 2009.

Hamilton, J. F., Bryant, D. J., Edwards, P. M., Ouyang, B., Bannan, T. J., Mehra, A., Mayhew, A. W., Hopkins, J. R., Dunmore, R. E., Squires, F. A., Lee, J. D., Newland, M. J., Worrall, S. D., Bacak, A., Coe, H., Percival, C., Whalley, L. K., Heard, D. E., Slater, E. J., Jones, R. L., Cui, T., Surratt, J. D., Reeves, C. E., Mills, G. P., Grimmond, S., Sun, Y., Xu, W., Shi, Z., and Rickard, A. R.: Key Role of NO3 Radicals in the Production of Isoprene Nitrates and Nitrooxyorganosulfates in Beijing, Environ. Sci. Technol., 55, 842-853, 10.1021/acs.est.0c05689, 2021.

Starn, T. K., Shepson, P. B., Bertman, S. B., Riemer, D. D., Zika, R. G., and Olszyna, K.: Nighttime isoprene chemistry at an urban-impacted forest site, 103, 22437-22447, https://doi.org/10.1029/98JD01201, 1998.

Stroud, C. A., Roberts, J. M., Williams, E. J., Hereid, D., Angevine, W. M., Fehsenfeld, F. C., Wisthaler, A., Hansel, A., Martinez-Harder, M., Harder, H., Brune, W. H., Hoenninger, G., Stutz, J., and White, A. B.: Nighttime isoprene trends at an urban forested site during the 1999 Southern Oxidant Study, 107, ACH 7-1-ACH 7-14, https://doi.org/10.1029/2001JD000959, 2002.

Wennberg, P. O., Bates, K. H., Crounse, J. D., Dodson, L. G., McVay, R. C., Mertens, L. A., Nguyen, T. B., Praske, E., Schwantes, R. H., Smarte, M. D., St Clair, J. M., Teng, A. P., Zhang, X., and Seinfeld, J. H.: Gas-Phase Reactions of Isoprene and Its Major Oxidation Products, Chem. Rev., 118, 3337-3390, 10.1021/acs.chemrev.7b00439, 2018.

---

## Author Comment (AC3) · 26 Apr 2021

**Responses to Referee # 3**

We thank the reviewer for the careful review of our manuscript. The comments and suggestions are greatly appreciated. All the comments have been addressed. In the following, please find our responses to the comments one by one and the corresponding revisions made to the manuscript. The original comments are shown in italics. The revised parts of the manuscript are highlighted.

*Anonymous Referee #3*

*General Comments*

*This study identifies important HOMs (highly oxygenated organic molecules) from isoprene + NO3 reaction through chamber experiments. The identification of HOMs from NO3 oxidation have been less studied than those from OH or O3 oxidation, so this study fills an important gap in atmospheric chemistry. This study uniquely and in great detail connects many measured compounds to possible mechanistic formation pathways.*

*I suggest this paper be published with some minor revisions as specified below.*

*These minor revisions include some improvements to the mechanistic understanding and providing more information on how to interpret these laboratory results within the context of how SOA forms from isoprene + NO3 in the ambient atmosphere.*

*Specific Comments:*

*Page 5, 149. From the measurements of RO2, HO2, and NO3, can you approximate the fate of the RO2 radical in your experiment? Were conditions such that the RO2 predominantly reacted with another RO2, NO3, or HO2? Do you have an estimate of the lifetime of the RO2 radical in your experiments and how this compares to the RO2 lifetime in the ambient atmosphere. RO2 radical lifetime is often longer in the atmosphere compared to experiments. Would this possibly enhance the SOA yield for ambient conditions for HOMs?*

**Response:**

In our experiments we measured radical concentrations directly. The reaction of $RO_2$ with $NO_3$ was determined to dominate over the reaction $RO_2$ with $RO_2$ and with $HO_2$ and $HO_2$ concentration, although the measured $RO_2$ and $HO_2$ are subject to uncertainties due to interference from $NO_3$. This is consistent with our understanding of the reaction system as there is no extra $HO_2$ source (Vereecken et al., 2021). However, a recent study found that a large portion of $RO_2$ is not measured by LIF and thus $RO_2$ was underestimated (Vereecken et al. 2021). Therefore, we expect the reaction of $RO_2+RO_2$ to be also important. Overall, we estimate that the $RO_2$ fate is dominated the reaction $RO_2+NO_3$ with significant contribution of $RO_2+RO_2$.

We also estimated $RO_2$ lifetime using the measured $RO_2$, $NO_3$, and $HO_2$ concentrations. The $RO_2$ lifetime is approximately 20-50 s in our experiments, which is generally comparable or shorter than the lifetime of $RO_2$ in the ambient atmosphere at night, varying from several 10 s to several 100 s (Fry et al., 2018), depending on the $NO_3$, $HO_2$, and $RO_2$ concentrations as well as specific $RO_2$. Assuming a $HO_2$, $RO_2$, and $NO_3$ concentration of 5 ppt, 5 ppt (Tan et al., 2019), and 300 ppt (Brown and Stutz, 2012) respectively, the $RO_2$ lifetime in our study is representative

of nighttime $RO_2$ lifetime of urban atmosphere.

If the $RO_2$ lifetime increased, the autoxidation of $RO_2$ would be more important relative to its bimolecular reactions. This may enhance HOM yield and thus enhance SOA yield. However, on the other hand, reduced rate of $RO_2+RO_2$ at longer $RO_2$ lifetime, producing low-volatility dimers, can reduce the SOA yield via reducing dimer yield (McFiggans et al., 2019; Pullinen et al., 2020).

In the revised manuscript, we have added discussion on the fate and lifetime of $RO_2$.

"In these experiments, $RO_2$ fate is estimated to be dominated by its reaction with $NO_3$ according to the measured $NO_3$, $RO_2$, and $HO_2$ concentration and their rate constants for the reactions with $RO_2$   (MCM v3.2(Jenkin et al., 1997; Jenkin et al., 2003; Saunders et al., 2003; Jenkin et al., 2015), via website: http://mcm.leeds.ac.uk/MCM) despite uncertainties of the measured $RO_2$ and $HO_2$ concentration due to interference from $NO_3$. As a large portion of $RO_2$ is not measured by LIF (Vereecken et al. 2021) and thus $RO_2$ is underestimated, we expected the reaction of $RO_2+RO_2$ to be also important. Overall, we estimate that the $RO_2$ fate is dominated the reaction $RO_2+NO_3$ with significant contribution of $RO_2+RO_2$."

"The $RO_2$ fate is dominated the reaction $RO_2+NO_3$ with significant contribution of $RO_2+RO_2$, which can also represent the $RO_2$ fate in the urban areas and areas influenced by urban plume. Yet, it cannot represent the chemistry in $HO_2$-dominated regions such as clean forest environment (Schwantes et al., 2015)."

"The $RO_2$ lifetime is approximately 20-50 s in our experiments, which is generally comparable or shorter than the lifetime of $RO_2$ in the ambient atmosphere at night, varying from several 10 s to several 100 s (Fry et al., 2018), depending on the $NO_3$, $HO_2$, and $RO_2$ concentrations. Assuming a $HO_2$, $RO_2$, and $NO_3$ concentration of 5 ppt, 5 ppt (Tan et al., 2019), and 300 ppt (Brown and Stutz, 2012) respectively, the $RO_2$ lifetime in our study is comparable to the nighttime $RO_2$ lifetime (50 s) found in urban locations and areas influenced by urban plume. In areas with longer $RO_2$ lifetime such as remote areas, the autoxidation of $RO_2$ is expected to be more important relative to its bimolecular reactions. This may enhance HOM yield and thus enhance SOA yield. However, on the other hand, at lower $RO_2$ concentration and thus longer $RO_2$ lifetime, reduced rates of $RO_2+RO_2$ reactions producing low-volatility dimers can reduce the SOA yield via reducing dimer yield (McFiggans et al., 2019; Pullinen et al., 2020)."

*Page 5, line 160. Please provide more detail here on using the H2SO4 sensitivity for the HOMs. Are there certain HOMs this assumption would apply more too? For example, does this assumption apply more to HOMs that are more oxygenated or have a higher C\*? Please specify the overall uncertainty in HOMs in the main text (It looks like you calculate this in the supplement). Is there need to add uncertainty here for using the H2SO4 sensitivity directly for the HOM sensitivity?*

**Response:**

Accepted.

In the revised manuscript, we have added more details on the $H_2SO_4$ sensitivity. Also we have added the overall uncertainty in HOM yield, which was shown in the supplement.

Since HOM contain more than six oxygen atoms and their cluster with nitrate ions are quite stable (Ehn et al., 2014),

the charge efficiency of HOM is thus assumed to be equal to that of $H_2SO_4$, which is close to the collision limit (Viggiano et al., 1997). We do not expect this sensitivity applies more to certain HOM than other HOM since HOM are all highly oxygenated with multiple functional groups. If HOM do not charge with nitrate ions at their collision limit or the cluster formed break during the short residence time in the charger, its concentration would be underestimated as pointed by Ehn et al. (2014). Thus, our assumption provides a lower limit of the HOM concentration. We have further discussed the uncertainty of using the sensitivity of $H_2SO_4$ for HOM as follows.

"Since HOM contain more than six oxygen atoms and their clusters with nitrate ions are quite stable (Ehn et al., 2014), the charge efficiency of HOM is thus assumed to be equal to that of $H_2SO_4$, which is close to the collision limit (Viggiano et al., 1997). If HOM do not charge with nitrate ions at their collision limit or the clusters formed break during the short residence time in the charger, its concentration would be underestimated as pointed by Ehn et al. (2014). Thus, our assumption provides a lower limit of the HOM concentration."

*Figure1. Please add the names for the top m/z on panel b like done for panel a. It looks like many of the top m/z's are the same, but maybe some are unique. It's hard to compare by eye because the m/z lines are so small. Coloring the m/z label by their type listed in the pie chart would also be useful for the reader.*

**Response:**

Accepted.

*Page 11 line 292: Because you can measure OH and NO3, can you approximate how much isoprene and the first-generation NO3 nitrates react with OH versus NO3 in your experiments? This may lend insight into the products you are detecting. For example, the C5H8O2 compounds mentioned above seems more likely to form from OH oxidation than the H-shift in scheme S1a and S1b (Kwan 2012 Fig 5)? The reaction rate constant for the first-generation nitrates reaction with NO3 is low compared to OH rate constant (Wennberg 2018). From this information, can you connect how your laboratory results should be interpreted to the ambient atmosphere? For example, how long lived are NO3 derived first-generation nitrates in the ambient atmosphere are they likely to react*

*again with NO3 or with OH at dawn?*

**Response:**

In our experiments, the reaction of OH with isoprene contributed less to the isoprene consumption than the reaction of $O_3$ with isoprene as the OH is mainly formed by isoprene+$O_3$ and the OH yield is less than one. Therefore, the reaction of OH with isoprene is negligible (<3%) for isoprene loss. This is consistent with the contribution determined using measured OH concentration, despite some uncertainty of measured OH concentration due to the interference by $NO_3$. In light of the negligible role of OH in isoprene consumption, we think that $C_5H_8O_2$ is more likely formed by the reaction with $NO_3$ and subsequent H-shift.

In the revised manuscript, we have discussed the role of OH in isoprene consumption.

"The contribution of the reaction of isoprene with trace amount of OH, mainly produced in the reaction of isoprene+$O_3$ via Criegee intermediates (Nguyen et al., 2016), is negligible as the OH yield is less than one (Malkin et al., 2010) and thus its contribution is less than that of isoprene+$O_3$. This is consistent with the contribution

For the first-generation $NO_3$ nitrates, their reaction rates with OH and with $NO_3$ calculated using the reaction constants (Wennberg et al., 2018) and OH and $NO_3$ concentrations are comparable, with both contributing significantly to the loss of first-generation $NO_3$ nitrates as the rate constant with OH is much higher than that with $NO_3$.

Regarding the lifetime of first-generation nitrates in the ambient atmosphere, according their rate constants with OH and $NO_3$ (Wennberg et al., 2018), their lifetime are 5 h and 1.3-4 h, respectively, with respect to the reaction with OH and $NO_3$ assuming a typical OH concentration of $2\times10^6$ molecules $cm^{-3}$ (Lu et al., 2014; Tan et al., 2019) and $NO_3$ concentration of 100-300 ppt in urban areas. Therefore, they likely react further with OH and $NO_3$ at dawn. Therefore, our results are relevant to the ambient urban atmosphere and areas influenced by urban plumes.

In the revised manuscript, we have added discussion on the relevance to ambient atmosphere as follows.

"We observed the second-generation products formed by the reaction of first-generation products. The lifetime of first-generation nitrates in the ambient atmosphere, according their rate constants with OH and $NO_3$ (Wennberg et al., 2018), are ~5 h and ~1.3-4 h, respectively, with respect to the reaction with OH and $NO_3$ assuming a typical OH concentration of $2\times10^6$ molecules $cm^{-3}$ (Lu et al., 2014; Tan et al., 2019) and $NO_3$ concentration of 100-300 ppt in urban areas (Brown and Stutz, 2012). Therefore, they have the chance to react further with OH and $NO_3$ at dawn. In our experiments, the lifetimes of these first-generation nitrates with respect to OH and $NO_3$ are comparable to the aforementioned lifetime due to comparable OH and $NO_3$ concentrations with these ambient conditions. Therefore, our findings on the second-generation products are relevant to the ambient urban atmosphere and areas influenced by urban plumes. Some of these products such as $C_5H_{810}N_2O_8$ and multi-generation nitrooxyorganosulfates have been observed in recent field studies in polluted megacities in east China (Hamilton et al., 2021; Xu et al., 2021)."

"3N dimer such as $C_5H_9N_3O_{10}$ as well as 2N-monomers such as $C_5H_8N_2O_8$   and $C_5H_8N_2O_{10}$ have been observed in a recent field study in polluted cities in east China (Xu et al., 2021)."

*Adding pictures of the molecules to schemes S1-S4 would be very beneficial for the reader.*

**Response:**

Accepted. We would also like to note that these schemes and other schemes in this study only show example isomers and pathways to form these molecules. It is likely that many of the reactions occurring are not the dominant channels as otherwise there would be much higher HOM yield. We have added these notes in the revised manuscript.

"We would like to note that the scheme and other schemes in this study only show example isomers and pathways to form these molecules. It is likely that many of the reactions occurring are not the dominant channels as otherwise there would be much higher HOM yield as discussed below."

*Page 13 line 350. Can you explain how this statement connects with these schemes more. I do not follow as both scheme 2 and scheme 3 have an example of a nitroxyhydroperoxide and a hydroxy nitrate? Also the likelihood of each pathway being relevant in your experiments seems more related to the RO2 fate (i.e., reaction with another RO2*

*or HO2) than with the loss rate of nitrooxy hydroperoxides and hydroxy nitrates in Ng et al., 2008. Can you include this into your explanation too?*

**Response:**

We apologize that there is a typo here. We meant that the reactions with hydroxy nitrate is more likely. In the revised manuscript, we have modified this sentence. As the reviewer pointed out, $RO_2$ fate is also an important factor, which affects the relative concentration of nitroxyhydroperoxide and hydroxy nitrate. In our experiments, $RO_2$ fate is dominated by the reaction with $NO_3$ and $RO_2$. Therefore, hydroxyl nitrate is expected to be higher than nitroxyhydroperoxide. Overall, we have revised the discussion as follows:

"Additionally, C5-hydroxynitrate concentration is expected to be higher than that of nitrooxyhydroperoxides because $RO_2+RO_2$ forming alcohol is likely more important than $RO_2+HO_2$ forming hydroperoxide in this study. Therefore, it is likely that $C5H_9N_2O_n\bullet$ M2a series was mainly formed from $C_5H_9NO_4$ instead of $C_5H_9NO_5$, while $C_5H_9N_2O_n\bullet$ M2b were formed from $C_5H_9NO_4$ followed by an alkoxy-peroxy step. That is, Scheme 2a and 3b were more likely."

*Page 13 line 370: Is it also possible that instead of C5 nitrooxy carbonyls reacting more slowly with NO3 than C5 hydroxy nitrates that instead less HOMs are formed from C5 nitrooxy carbonyls because of the carbonyl group leading to more fragmentation (e.g., in MACR OH oxidation H-shifts lead to losing CO - Crounse 2012)? Have you considered this?*

**Response:**

It is possible that more fragmentation in the further H-shift of peroxy radicals formed in the reaction of C5 nitrooxy carbonyls can also contribute to our observation of low abundance of $C_5H_7N_2O_n$. In the revised manuscript, we have added discussion on this point.

"This fact is consistent with the finding of Ng et al. (2008) that C5-nitrooxycarbonyls react slowly with $NO_3$. Additionally, the peroxy radical formed in the reaction of C5-nitrooxycarbonyls with $NO_3$ likely leads to more fragmentation in H-shift as found in the OH oxidation of methacrolein (Crounse et al., 2012), which may also contribute to the low abundance of $C_5H_7N_2O_n$."

*Page 16 line 447: The rate constants for RO2 + RO2 reaction are heavily structure dependent, so this assumption does not really hold in atmospheric chemistry. This should be considered here. For example, in schemes 2 and 3, the dominant RO2 isomers of C5H9N2O9 and C5H9N2O10 will not be the one pictured. The one pictured will most likely lead to HOMs. The dominant one will be the peroxy radical in the tertiary position, which will likely lead to fragmentation and not HOMs. This tertiary peroxy radical will react with other RO2 much more slowly than secondary or primary peroxy radicals (Jenkin 1998, (Jenkin et al., 1998)), so you would not necessarily expect very much ROOR from these RO2 radicals even though they are dominantly detected. Have you considered this?*

**Response:**

We agree with the reviewer that tertiary $RO_2$ formed may react slower than secondary and primary $RO_2$. $C_5H_9N_2O_9\bullet$ or $C_5H_9N_2O_{10}\bullet$ is likely a mixture of different isomers, including both the secondary and tertiary $RO_2$. And we have

no evidence that $C_5H_9N_2O_9\bullet$ or $C_5H_9N_2O_{10}\bullet$ contains more fraction of tertiary $RO_2$ than primary and secondary $RO_2$. For example, the dominant precursor of $C_5H_9N_2O_9\bullet$, isoprene hydroxyl nitrates, are likely (1-$ONO_2$, 2-OH) and (1-$ONO_2$, 4-OH) isoprene hydroxyl nitrate (IHN). The dominant $RO_2$ formed by the reaction of these IHN with $NO_3$ is likely secondary or tertiary $RO_2$. In the revised manuscript, we have added discussion on different reaction rate of different $RO_2$.

"Admittedly, the assumption of different $RO_2$ having similar rate constant in accretion reactions may not be valid. For example, self-reaction of tertiary $RO_2$ is slower than secondary and primary $RO_2$ (Jenkin et al., 1998; Finlayson-Pitts and Pitts, 2000). Different rate constant may also lead to the observation that the most abundant dimers could not be explained the most abundant $RO_2$."

*Page 21 line 588: How was this HOM yield calculated? Is it from the first injection of isoprene or over the entire experiment?*

**Response:**

HOM yield is calculated for the first isoprene addition period. In the revised manuscript, we have described this and used "primary HOM yield" in place of "HOM yield" to avoid ambiguity.

"The HOM yield in the oxidation of isoprene by $NO_3$ was estimated for using the sensitivity of $H_2SO_4$. It was derived for the first isoprene addition period to minimize the contribution of multi-generation products and to better compare with the data in literature, thus denoted as primary HOM yield (Pullinen et al., 2020) and was estimated to be 1.2% $^{+1.3\%}_{-0.7\%}$ . "

*Conclusions: As related to the questions above, please include in more detail how to interpret these laboratory results within the context of how SOA forms from isoprene + NO3 in the ambient atmosphere. How do your laboratory conditions compare to the ambient atmosphere (e.g., RO2 fate (reaction with NO3, RO2, HO2, isomerize), RO2 lifetime, fate of the first-generation organic nitrates reaction with NO3 at night or OH at sunrise)?*

**Response:**

Accepted. We have included more discussion on the ambient relevance of our laboratory study in the conclusion part as mentioned in the response to former comments.

*Technical comments:*

*Scheme 2a: missing NO3 group on second molecule. In Scheme 2b, is the 3rd label really a H-shift? It looks like this should be reaction with RO2/NO3?*

**Response:**

We thank the reviewer for pointing out our mis-labelings. In the revised manuscript, we have corrected them.

*Scheme 3b: missing NO3 group on second molecule. And the last molecule OOH should be OH?*

**Response:**

We thank the reviewer for pointing out our mis-labelings. In the revised manuscript, we have corrected them.

*Page 9 line 235 there are two "in"*

**Response:**

Corrected.

*Figure S3, isoprene is spelled incorrectly.*

**Response:**

Corrected.

**References:**

[revised manuscript text omitted]

---

## Author Response (AR1)

**Responses to Referee #1**

We thank the reviewer for the careful review of our manuscript. The comments and suggestions are greatly appreciated. All the comments have been addressed. In the following, please find our responses to the comments one by one and the corresponding revisions made to the manuscript. The original comments are shown in italics. The revised parts of the manuscript are highlighted.

**Anonymous Referee #1**

Received and published: 18 December 2020

Isoprene is one of most critical biogenic VOCs precursor world widely towards forming secondary organic aerosols (SOA). This work investigated detailly the HOM formation from NO3 oxidation of isoprene. Molecules of isoprene-HOM monomer, dimer, and trimer containing 1-5 nitrogen atoms were detected, and their detailed formation pathways were discussed. These HOMs can contribute to SOA significantly globally. I, therefore, recommend this manuscript can be published in ACP after some minor revision.

Specific comments

1. What's the definition of HOM in this work? Does it follow the definition in Bianchi et al., Chemical Reviews 2019, e.g. contains at least 6 oxygens formed from RO2 auto-oxidation.

**Response:**

In our study, we used the same definition of HOM as by Bianchi et al. (2019). In the revised manuscript, we have clarified HOM definition explicitly.

"In this study we refer to the definition of HOM by Bianchi et al. (2019), i.e., HOM typically contain six or more oxygen atoms formed via autoxidation and related chemistry of peroxy radicals."

2. Did the authors find some molecules that can be identified from NO3 oxidation but not contain any N atom?

**Response:**

We have identified ~400 peaks in total and most HOM contain nitrogen atoms with few exceptions such as  $C_5H_{10}O_8$  and  $C_5H_8O_{11}$ , which were minor peaks (<~1% of the maximum peak).  $C_5H_8O_{10/11}$  overlap with the peak of  $C_5H_{10}NO_{9/10}$ , and hence cannot be assigned with high confidence. In the revised manuscript, we have added the following information about this question.

"Almost all peaks are assigned HOM containing nitrogen atoms with possibly few exceptions such as  $C_5H_{10}O_8$ and  $C_5H_8O_{11}$  with very minor abundance (<~1% of the maximum peak)."

3. Line 83-84: There was some discussion on NO3 oxidation of monoterpene to form HOM, e.g. Yan et al., 2016; 2020.

**Response:**

In the revised manuscript, we have added more references including Yan et al. (2020) and Yan et al. (2016).

4. Line 149-150: I may suggest adding more statements on how to rule out the reaction with O3 and OH.

**Response:**

In our study, the reaction of isoprene with  $O_3$  accounted for ~3% of the isoprene consumption for the whole reaction period and <1% in the first isoprene addition period. As OH is mainly formed via Criegee intermediates in isoprene+O3 (Nguyen et al., 2016), the reaction of isoprene with OH contributes less than that isoprene+O3. Thus isoprene + OH was negligible because the OH yield is less than one (Malkin et al., 2010). This is consistent with our finding based on OH concentrations measured by LIF that the reaction isoprene with OH only contributed very minor fraction of isoprene consumption. In the revised manuscript, we have further discussed this topic.

"Experiments were designed such that the chemical system was dominated by the reaction of isoprene with NO3 and the reaction of isoprene with O3 did not play a major role (<3% of the isoprene consumption)."

"The contribution of the reaction of isoprene with trace amount of OH, mainly produced in the reaction of isoprene+O3 via Criegee intermediates (Nguyen et al., 2016), is negligible as the OH yield is less than one (Malkin et al., 2010) and thus its contribution is less than that of isoprene+O3. This is consistent with the contribution determined using measured OH concentration, despite some uncertainty in measured OH concentration due to the interference from  $NO_3$ ."

5. Line 158-150: may need to add the reference Jokinen et al., ACP, 2012.

**Response:**

Accepted.

6. The first panel of Table 1: why molecules with 1 N atom (one nitrate group) can be formed from isoprene+NO3+NO3.

**Response:**

As we discussed in L230-235 and Scheme S1b, the reaction of isoprene with NO3 can form products containing no N atoms, such as C5H8O2. C5H8O2 can further react with NO3 forming products containing one N atoms. *7. The 2nd panel of Table 2: what is PN?*

**Response:**

PN denotes peroxynitrate. In the revised manuscript, we have changed it to "RO2NO2" and added the explanation in the caption of Table 1, as follows.

"RO2 denotes peroxy radical and ROOH, ROH, R=O, and  $RO_2NO_2$  denote the termination products containing hydroperoxy, hydroxyl, carbonyl group, and peroxynitrate, respectively."

8. Figure 3: how C5H10N2O7 formed? Besides the two nitrate groups, only one oxygen.

**Response:**

 $C_5H_{10}N_2O_7$  has a very low signal in the mass spectra and overlaps with the peak of  $C_5H_8NO_8$ , which is much higher. Therefore, to be cautious, we omit this compound in the revised manuscript as we are not confident with the assignment of this peak.

9. Scheme 2: Panel 1: -ONO2 is missed from the 2nd molecule, RO radical (C5H9N2O9.) should not be detected. Panel 2: the 3rd reaction stop should not be H-shift.

**Response:**

We thank the reviewer for pointing out our mis-labelling. We have corrected them in the revised manuscript. We agree that we did not detect RO radical ( $C_5H_9N_2O_9$ ). The molecule formula is only to help better track the mechanism. In the revised manuscript, we have marked the formula that we detected in bold. And we explicitly note this in the caption of Scheme 1-3.

10. Scheme 3: Panel 2: the structure of the final molecule maybe not correct.

**Response:**

Accepted.

In the revised manuscript, we have corrected this error.

11. How molecules with 7 H atoms formed? E.g. C5H7N2O9.

**Response:**

 $C_5H_7N_2O_9$ • is a peroxy radical, which can be formed from the reaction  $C_5H_7NO_4$  (C5-nitrooxycarbonyl) with NO3 and O2 as we discussed in lines 366-367 (original manuscript).

 $C_5H_7NO_4+NO_3\bullet+O_2 \rightarrow C_5H_7N_2O_9\bullet.$

Molecules with seven H atoms and one nitrogen atom  $(C_5H_7NO_n)$  can be formed via the reaction peroxy radicals containing eight H atoms, forming a ketone and an alcohol.

 $C_5H_8NO_{n+1} \bullet + C_5H_8NO_{n+1} \bullet \rightarrow C_5H_9NO_n + C_5H_7NO_n + O_2$

12. Is there any observational evidence on the formation of NO3-isoprene-HOM dimer and trimer in the real atmosphere?

**Response:**

We are not aware of field studies reporting NO3+isoprene-HOM dimers and trimers. This might be attributed to the difficulty to distinguish NO3+isoprene-HOM trimers from the dimers formed by cross reaction of the RO2 from monoterpene oxidation (C10-RO2) with that from isoprene oxidation (C5-RO2) as their molecular formula can be identical. Similarly, NO3+isoprene-HOM dimers can have the identical molecular formula to the HOM monomers from monoterpene oxidation. In one field study, possible contribution of dimer formation in isoprene oxidation to C6-10 HOM is discussed (Chen et al., 2020), although it is attributed to be more likely from monoterpene oxidation. Despite the absence of reporting in field observations, NO3+isoprene-HOM dimers have been observed in previous laboratory studies such as Ng et al. (2008) and Kwan et al. (2012) as we discussed in the manuscript.

**References**

Bianchi, F., Kurten, T., Riva, M., Mohr, C., Rissanen, M. P., Roldin, P., Berndt, T., Crounse, J. D., Wennberg,
P. O., Mentel, T. F., Wildt, J., Junninen, H., Jokinen, T., Kulmala, M., Worsnop, D. R., Thornton, J. A., Donahue,
N., Kjaergaard, H. G., and Ehn, M.: Highly Oxygenated Organic Molecules (HOM) from Gas-Phase
Autoxidation Involving Peroxy Radicals: A Key Contributor to Atmospheric Aerosol, Chem. Rev., 119, 3472-3509, 10.1021/acs.chemrev.8b00395, 2019.

Chen, Y. L., Takeuchi, M., Nah, T., Xu, L., Canagaratna, M. R., Stark, H., Baumann, K., Canonaco, F., Prevot, A. S. H., Huey, L. G., Weber, R. J., and Ng, N. L.: Chemical characterization of secondary organic aerosol at a rural site in the southeastern US: insights from simultaneous high-resolution time-of-flight aerosol mass spectrometer (HR-ToF-AMS) and FIGAERO chemical ionization mass spectrometer (CIMS) measurements, Atmos. Chem. Phys., 20, 8421-8440, 10.5194/acp-20-8421-2020, 2020.

Kwan, A. J., Chan, A. W. H., Ng, N. L., Kjaergaard, H. G., Seinfeld, J. H., and Wennberg, P. O.: Peroxy radical chemistry and OH radical production during the NO3-initiated oxidation of isoprene, Atmos. Chem. Phys., 12, 7499-7515, 10.5194/acp-12-7499-2012, 2012.

Malkin, T. L., Goddard, A., Heard, D. E., and Seakins, P. W.: Measurements of OH and HO2 yields from the gas phase ozonolysis of isoprene, Atmos. Chem. Phys., 10, 1441-1459, 10.5194/acp-10-1441-2010, 2010.

Ng, N. L., Kwan, A. J., Surratt, J. D., Chan, A. W. H., Chhabra, P. S., Sorooshian, A., Pye, H. O. T., Crounse, J. D., Wennberg, P. O., Flagan, R. C., and Seinfeld, J. H.: Secondary organic aerosol (SOA) formation from reaction of isoprene with nitrate radicals (NO3), Atmos. Chem. Phys., 8, 4117-4140, 10.5194/acp-8-4117-2008, 2008.

Yan, C., Nie, W., Aijala, M., Rissanen, M. P., Canagaratna, M. R., Massoli, P., Junninen, H., Jokinen, T., Sarnela, N., Hame, S. A. K., Schobesberger, S., Canonaco, F., Yao, L., Prevot, A. S. H., Petaja, T., Kulmala, M., Sipila, M., Worsnop, D. R., and Ehn, M.: Source characterization of highly oxidized multifunctional compounds in a boreal forest environment using positive matrix factorization, Atmos. Chem. Phys., 16, 12715-12731, 10.5194/acp-16-12715-2016, 2016.

Yan, C., Nie, W., Vogel, A. L., Dada, L., Lehtipalo, K., Stolzenburg, D., Wagner, R., Rissanen, M. P., Xiao, M., Ahonen, L., Fischer, L., Rose, C., Bianchi, F., Gordon, H., Simon, M., Heinritzi, M., Garmash, O., Roldin, P., Dias, A., Ye, P., Hofbauer, V., Amorim, A., Bauer, P. S., Bergen, A., Bernhammer, A. K., Breitenlechner, M., Brilke, S., Buchholz, A., Mazon, S. B., Canagaratna, M. R., Chen, X., Ding, A., Dommen, J., Draper, D. C., Duplissy, J., Frege, C., Heyn, C., Guida, R., Hakala, J., Heikkinen, L., Hoyle, C. R., Jokinen, T., Kangasluoma, J., Kirkby, J., Kontkanen, J., Kurten, A., Lawler, M. J., Mai, H., Mathot, S., Mauldin, R. L., Molteni, U., Nichman, L., Nieminen, T., Nowak, J., Ojdanic, A., Onnela, A., Pajunoja, A., Petaja, T., Piel, F., Quelever, L. L. J., Sarnela, N., Schallhart, S., Sengupta, K., Sipila, M., Tome, A., Trostl, J., Vaisanen, O., Wagner, A. C., Ylisirnio, A., Zha, Q., Baltensperger, U., Carslaw, K. S., Curtius, J., Flagan, R. C., Hansel, A., Riipinen, I., Smith, J. N., Virtanen, A., Winkler, P. M., Donahue, N. M., Kerminen, V. M., Kulmala, M., Ehn, M., and Worsnop, D. R.: Size-dependent influence of NOx on the growth rates of organic aerosol particles, Science Advances, 6, 9, 10.1126/sciadv.aay4945, 2020.

**Responses to Referee #2**

We thank the reviewer for the careful review of our manuscript. The comments and suggestions are greatly appreciated. All the comments have been addressed. In the following, please find our responses to the comments one by one and the corresponding revisions made to the manuscript. The original comments are shown in italics. The revised parts of the manuscript are highlighted.

Anonymous Referee #2

Received and published: 21 December 2020 General:

The authors investigate organic nitrates formed from the oxidation of isoprene with NO3 radicals, illustrate the formation mechanisms of these organic nitrates (including HOM monomer, dimer, and trimers), their yield, and their contribution to SOA yield. The study is well designed and the data are well presented. If the authors can address my points and questions below, I would recommend the publication of the manuscript in Atmospheric Chemistry and Physics.

Specific:

1. Line 46 – It's a bit ambiguous for "both nucleation and growth of SOA". HOM are im-portant in nucleation of gaseous vapours, and they contribute to the growth of aerosol particles. Maybe a bit better to say e.g. HOM play a pivotal role in the atmospheric nucleation and also particle growth on pre-existing particles (secondary organic aerosol, SOA).

**Response:**

Accepted.

In the revised manuscript, we have revised this sentence as follows.

"HOM play a pivotal role in the atmospheric nucleation and also particle growth of pre-existing particles thus contributing to secondary organic aerosol (SOA)."

2. Line 92 – How about the chemical lifetime of the reaction of isoprene with NO3?

**Response:**

The chemical lifetime of isoprene with respect to  $NO_3$  is calculated to be ~1.6 h and ~600 s using a  $NO_3$  concentration of 10 ppt and 100 ppt, respectively. We did not intend to compare the lifetime of isoprene with respect to OH and  $NO_3$ . In the revised manuscript, we have revised this sentence as follows.

"Although isoprene from plants are mainly emitted under light conditions, i.e., in the daytime, isoprene can remain high after sunset in significant concentrations (Starn et al., 1998; Stroud et al., 2002; Brown et al., 2009) because of the reduced consumption by OH and is found to decay rapidly."

3. Line 94 – How significant is the reaction of isoprene with NO3 contributing to NO3 loss at night? Is it dominating in isoprene-dominated region? How about in monoterpene-dominated region? And during the day, how does it compare to the isoprene oxidation with OH?

**Response:**

The contribution of the reaction of isoprene with NO3 to NO3 loss depends on VOC composition. According to a number of field studies, the reaction of isoprene can be the dominant NO3 loss channel in isoprene-dominated region, e.g. in Northeast US (Brown et al., 2009). In the monoterpene-dominated regions such as boreal forests, the reaction of isoprene with NO3 may be not the dominant loss of NO3. During most of the day, the reaction of isoprene with NO3 cannot compete with its reaction with OH due to the fast reaction of NO3 with NO and fast photolysis of NO3. In the late afternoon, the isoprene oxidation by NO3 can be comparable to that by OH under certain conditions e.g. reduced solar radiation and lower NO concentrations, and thus reduced OH concentration and reduced NO3 loss rate (Ayres et al., 2015; Hamilton et al., 2021).

In the revised manuscript, we have modified this sentence to further define the significance of the reaction of isoprene

with NO3 as follows.

"Regarding the budget of NO3, the reaction of isoprene with NO3 can contribute to a significant or even dominant fraction of NO3 loss at night in regions where VOC is dominated by isoprene such as Northeast US (Brown et al., 2009). Under some circumstances, the reaction of isoprene with NO3 can contribute to a significant fraction during the afternoon and afterwards (Ayres et al., 2015; Hamilton et al., 2021)."

4. Line 143 – Please add the RH to describe how dry the condition was, e.g. RH<2% or lower. Also add the temperature inside the SAPHIR chamber somewhere in the Experimental part, e.g. line 125.

**Response:**

Accepted.

In the revised manuscript, we have added the description of temperature and RH.

"Experiments were conducted under dry condition (RH<2 %) and temperature was at 302±3 K."

5. Line 158 – With a mass resolution of 4000, I am a bit curious is it difficult to distinguish different compounds at the same m/z (especially with one dominating compound), such as CHON1 from CHON3,5 compounds, CHON2 from CHON4 compounds, 1N-radicals from 2N-compounds, etc? It would be nicer to show a few masses of peak-fitting results of some organic nitrates in the supplement.

**Response:**

Accepted. We agree that it is not always easy to distinguish different compounds at the same m/z as their peaks overlap. Our approach is to "toggle" the inclusion of one peak and check the changes in the residue of peak fitting. If the toggling of inclusion of the peak does not significantly change the residue, we tend to not include the peak. Additionally, we also consider the double bond equivalence in the formula. For example, to distinguish CHON2 from CHON4, as in  $C_{10}H_{16}N_2O_{12}$ , and  $C_{10}H_{20}N_4O_{10}$  at the same unit m/z, we consider 1) it is less likely to form H20 in the reactions of isoprene with NO3; 2) it is unlikely to form a compound with N4O10 considering the functionality (e.g. a NO3- and a NO- group). When these peaks could not be distinguished, we did not include them in the peak list even if they may be actually present.

In the revised manuscript, we have shown some examples of peak-fitting results in the supplement.

6. Line 163 – Is the wall loss rate the same at different temperature in the SAPHIR chamber? Did you have the same chamber temperature as in Zhao et al., 2018?

**Response:**

The wall loss rate might be influenced by temperature in our chamber. It affects gas-wall equilibrium by changing evaporation rates and affects the condensation by changing diffusion in the boundary layer of the chamber. Because the reaction of  $NO_3$  with isoprene cannot be stopped instantaneously as easy as photo-oxidation reactions by switching off illumination, we cannot directly determine the vapor wall loss rate from the experiments themselves. The temperature in this study  $302\pm3$  K is not the same as but within the range of our previous photo-oxidation

experiments (298-314 K) (Zhao et al., 2018). Moreover, HOM yield is not sensitive to the vapor wall loss rate. An increase of wall loss rate by 100% or a decrease by 50% only changes the HOM yield by 11% and -6%, respectively. Therefore, we used the vapor loss rate determined in the photo-oxidation experiments. In the revised manuscript, we have discussed the influence of vapor wall loss rate as follows.

"Although the wall loss rate of vapors in this study might not be exactly the same as in our previous photo-oxidation experiments (Zhao et al., 2018), HOM yield is not sensitive to the vapor wall loss rate. An increase of vapor wall loss rate by 100% or a decrease by 50% only changes the HOM yield by 11% and -6%, respectively."

7. Line 195-197 – Why are the 2N-monomers dominating over 1N-monomers? Is it (partly) due to the reaction of NO3 radical with the remaining double bond of 1Nmonomer (forming 2N-monomers) being more reactive compared to the reaction of NO3 radical with the first double bond (C1) of isoprene (forming 1N-monomers)?

**Response:**

According to kinetic data of the reaction of NO3 with isoprene and with the first-generation products, isoprene hydroxy nitrate, isoprene carbonyl nitrate, and isoprene peroxy nitrate, the reaction of NO3 with the remaining double is slower than its reaction with isoprene (Wennberg et al., 2018). The higher abundance of HOM 2N-monomers than 1N-monomers is likely because HOM production rate via the autoxidation of 1N-RO2 following the reaction of isoprene with NO3 may be slower than that of the reaction of 1N-monomers (including both HOM and non-HOM monomers) with NO3. We would like to note that some less oxygenated 1N-monomers such as C5H9NO4/5 and C5H7NO4 may have high abundance but are not detected by NO3--CIMS and are not HOM and thus not included in HOM 1N-monomers.

In the revised manuscript, we have added the following discussion.

"The higher abundance of HOM 2N-monomers than 1N-monomers is likely because HOM production rate via the autoxidation of  $1N-RO_2$  following the reaction of isoprene with  $NO_3$  may be slower than that of the reaction of 1N-monomers (including both HOM and non-HOM monomers) with  $NO_3$ . We would like to note that some less oxygenated 1N-monomers such as  $C_5H_9NO_{4/5}$  and  $C_5H_7NO_4$  may have high abundance but are not detected by  $NO_3^-$ -CIMS and are not HOM and thus not included in HOM 1N-monomers."

8. Line 254-255 – If the fast loss of C5H9NO10 after isoprene addition is due to faster wall loss, why did the compound decay slower after 21:40 PM (which I think is partly due to wall loss) compared to those after isoprene addition? Maybe some other rea-sons are more important for its fast loss. Maybe it is similar to C5H9N2On radicals, that both its reactions and additionally wall loss contribute.

**Response:**

We meant the faster decay  $C_5H_9NO_{10}$  relative to  $C_5H_9NO_6$ . After 21:40 PM,  $C_5H_9NO_{10}$  showed a decay while  $C_5H_9NO_6$  did not.

In the revised manuscript, we have modified this sentence to avoid ambiguity.

"The faster loss of  $C_5H_9NO_{10}$  than  $C_5H_9NO_6$  may result from the faster wall loss due to its lower volatility."

9. Line 264-265 – If the peak intensity of C5H8NO7 radicals in Fig. S4 is plotted in log scale, maybe it's more obvious that it increased during isoprene addition? As far as in the current figure, I cannot see that clearly and can only see it continuously increasing over time. But it's more clear for C5H8NO8 radicals that is responding to isoprene addition.

**Response:**

The description of  $C_5H_8NO_7$ • and  $C_5H_8NO_8$ • was swapped by mistake and it is  $C_5H_8NO_7$ • that showed more mostly a time profile of second-generation products instead. In the revised manuscript, we have corrected this error and reorganized this paragraph as follows.

"The second-generation products may be different isomers formed in pathways other than Scheme 1. Second-generation  $C_5H_9NO_6$  can be formed via  $C_5H_8NO_7$ , which can also be formed by the reaction of NO3 and O2 with

 $C_5H_8O_2$  as mentioned above (Scheme S2b), or by the reaction of OH with  $C_5H_7NO_4$  (Scheme S2a). The time profiles of  $C_5H_8NO_7^{\bullet}$  did show more contribution of second-generation processes because it continuously increased with time in general. If the pathways via the reaction of NO3 and O2 with  $C_5H_8O_2$  and the reaction of OH with  $C_5H_7NO_4$ contribute most to  $C_5H_9NO_6$ ,  $C_5H_9NO_6$  would show mostly a time profile of second-generation products. Similarly, second-generation  $C_5H_9NO_7$  can be formed via  $C_5H_8NO_7^{\bullet}$  or  $C_5H_8NO_8^{\bullet}$ . The time series of  $C_5H_8NO_8^{\bullet}$  showed the contribution of both the first- and second-generation processes, which generally increased with time while also responding to isoprene addition (Fig. S4). Similar to  $C_5H_9NO_6$ , the second-generation pathway for  $C_5H_9NO_7$ ,  $C_5H_9NO_9$ , and  $C_5H_9NO_{10}$  are shown in Scheme S1, S3, S4. For the RO2 in  $C_5H_8NO_n^{\bullet}$  series other than  $C_5H_8NO_{7/8}^{\bullet}$ , the peak of  $C_5H_8NO_n^{\bullet}$  overlaps with  $C_5H_{10}N_2O_n$  in the mass spectra, which is a much larger peak, and thus cannot be easily differentiated from  $C_5H_{10}N_2O_n$ . Therefore, it is not possible to obtain reliable separate time profiles in order to differentiate their major sources. It is worth noting that nitrate CIMS may not be able to sensitively detect all isomers of  $C_5H_9NO_6$  due to the sensitivity limitation. Therefore, we cannot exclude the possibility that the absence of some first-generation isomers of  $C_5H_9NO_6$  was due to the low sensitivity of these isomers."

10. Line 267-270 - For the RO2 in C5H8NOn radicals series, do you mean the radicals with  $n \ge 9$  cannot be distinguished from 2N-monomers, but C5H8NO7 radicals and C5H8NO8 radicals (you shown in Fig. S4) can be distinguished because they don't have interference compounds nearby them?

**Response:**

Yes. To make it more clear, in the revised manuscript, we have modified this sentence as follows.

"For the RO2 in  $C_5H_8NO_n$ • series other than  $C_5H_8NO_{7/8}$ •, the peak of  $C_5H_8NO_n$ • overlaps with  $C_5H_{10}N_2O_n$  in the mass spectra, which is a much larger peak, and thus cannot be differentiated from  $C_5H_{10}N_2O_n$ ."

11. Line 303 – Other than SOAS, C5 organic nitrates (C5H7-11NO4-9) were also observed in both gas phase and particle phase with FIGAERO-CIMS in a rural area in Germany (Huang et al., EST, 2019), although their measurement site was not an isoprene-dominating region.

**Response:**

We thank the reviewer's reminder. In the revised manuscript, we have added this citation as follows.

" $C_5H_xNO_{4.9}$  and  $C_5H_xNO_{4.10}$  have been observed in the gas phase and particle, respectively, in a rural area in southwest Germany (Huang et al., 2019)."

12. Line 349-350 – "C5H9N2On radicals M2b were formed from C5H9NO4 followed by an alkoxy-peroxy step" is from Scheme 3b, not from Scheme 2.

**Response:**

Corrected.

13. Line 356 - It's not the case for C5H10N2On and C5H8N2On with n=7. With increasing oxygen number, they increased and then decreased.

**Response:**

We apologize that Fig. 3b was not updated. In the revised manuscript, we have updated the legend with  $C_5H_{10}N_2O_7$  omitted as we are not confident with their assignment. For  $C_5H_8N_2O_n$ , we have modified this sentence as follows.

"The intensity of  $C_5H_8N_2O_n$  first increased and then decreased with oxygen number while  $C_5H_{10}N_2O_n$  decreased with oxygen number with the  $C_5H_{10}N_2O_8$  and  $C_5H_8N_2O_8$  being the most abundant within their respective series."

14. Line 506 – For these 1N-dimers, have you checked the monoterpene concentration from PTR in the SAPHIR chamber? Is the monoterpene concentration in the chamber low enough not to produce any C10H16NOn molecules to interference/contaminate the results?

**Response:**

The monoterpene concentration in the chamber during the study is below the limit of detection, which is ~50 ppt ( $3\sigma$ ). Therefore, there is unlikely interference to C10H16NOn from monoterpenes. In the revised manuscript, we have added

the following note:

"We would like to note that there is unlikely interference to  $C_{10}$ -HOM from monoterpenes, which has been reported previously (Bernhammer et al., 2018), as the concentration of monoterpenes in the chamber during this study was below the limit of detection, which was ~50 ppt (3 $\sigma$ )."

15. Line 523-524 - Why only C5H8NOn radicals with n > 8 can react with isoprene? The C5H8NOn radicals detected (in Table 1) has an n range of 7-12, and the n range for C10H16NOn are expected to be between 9 and 14, based on R13. But in Table 2, the n range for C10H16NOn is between 10 and 16. Could you infer whether the smaller n (n <= 8) for C5H8NOn radicals can work or not to form C10H16NOn from R13? If not, could you give a possible explanation why the smaller n cannot work? And also how was C10H16NO16 formed?

**Response:**

We have further checked our data and the n range for  $C_{10}H_{16}NO_n$  should be 10-14. In the revised manuscript, we have corrected this. We expected the n range of  $C_{10}H_{16}NO_n$  to be 7-14 because besides the  $C_5H_8NO_n$  detected by our  $NO_3^-$ -CIMS, there should be  $C_5H_8NO_n(n=5,6)$  according to the reaction mechanism. Among these  $C_{10}H_{16}NO_n$  compounds,  $C_{10}H_{16}NO_{7-9}$  are expected to be detectable by  $NO_3^-$ -CIMS. The absence of these compounds in the mass spectra is likely attributed to their low concentration, which might result from low precursor concentration, low reaction rate with isoprene, and/or fast reaction with other radicals. In addition, we would like to note that the  $C_{10}H_{16}NO_n$  series has low signal in the mass spectra and their assignment and thus range of n may be subject to uncertainties. In the revised, we have added this note and revised this part as follows.

"Only  $C_{10}H_{16}NO_n^{\bullet}$  with n≥10 were detected, while according to the mechanism of self-reaction between  $C_5H_8NO_n^{\bullet}$ , the n range of  $C_{10}H_{16}NO_n^{\bullet}$  is expected to be 7-14. The absence of  $C_{10}H_{16}NO_{n(n<10)}^{\bullet}$  is likely attributed to their low abundance, which might result from low precursor concentrations, low reaction rates with isoprene, and/or faster reactive losses with other radicals."

"We note that due to their low signals in the mass spectra, their assignment and thus range of n may be subject to uncertainties."

16. Line 568-571 – Can you also check the sesquiterpene concentration in the chamber to exclude the contamination of its products?

**Response:**

The sesquiterpene concentration in the chamber during this study was below the limit of detection, which was  $\sim$ 50 ppt (3 $\sigma$ ). Therefore, there is unlikely contamination from sesquiterpene.

17. Line 603-604 – From the results, it's suggesting whether they were first-generation products or second-generation products, or a mix/combination of both, if I didn't misunderstand.

**Response:**

Yes. In the revised manuscript, we have revised this sentence as follows.

"The time profiles provide additional constraints on their formation mechanism beside the molecular formula, suggesting whether they were first-generation products, second-generation products or a combination of both." *Technical:*

1. Line 98 – "initials". Do you mean "initial"?

**Response:**

In the revised manuscript, we have changed it to "initial".

2. Line 116 – Change to "in the isoprene+NO3 system" throughout the manuscript.

**Response:**

In the revised manuscript, we have changed the phrase to "in the isoprene+NO3 reaction" throughout the manuscript. 3. *Line 123 – The first letter "s" in the word "system" is in Italic.*

Response: Corrected.

4. Line 148 – Can you plot the Fig. S1 similar to Fig. S2 so that easier to compare with Fig. S2? Or just label those additions directly in Fig. S2? Because there is no x axis in the SAPHIR box in the figure, but it seems the x axis is time.

**Response:**

Accepted. In the revised manuscript, we have added further labels in Fig. S2 to show the times of additions of isoprene, NO2, and O3.

5. Line 186 – Please specify in the Figure 1 caption that the m/z has included the reagent ion (15NO3-).

**Response: Accepted.**

6. Line 204 – (a) the labelled species in the figure 2 caption at m/z 351 and 353 should be C5H8N2O8 and C5H10N2O8 with H[15N]2O6-. Also in the figure itself, they are labeled as C5H8N2O14H[15N]2O6- and C5H10N2O14H[15N]2O6-. Please double check them and correct. (b) I suppose the number on the upper left corner of each compound is the m/z including the reagent ion (15NO3-). Please also specify that in the Fig. 2 caption. (c) What's the blue circle in the figure at m/z around 390?

**Response:** We thank the reviewer for pointing out this typo. We have corrected the captions and labeling in the figure, and also added the note of the m/z.

The blue circle is  $C_6F_{11}HO_3^-$ , coming from chamber wall. In the revised manuscript, we chose to not plot this compound in the figure to avoid confusion.

7. Line 211 –In Scheme 1 caption, it says (b) is for n = 8, 10, 12. But in the Scheme 1b itself, it's showing compounds with n = 4, 6, 8. Complete the Scheme 1b to show the compounds with n = 10 and 12.

**Response:**

Accepted. In the revised manuscript, we have added the pathway to form  $C_5H_9NO_{10}^{\bullet}$ . The peak of  $C_5H_9NO_{12}^{\bullet}$  overlaps with  $C_5H_9NO_8^{\bullet}H(^{15}NO_3)_2^{-}$ . After further checking the spectra, to be cautious we decided to not assign it. 8. *Line 235 – Remove the first "in"*.

Response: accepted.

9. Line 310 - n = 9, 11 are odd number of oxygen atoms, instead of even number.

**Response:**

We have corrected this error in the revised manuscript.

10. Line 339 – In Scheme 2a, -ONO2 is missed from the 2nd molecule.

**Response:**

We have corrected this mislabeling in the revised manuscript.

11. Line 345 – In Scheme 3b, -ONO2 is missed from the 2nd molecule.

**Response:**

We have corrected this mislabeling in the revised manuscript.

12. Line 372 – Change "VOCs" to "VOC". You have been using "VOC" previously in the manuscript. So make it consistent.

**Response:**

Accepted.

13. Line 385 – Could you separate these 3N-monomers in Fig. S8 into two panels, or use log scale? It's not obvious to see their time profile, especially for compounds with oxygen number bigger than 12, which were overlapping on top of each other.

**Response:**

Accepted. We use two panels to show them in the revised manuscript.

14. Line 390 – Change "Such an mechanism" to "Such a mechanism".

**Response: Accepted.**

15. Line 438-439 – Could you visualize the mass defect plots in Fig. 4 in a better way? Compound series were so close to each other to see clearly which compound belongs to which line. And also it's better to label the number of oxygen atoms to guide audience since you are discussing a lot of oxygen numbers. Both for Fig. 4 and Fig. 2. For Fig. S11, compound series were not so many and therefore not so difficult to distinguish.

**Response:** Accepted. In the revised manuscript, we have added the number of oxygen atoms. Also we have adjusted the scale of y axis, removed the grid line, and colored the HOM series lines in different colors to show the data points more clearly.

16. Line 443 – Since you have mentioned the compounds clustered with H[15N]206- were labelled in grey in Figure 2 caption. Do the same for Figure 4 caption for the compound in grey (C10H17N3O12-14H[15N]206-).

**Response: Accepted.**

17. Line 551-552 - Most abundant trimers were C15H24N4On (n=17-23), but in Fig. S11, n=16-22. Double check. Also double check all the n ranges for each compound series in the figures, tables, and text throughout the manuscript and supplement.

**Response:** We thank the reviewer for pointing this inconsistency. In the revised manuscript, we have corrected the numbers. Also we have checked throughout the manuscript for all series and fixed the inconsistency in the n ranges. *18. Line* 556 - The formation pathways of dimer RO2 C10H16N3On (n=14-20) and C10H17N2On are shown in R10 and R11.

**Response:**

In the revised manuscript, we have corrected this error.

**Responses to Referee #3**

We thank the reviewer for the careful review of our manuscript. The comments and suggestions are greatly appreciated. All the comments have been addressed. In the following, please find our responses to the comments one by one and the corresponding revisions made to the manuscript. The original comments are shown in italics. The revised parts of the manuscript are highlighted.

Anonymous Referee #3

Received and published: 21 December 2020

**General Comments**

This study identifies important HOMs (highly oxygenated organic molecules) from isoprene + NO3 reaction through chamber experiments. The identification of HOMs from NO3 oxidation have been less studied than those from OH or O3 oxidation, so this study fills an important gap in atmospheric chemistry. This study uniquely and in great detail connects many measured compounds to possible mechanistic formation pathways.

I suggest this paper be published with some minor revisions as specified below.

These minor revisions include some improvements to the mechanistic understanding and providing more information on how to interpret these laboratory results within the context of how SOA forms from isoprene + NO3 in the ambient atmosphere.

Specific Comments:

Page 5, 149. From the measurements of RO2, HO2, and NO3, can you approximate the fate of the RO2 radical in your experiment? Were conditions such that the RO2 predominantly reacted with another RO2, NO3, or HO2? Do you have an estimate of the lifetime of the RO2 radical in your experiments and how this compares to the RO2 lifetime in the ambient atmosphere. RO2 radical lifetime is often longer in the atmosphere compared to experiments. Would this possibly enhance the SOA yield for ambient conditions for HOMs?

**Response:**

In our experiments we measured radical concentrations directly. The reaction of  $RO_2$  with  $NO_3$  was determined to dominate over the reaction  $RO_2$  with  $RO_2$  and with  $HO_2$  and  $HO_2$  concentration, although the measured  $RO_2$  and  $HO_2$  are subject to uncertainties due to interference from  $NO_3$ . This is consistent with our understanding of the reaction system as there is no extra  $HO_2$  source (Vereecken et al., 2021). However, a recent study found that a large portion of  $RO_2$  is not measured by LIF and thus  $RO_2$  was underestimated (Vereecken et al. 2021). Therefore, we expect the reaction of  $RO_2$ +RO2 to be also important. Overall, we estimate that the  $RO_2$  fate is dominated the reaction  $RO_2$ + $NO_3$  with significant contribution of  $RO_2$ + $RO_2$ .

We also estimated  $RO_2$  lifetime using the measured  $RO_2$ ,  $NO_3$ , and  $HO_2$  concentrations. The  $RO_2$  lifetime is approximately 20-50 s in our experiments, which is generally comparable or shorter than the lifetime of  $RO_2$  in the ambient atmosphere at night, varying from several 10 s to several 100 s (Fry et al., 2018), depending on the  $NO_3$ ,  $HO_2$ , and  $RO_2$  concentrations as well as specific  $RO_2$ . Assuming a  $HO_2$ ,  $RO_2$ , and  $NO_3$  concentration of 5 ppt, 5 ppt (Tan et al., 2019), and 300 ppt (Brown and Stutz, 2012) respectively, the  $RO_2$  lifetime in our study is representative of nighttime RO2 lifetime of urban atmosphere.

If the RO2 lifetime increased, the autoxidation of RO2 would be more important relative to its bimolecular reactions. This may enhance HOM yield and thus enhance SOA yield. However, on the other hand, reduced rate of  $RO_2+RO_2$  at longer RO2 lifetime, producing low-volatility dimers, can reduce the SOA yield via reducing dimer yield (McFiggans et al., 2019; Pullinen et al., 2020).

In the revised manuscript, we have added discussion on the fate and lifetime of RO2.

"In these experiments,  $RO_2$  fate is estimated to be dominated by its reaction with  $NO_3$  according to the measured  $NO_3$ ,  $RO_2$ , and  $HO_2$  concentration and their rate constants for the reactions with  $RO_2$  (MCM v3.2(Jenkin et al., 1997; Jenkin et al., 2003; Saunders et al., 2003; Jenkin et al., 2015), via website: http://mcm.leeds.ac.uk/MCM) despite uncertainties of the measured  $RO_2$  and  $HO_2$  concentration due to interference from  $NO_3$ . As a large portion of  $RO_2$  is not measured by LIF (Vereecken et al. 2021) and thus  $RO_2$  is underestimated, we expected the reaction of  $RO_2$ +RO2 to be also important. Overall, we estimate that the  $RO_2$  fate is dominated the reaction  $RO_2$ +NO3 with significant contribution of  $RO_2$ +RO2."

"The RO2 fate is dominated the reaction  $RO_2$ +NO3 with significant contribution of  $RO_2$ +RO2, which can also represent the RO2 fate in the urban areas and areas influenced by urban plume. Yet, it cannot represent the chemistry in HO2-dominated regions such as clean forest environment (Schwantes et al., 2015)."

"The RO2 lifetime is approximately 20-50 s in our experiments, which is generally comparable or shorter than the lifetime of RO2 in the ambient atmosphere at night, varying from several 10 s to several 100 s (Fry et al., 2018), depending on the NO3, HO2, and RO2 concentrations. Assuming a HO2, RO2, and NO3 concentration of 5 ppt, 5 ppt (Tan et al., 2019), and 300 ppt (Brown and Stutz, 2012) respectively, the RO2 lifetime in our study is comparable to the nighttime RO2 lifetime (50 s) found in urban locations and areas influenced by urban plume. In areas with longer RO2 lifetime such as remote areas, the autoxidation of RO2 is expected to be more important relative to its bimolecular reactions. This may enhance HOM yield and thus enhance SOA yield. However, on the other hand, at lower RO2 concentration and thus longer RO2 lifetime, reduced rates of RO2+RO2 reactions producing low-volatility dimers can reduce the SOA yield via reducing dimer yield (McFiggans et al., 2019; Pullinen et al., 2020)."

Page 5, line 160. Please provide more detail here on using the H2SO4 sensitivity for the HOMs. Are there certain HOMs this assumption would apply more too? For example, does this assumption apply more to HOMs that are more oxygenated or have a higher C\*? Please specify the overall uncertainty in HOMs in the main text (It looks like you calculate this in the supplement). Is there need to add uncertainty here for using the H2SO4 sensitivity directly for the HOM sensitivity?

**Response:**

Accepted.

In the revised manuscript, we have added more details on the  $H_2SO_4$  sensitivity. Also we have added the overall uncertainty in HOM yield, which was shown in the supplement.

Since HOM contain more than six oxygen atoms and their cluster with nitrate ions are quite stable (Ehn et al., 2014),

the charge efficiency of HOM is thus assumed to be equal to that of  $H_2SO_4$ , which is close to the collision limit (Viggiano et al., 1997). We do not expect this sensitivity applies more to certain HOM than other HOM since HOM are all highly oxygenated with multiple functional groups. If HOM do not charge with nitrate ions at their collision limit or the cluster formed break during the short residence time in the charger, its concentration would be underestimated as pointed by Ehn et al. (2014). Thus, our assumption provides a lower limit of the HOM concentration. We have further discussed the uncertainty of using the sensitivity of  $H_2SO_4$  for HOM as follows.

"Since HOM contain more than six oxygen atoms and their clusters with nitrate ions are quite stable (Ehn et al., 2014), the charge efficiency of HOM is thus assumed to be equal to that of  $H_2SO_4$ , which is close to the collision limit (Viggiano et al., 1997). If HOM do not charge with nitrate ions at their collision limit or the clusters formed break during the short residence time in the charger, its concentration would be underestimated as pointed by Ehn et al. (2014). Thus, our assumption provides a lower limit of the HOM concentration."

Figure 1. Please add the names for the top m/z on panel b like done for panel a. It looks like many of the top m/z's are the same, but maybe some are unique. It's hard to compare by eye because the m/z lines are so small. Coloring the m/z label by their type listed in the pie chart would also be useful for the reader.

**Response:**

Accepted.

Page 11 line 292: Because you can measure OH and NO3, can you approximate how much isoprene and the firstgeneration NO3 nitrates react with OH versus NO3 in your experiments? This may lend insight into the products you are detecting. For example, the C5H802 compounds mentioned above seems more likely to form from OH oxidation than the H-shift in scheme S1a and S1b (Kwan 2012 Fig 5)? The reaction rate constant for the first-generation nitrates reaction with NO3 is low compared to OH rate constant (Wennberg 2018). From this information, can you connect how your laboratory results should be interpreted to the ambient atmosphere? For example, how long lived are NO3 derived first-generation nitrates in the ambient atmosphere are they likely to react again with NO3 or with OH at dawn?

**Response:**

In our experiments, the reaction of OH with isoprene contributed less to the isoprene consumption than the reaction of  $O_3$  with isoprene as the OH is mainly formed by isoprene+ $O_3$  and the OH yield is less than one. Therefore, the reaction of OH with isoprene is negligible (<3%) for isoprene loss. This is consistent with the contribution determined using measured OH concentration, despite some uncertainty of measured OH concentration due to the interference by NO3. In light of the negligible role of OH in isoprene consumption, we think that  $C_5H_8O_2$  is more likely formed by the reaction with NO3 and subsequent H-shift.

In the revised manuscript, we have discussed the role of OH in isoprene consumption.

"The contribution of the reaction of isoprene with trace amount of OH, mainly produced in the reaction of isoprene+ $O_3$  via Criegee intermediates (Nguyen et al., 2016), is negligible as the OH yield is less than one (Malkin et al., 2010) and thus its contribution is less than that of isoprene+ $O_3$ . This is consistent with the contribution

determined using measured OH concentration, despite some uncertainty in measured OH concentration due to the interference from NO3."

For the first-generation  $NO_3$  nitrates, their reaction rates with OH and with  $NO_3$  calculated using the reaction constants (Wennberg et al., 2018) and OH and  $NO_3$  concentrations are comparable, with both contributing significantly to the loss of first-generation  $NO_3$  nitrates as the rate constant with OH is much higher than that with  $NO_3$ .

Regarding the lifetime of first-generation nitrates in the ambient atmosphere, according their rate constants with OH and NO3 (Wennberg et al., 2018), their lifetime are 5 h and 1.3-4 h, respectively, with respect to the reaction with OH and NO3 assuming a typical OH concentration of  $2 \times 10^6$  molecules cm-3 (Lu et al., 2014; Tan et al., 2019) and NO3 concentration of 100-300 ppt in urban areas. Therefore, they likely react further with OH and NO3 at dawn. Therefore, our results are relevant to the ambient urban atmosphere and areas influenced by urban plumes.

In the revised manuscript, we have added discussion on the relevance to ambient atmosphere as follows.

"We observed the second-generation products formed by the reaction of first-generation products. The lifetime of first-generation nitrates in the ambient atmosphere, according their rate constants with OH and NO3 (Wennberg et al., 2018), are ~5 h and ~1.3-4 h, respectively, with respect to the reaction with OH and NO3 assuming a typical OH concentration of  $2 \times 10^6$  molecules cm-3 (Lu et al., 2014; Tan et al., 2019) and NO3 concentration of 100-300 ppt in urban areas (Brown and Stutz, 2012). Therefore, they have the chance to react further with OH and NO3 at dawn. In our experiments, the lifetimes of these first-generation nitrates with respect to OH and NO3 are comparable to the aforementioned lifetime due to comparable OH and NO3 concentrations with these ambient conditions. Therefore, our findings on the second-generation products are relevant to the ambient urban atmosphere and areas influenced by urban plumes. Some of these products such as C5H810N2O8 and multi-generation nitrooxyorganosulfates have been observed in recent field studies in polluted megacities in east China (Hamilton et al., 2021; Xu et al., 2021)."

"3N dimer such as C5H9N3O10 as well as 2N-monomers such as C5H8N2O8 and C5H8N2O10 have been observed in a recent field study in polluted cities in east China (Xu et al., 2021)."

Adding pictures of the molecules to schemes S1-S4 would be very beneficial for the reader.

**Response:**

Accepted. We would also like to note that these schemes and other schemes in this study only show example isomers and pathways to form these molecules. It is likely that many of the reactions occurring are not the dominant channels as otherwise there would be much higher HOM yield. We have added these notes in the revised manuscript.

"We would like to note that the scheme and other schemes in this study only show example isomers and pathways to form these molecules. It is likely that many of the reactions occurring are not the dominant channels as otherwise there would be much higher HOM yield as discussed below."

Page 13 line 350. Can you explain how this statement connects with these schemes more. I do not follow as both scheme 2 and scheme 3 have an example of a nitroxyhydroperoxide and a hydroxy nitrate? Also the likelihood of each pathway being relevant in your experiments seems more related to the RO2 fate (i.e., reaction with another RO2

or HO2) than with the loss rate of nitrooxy hydroperoxides and hydroxy nitrates in Ng et al., 2008. Can you include this into your explanation too?

**Response:**

We apologize that there is a typo here. We meant that the reactions with hydroxy nitrate is more likely. In the revised manuscript, we have modified this sentence. As the reviewer pointed out,  $RO_2$  fate is also an important factor, which affects the relative concentration of nitroxyhydroperoxide and hydroxy nitrate. In our experiments,  $RO_2$  fate is dominated by the reaction with  $NO_3$  and  $RO_2$ . Therefore, hydroxyl nitrate is expected to be higher than nitroxyhydroperoxide. Overall, we have revised the discussion as follows:

"Additionally, C5-hydroxynitrate concentration is expected to be higher than that of nitrooxyhydroperoxides because  $RO_2+RO_2$  forming alcohol is likely more important than  $RO_2+HO_2$  forming hydroperoxide in this study. Therefore, it is likely that C5H9N2On• M2a series was mainly formed from C5H9NO4 instead of C5H9NO5, while C5H9N2On• M2b were formed from C5H9NO4 followed by an alkoxy-peroxy step. That is, Scheme 2a and 3b were more likely."

Page 13 line 370: Is it also possible that instead of C5 nitrooxy carbonyls reacting more slowly with NO3 than C5 hydroxy nitrates that instead less HOMs are formed from C5 nitrooxy carbonyls because of the carbonyl group leading to more fragmentation (e.g., in MACR OH oxidation H-shifts lead to losing CO - Crounse 2012)? Have you considered this?

**Response:**

It is possible that more fragmentation in the further H-shift of peroxy radicals formed in the reaction of C5 nitrooxy carbonyls can also contribute to our observation of low abundance of  $C_5H_7N_2O_n$ . In the revised manuscript, we have added discussion on this point.

"This fact is consistent with the finding of Ng et al. (2008) that C5-nitrooxycarbonyls react slowly with NO3. Additionally, the peroxy radical formed in the reaction of C5-nitrooxycarbonyls with NO3 likely leads to more fragmentation in H-shift as found in the OH oxidation of methacrolein (Crounse et al., 2012), which may also contribute to the low abundance of  $C_5H_7N_2O_n$ ."

Page 16 line 447: The rate constants for RO2 + RO2 reaction are heavily structure dependent, so this assumption does not really hold in atmospheric chemistry. This should be considered here. For example, in schemes 2 and 3, the dominant RO2 isomers of C5H9N2O9 and C5H9N2O10 will not be the one pictured. The one pictured will most likely lead to HOMs. The dominant one will be the peroxy radical in the tertiary position, which will likely lead to fragmentation and not HOMs. This tertiary peroxy radical will react with other RO2 much more slowly than secondary or primary peroxy radicals (Jenkin 1998, (Jenkin et al., 1998)), so you would not necessarily expect very much ROOR from these RO2 radicals even though they are dominantly detected. Have you considered this?

**Response:**

We agree with the reviewer that tertiary RO2 formed may react slower than secondary and primary RO2.  $C_5H_9N_2O_9$ • or  $C_5H_9N_2O_{10}$ • is likely a mixture of different isomers, including both the secondary and tertiary RO2. And we have

no evidence that  $C_5H_9N_2O_9\bullet$  or  $C_5H_9N_2O_{10}\bullet$  contains more fraction of tertiary RO2 than primary and secondary RO2. For example, the dominant precursor of  $C_5H_9N_2O_9\bullet$ , isoprene hydroxyl nitrates, are likely (1-ONO2, 2-OH) and (1-ONO2, 4-OH) isoprene hydroxyl nitrate (IHN). The dominant RO2 formed by the reaction of these IHN with NO3 is likely secondary or tertiary RO2. In the revised manuscript, we have added discussion on different reaction rate of different RO2.

"Admittedly, the assumption of different  $RO_2$  having similar rate constant in accretion reactions may not be valid. For example, self-reaction of tertiary  $RO_2$  is slower than secondary and primary  $RO_2$  (Jenkin et al., 1998; Finlayson-Pitts and Pitts, 2000). Different rate constant may also lead to the observation that the most abundant dimers could not be explained the most abundant  $RO_2$ ."

Page 21 line 588: How was this HOM yield calculated? Is it from the first injection of isoprene or over the entire experiment?

**Response:**

HOM yield is calculated for the first isoprene addition period. In the revised manuscript, we have described this and used "primary HOM yield" in place of "HOM yield" to avoid ambiguity.

"The HOM yield in the oxidation of isoprene by NO3 was estimated for using the sensitivity of H2SO4. It was derived for the first isoprene addition period to minimize the contribution of multi-generation products and to better compare with the data in literature, thus denoted as primary HOM yield (Pullinen et al., 2020) and was estimated to be 1.2%  $^{+1.3\%}_{-7\%}$ ."

Conclusions: As related to the questions above, please include in more detail how to interpret these laboratory results within the context of how SOA forms from isoprene + NO3 in the ambient atmosphere. How do your laboratory conditions compare to the ambient atmosphere (e.g., RO2 fate (reaction with NO3, RO2, HO2, isomerize), RO2 lifetime, fate of the first-generation organic nitrates reaction with NO3 at night or OH at sunrise)?

**Response:**

Accepted. We have included more discussion on the ambient relevance of our laboratory study in the conclusion part as mentioned in the response to former comments.

Technical comments:

Scheme 2a: missing NO3 group on second molecule. In Scheme 2b, is the 3rd label really a H-shift? It looks like this should be reaction with RO2/NO3?

**Response:**

We thank the reviewer for pointing out our mis-labelings. In the revised manuscript, we have corrected them. *Scheme 3b: missing NO3 group on second molecule. And the last molecule OOH should be OH?*

**Response:**

We thank the reviewer for pointing out our mis-labelings. In the revised manuscript, we have corrected them. *Page 9 line 235 there are two "in"*

**Response:**

Corrected.

Figure S3, isoprene is spelled incorrectly.

**Response:**

Corrected.

**References:**

[revised manuscript text omitted]

**Dear editor,**

Our colleague Kristian Holten Møller made helpful remarks on our reaction schemes per private conversation. In the following, we show our responses to the comments one by one and the corresponding revisions made to the manuscript. The original comments are shown in italics. The revised parts of the manuscript are highlighted. We will therefore acknowledge Kristian's support in the paper.

Schemes 1a-3b: The arrows with NO3 should include O2 as well.

**Response:**

Accepted. We have made corrections in the revised manuscript.

Schemes 2-3: All NO3 additions shown seem to form the less substituted alkyl radical prior to O2-addition, which is likely the minor pathway (thought this may be ok for the purposes of the schemes).

**Response:**

We would like to note that the schemes in this study only show example isomers and pathways to form these HOM molecules. It is likely that many of the reactions occurring are not the dominant channels. Also it is challenging to determine which pathway is more likely than others within the scope of this study. For 1-NO3-isoprene-4-OO in Scheme 2 and 3, some of the more substituted peroxy radicals do not undergo further autoxidation, and thus we show the reactions of less substituted peroxy radical as an example. In the revised manuscript, we have also shown the pathways starting from 1-NO3-isoprene-2-OO peroxy radicals, which is indicated in a recent study by Vereecken et al. (2021) to be the dominant  $RO_2$  in the reaction of isoprene with NO3. In this pathway, more substituted alkyl radicals formed after the second NO3 addition undergo autoxidation (Scheme S6 and S7).

---

## Author Response (AR2)

Dear Editor,

We thank you for your helpful comments. We have addressed all these comments. In the following, please find our responses to the comments point by point and the corresponding revisions made to the manuscript. The original comments are shown in italics. The revised parts of the manuscript are highlighted.

*Dear authors,*

*Thank you for your detailed response. The reviewers comments have been sufficiently addressed. I only have a few minor comments that might be useful to take into consideration for completion of literature discussions:*

*1. Massoli et al. (ACS Earth Space Chem. 2018, 2, 7, 653–672) reported HOMs from SOAS. It would be useful to mention (or discuss if applicable) results from Massoli et al. in the context of the current study.*

**Response:**

Accepted. We have added this paper in our discussion in the revised manuscript. Accordingly, we have revised our discussion on the comparison with literature.

"A number of $C_5$ organic nitrates have been observed in field studies. For example, $C_5H_{7-11}NO_{6-8}$ and $C_5H_{7-11}NO_{4-9}$ have been observed in the gas phase (Massoli et al., 2018) and the particle phase (Lee et al., 2016; Chen et al., 2020), respectively in a rural area of southeast US, where isoprene is abundant. Xu et al. (2021) observed a number of $C_5$ 1N-HOM such as $C_5H_{7,9,11}NO_{6,7}$ in polluted megacities of Nanjing and Shanghai of east China during summer. While many of these HOM have daytime sources and are attributed to photo-oxidation in the presence of $NO_x$., nighttime oxidation with $NO_3$ also contribute to their formation (Lee et al., 2016; Chen et al., 2020; Xu et al., 2021). $C_5H_{7-11}NO_{4-9}$ were also observed in chamber experiments of the reaction of isoprene with OH in the presence of $NO_x$ (Lee et al., 2016). $C_5H_xNO_{4-9}$ and $C_5H_xNO_{4-10}$ have been also observed in the gas phase and particle phase, respectively, in a monoterpene-dominating rural area in southwest Germany (Huang et al., 2019)." (Sect. 3.2.2)

"2N-monomers have also been observed in previous field studies. For example, Massoli et al. (2018) observed $C_5H_{10}N_2O_{8-10}$ in rural Alabama US during the SOAS campaign. Xu et al. (2021) observed $C_5H_{8,10}N_2O_8$ and $C_5H_{10}N_2O_8$ in polluted megacities of Nanjing and Shanghai during summer." (Sect. 3.2.3)

*2. In response to the last comment by Reviewer #1 regarding isoprene dimers in the real atmosphere, it was noted that C6-C10 HOM were reported in Chen et al. (2020). While these species are more likely to arise from monoterpene oxidations, Chen et al. noted that other sources including dimer formation from isoprene is also possible. I think it would be useful to note this in the current manuscript to inform the readers as the community continue to elucidate the sources of these compounds in the atmosphere.*

**Response:**

Accepted.

In the revised manuscript, we have added the following discussion.

"Few field studies have reported HOM dimers formed via the reaction $NO_3$ with isoprene. This might be because

NO$_3$+isoprene-HOM dimers can have the identical molecular formula to the HOM monomers from monoterpene oxidation. Possible contribution of dimer formation in the isoprene oxidation to C6-10 HOM in the particle phase observed at a rural site Yorkville, US is reported by Chen et al. (2020), although these HOM are attributed to be more likely from monoterpene oxidation." (Sect. 3.3)

"We are not aware of field studies reporting NO$_3$+isoprene-HOM trimers, which is likely due to the same reason for dimers discussed above. It is challenging to distinguish HOM trimers formed in the reaction NO$_3$ with isoprene from the dimers formed by cross reaction of the RO$_2$ from monoterpene oxidation (C10-RO$_2$) with that from isoprene oxidation (C5-RO$_2$) as their molecular formula can be identical." (Sect. 3.4)

*3. On a related note, other than Huang et al. and Lee et al. (line 350-357), these species have also have reported in Massoli et al. and Chen et al.*

**Response:**

Accepted. We thank you for pointing out our overlook. In the revised manuscript, we have improved our discussion to also include these two studies as in our response to the comment #1.

Besides the revisions above, we have also corrected a few typos and format throughout the manuscript.